# SUMMR: Self-supervised Joint Representation Learning for Symmetric Multimodal Retrieval

## Abstract

Existing works on multimodality-to-multimodality (MM2MM) retrieval mainly focus on *asymmetric* retrieval, where text-image pairs in query and context serve distinct roles. In this work, we address the critical yet underexplored challenge of *symmetric* retrieval, where queries and contexts are interchangeable. We propose SUMMR, a novel two-stage self-supervised framework that leverages unlabeled web-scale image-text pairs, contrasting previous methods that heavily rely on costly supervised data. Based on the observation that both semantic alignment and discrepancies exist between the two modalities, we first learn a mask to disentangle shared and unique information within each image-text pair, allowing us to align the shared concepts while preserving modality-specific details. Then, we leverage this mask to automatically generate positive and negative samples for self-supervised contrastive learning of the final joint embedding. Complementing this framework, we introduce a novel benchmark featuring high-quality human-annotated positive and hard-negative pairs to evaluate symmetric MM2MM retrieval. On this benchmark, extensive experiments against *ten* SOTA methods show SUMMR surpasses the strongest supervised VLM by 3.42 points, with over 50x fewer model parameters and a 5x smaller embedding dimension. *Code will be available upon publication.*

## 1 Introduction

In information retrieval, the ability to combine different modalities for search is both essential and beneficial, as the cross-modal fusion provides more complete and nuanced representations. Yet, multimodal retrieval has received surprisingly little attention compared to single-modal tasks. One particularly challenging and overlooked task is symmetric multimodality-to-multimodality (MM2MM) retrieval. As illustrated in Fig. 1, current multimodal retrieval works mainly focus on asymmetric paradigms, where the query and content have distinct roles. In contrast, symmetric retrieval, where the query and content are semantically equivalent and interchangeable, is critical for many real-world applications. Consider an e-commerce scenario (Fig. 1d): a user searches with an image of a T-shirt's front and a description of its back. The desired result is an image of the back paired with a description of the front. To succeed, a model must grasp the holistic compositional meaning, recognizing that these two different multimodal pairs represent a single, coherent product. This requires inferring latent attributes not explicitly present in each modality - such as the T-shirt's color (white) or intended demographic (a boy) - to understand the full context. Beyond e-commerce, symmetric MM2MM retrieval has potential applications in areas such as news article retrieval, recipe recommendations, travel destination discovery, and interior design style matching, to name just a few. This highlights the urgent need for a comprehensive approach to symmetric MM2MM retrieval.

Unfortunately, the advancement of multimodal retrieval is not just slow, but fundamentally stalled by the reliance on a legacy paradigm: **supervised learning**. Methods ranging from late-fusion models like UniIR (Wei et al., 2024) to large VLM-based systems like VLM2Vec (Jiang et al., 2025) and MM-Embed (Lin et al., 2025) all depend on this approach. The very nuance that makes this task powerful also makes it nearly impossible to annotate at scale. As pointed out by Chen et al. (2023), manually curating high-quality datasets of positive and hard-negative pairs requires subtle human judgment about semantic equivalence, a process that is prohibitively expensive and time-consuming. Even recent attempts to automate this process via data synthesis (Zhang et al., 2024) are limited by the capabilities of the generative models used and struggle with filtering low-quality examples. This data bottleneck acts as a critical roadblock. Moreover, this reliance on small-scale, meticulously

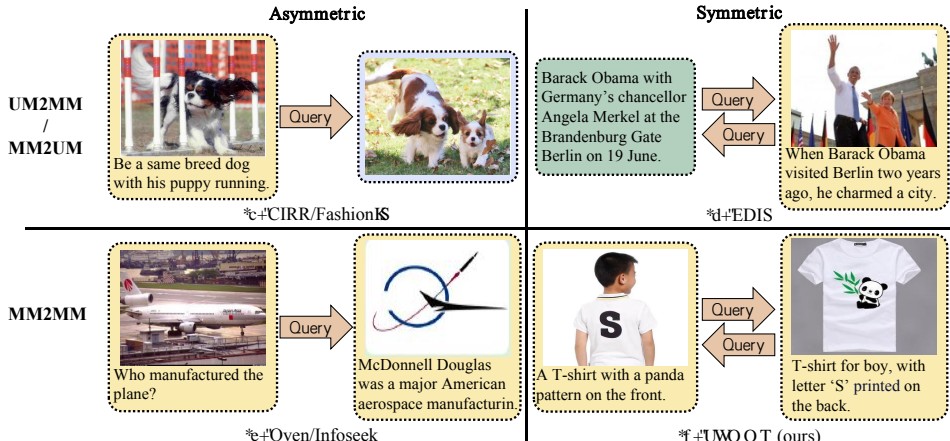

Figure 1: A comparison of existing multimodal retrieval paradigms with the symmetric MM2MM task addressed in this paper. Tasks are categorized based on two properties: whether the retrieval is symmetric (query and content are interchangeable) and whether both are multimodal (MM2MM vs. UM2MM/MM2UM).

curated datasets runs counter to a dominant trend in modern AI: models trained on massive, web-scale data consistently outperform those trained on smaller, specialized datasets in domains ranging from embedding models (e.g., CLIP (Radford et al., 2021) and DINO (Oquab et al., 2024)) to generative models (e.g., LLMs (Hurst et al., 2024) and Stable Diffusion (Rombach et al., 2022)). The field is thus stuck, not for a lack of powerful model architectures, but from the inability to teach them effectively and at scale.

This work breaks the deadlock by introducing a new, self-supervised paradigm designed specifically for symmetric MM2MM retrieval. Our approach bypasses the annotation bottleneck by exploiting the unique structure of the symmetric task itself. We build on a foundational insight: because a query and its positive counterpart are semantically equivalent, any unlabeled image-text pair from the web can serve as a source of supervision. We posit that such a pair contains both shared concepts (the "intersection") and modality-specific details (the "difference"). This allows us to programmatically generate training data: **negative samples** are created by masking unique details (the difference), causing an irrecoverable information loss, while **positive samples** are created by masking shared concepts (the intersection) from one modality, as the full meaning can be reconstructed from the combined context. We realize this principle in SUMMR, a two-stage self-supervised framework that first learns to identify this intersection and then leverages it to train a final, highly effective joint embedding—all without a single manual label. On a **novel, challenging benchmark** we introduce for this task, SUMMR outperforms *ten* leading universal multimodal embedding models, surpassing the strongest VLM by 3.42 points with over 50x fewer model parameters and a 5x smaller embedding dimension, demonstrating that SOTA performance is achievable without any human-annotated data. The core contributions of this paper are threefold:

- The first systematic investigation of the symmetric MM2MM retrieval task, for which we propose a novel two-stage self-supervised framework, SUMMR, that operates on unlabeled web-scale data.
- A unique disentanglement method that learns to identify and separate the shared (intersection) and unique (difference) information between modalities, enabling the automatic generation of positive and hard-negative samples for robust contrastive learning. Experimental results show that our method outperforms the previous VLM-based SOTA method by 3.42 points, despite being much smaller in model size and embedding dimensions, and without any labeled training data.
- A new high-quality benchmark dataset and evaluation pipeline, featuring human-annotated pairs, to facilitate rigorous and realistic evaluation of this newly formalized task.

## 2 RELATED WORKS

Multimodal retrieval has evolved to handle increasingly complex queries and content, with systems typically categorized by the modalities involved: from unimodal queries to multimodal results (UM2MM) (Liu et al., 2023; Chang et al., 2022) and vice-versa (MM2UM) (Hu et al., 2023; Chen

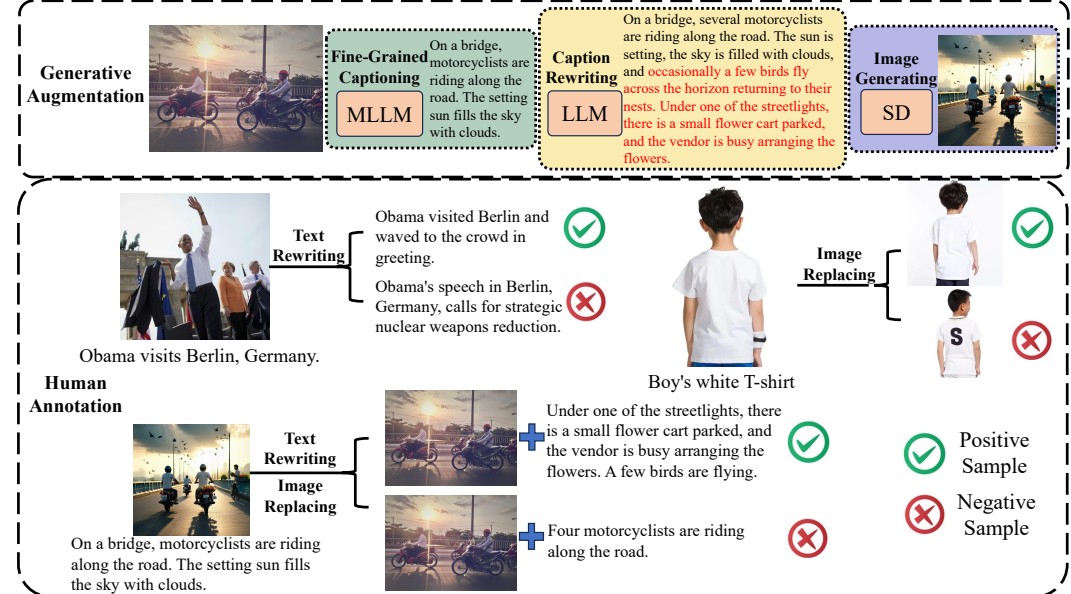

Figure 2: The data augmentation and annotation pipeline. To collect the hard samples, we employ an image editing process based on VLMs, LLMs and SDs. Subsequently, symmetric positive-negative pairs are manually annotated through text rewriting and image replacing, ensuring mutual consistency in both query-positive and query-negative interactions. More examples are provided in App. B

et al., 2023; Wu et al., 2021; Liu et al., 2021; Zhang et al., 2024). While some recent work has explored multimodal-to-multimodal (MM2MM) retrieval (Hu et al., 2023; Chen et al., 2023), these approaches predominantly focus on asymmetric tasks like visual question answering, where the query and retrieved document serve distinct, non-interchangeable roles. This focus has left a critical gap in addressing symmetric retrieval scenarios - common in applications like e-commerce, content-based recommendation, or design matching - where a query and its corresponding document are semantically equivalent and swappable. Our work is the first to systematically investigate symmetric MM2MM, motivating the development of a dedicated benchmark and a novel learning framework tailored to its unique challenges.

To learn a joint representation, existing methods generally employ either score fusion or feature fusion architectures. Score fusion combines unimodal scores via simple averaging but often fails to capture complex cross-modal interactions (Zhou et al., 2025; Wei et al., 2024). Feature fusion offers deeper integration, either through computationally expensive VLMs (Lin et al., 2025; Jiang et al., 2025; Liu et al., 2025; Zhang et al., 2025; Gu et al., 2025; Kong et al., 2025; Chen et al., 2025) or more efficient late-fusion architectures that use a dedicated encoder to fuse unimodal features (Wei et al., 2024; Zhang et al., 2024). A significant bottleneck for most SOTA methods is their reliance on supervised learning, which requires costly, manually annotated query-document pairs that are particularly scarce for MM2MM tasks. In contrast, our proposed framework, SUMMR, leverages an efficient late-fusion architecture and introduces a novel self-supervised learning strategy. This allows us to train on readily available, unlabeled image-text pairs, overcoming the dual challenges of data dependency and computational cost. A more detailed literature review is provided in App. A.

## 3 SYMMETRIC MM2MM RETRIEVAL TASK

**Problem Formulation:** We formulate the symmetric MM2MM (*sym-MM2MM*) retrieval task as learning an encoder $\mathcal{E}$ that maps an image-text sample $\boldsymbol{X} = (\boldsymbol{I}, \boldsymbol{T})$ to an embedding vector $\boldsymbol{f} = \mathcal{E}(\boldsymbol{I}, \boldsymbol{T})$. For a given query embedding $\boldsymbol{f}$, its cosine similarity with a positive counterpart $\boldsymbol{f}^+$ must exceed that of a negative one $\boldsymbol{f}^-$, that is, $\langle \boldsymbol{f}^+, \boldsymbol{f} \rangle > \langle \boldsymbol{f}^-, \boldsymbol{f} \rangle$, where $\langle \cdot, \cdot \rangle$ is cosine similarity.

**Symmetric MM2MM (*sym-MM2MM*) Retrieval Benchmark:** The task of symmetric MM2MM retrieval remains largely unexplored. Consequently, existing benchmarks such as OVEN (Hu et al., 2023) and InfoSeek (Chen et al., 2023), which are primarily designed for asymmetric search, are unsuitable for its evaluation. To address this gap and enable robust evaluation, we introduce a new

benchmark, named *sym-MM2MM*. We define a high-quality positive pair as a query and a candidate with consistent joint semantics but different content, and a hard-negative pair as two samples with similar content but inconsistent joint semantics.

We construct this benchmark using a scalable, model-assisted framework that leverages a Vision-Language Model (VLM), a Large Language Model (LLM), and a text-to-image diffusion model (Stable Diffusion or SD 3.5). As shown in Fig. 2, our pipeline first performs generative augmentation. Starting with source images from COCO (Lin et al., 2014), LAION (Schuhmann et al., 2022), and WuKong (Gu et al., 2022), a VLM (GPT-4o) generates a fine-grained description $T_{orig}$. An LLM (GPT-4o) then rewrites this into a modified description $T_{mod}$ by adding or removing objects to create a controlled information gap. Next, SD 3.5 synthesizes a new image $I_{mod}$ from $T_{mod}$. This process yields a positive pair by combining $(I_{orig}, T_{mod})$ and $(I_{mod}, T_{orig})$, while other combinations can form hard negatives. These automatically generated pairs, however, only serves as a candidate pool. To ensure the quality and difficulty required for a robust benchmark, these pairs undergo a rigorous, multi-stage human curation process. This curation involves two main activities. First, annotators meticulously verify the machine-generated pairs. This is a highly stringent process: nearly 50% of the candidate 'positive pairs' were rejected due to semantic inconsistencies where the synthesized image failed to accurately reflect the text. Beyond simple verification, for the accepted pairs, annotators would often perform further edits, such as rewriting text to increase difficulty. Second, a substantial portion of the benchmark was created entirely by our annotators through direct manual curation, constructing challenging positive and negative pairs from the original samples via text rewriting or image replacement, as illustrated in Fig. 2. This intensive process was conducted by two annotators, each dedicating approximately 20 hours to the task. In total, this effort yielded our final benchmark of 214 high-quality triplets $(X, X^+, X^-)$, each containing an original sample, its positive counterpart, and a hard-negative. The benchmark's modest size is a direct reflection of the core challenge that our paper seeks to address: high-quality, human-annotated sym-MM2MM data is extremely difficult and expensive to create. Note that to construct more challenging test cases, we occasionally introduce a variant $X'$ for some $X$, which is created by manually modifying the text of the original sample, as Fig. 5 shows. It is also worthy to mention that during annotation, we select the sample belong to different categories to increase the diversity, with the final category distribution shown in Fig. 6.

For evaluation, we employ two complementary metrics. First, we use the standard **Recall@k** metric, where queries are used to retrieve from a large candidate pool of 1M LAION pairs plus our benchmark pairs. As this large pool may contain false negatives that skew results, we also introduce a more accurate **Precision** score. This metric is calculated exclusively on our curated benchmark pairs, measuring the model's ability to correctly rank the positive sample above the hard-negative one. It is defined as $1/N \sum_{i=1}^{N} 1[\langle f_i^+, f_i \rangle > \langle f_i^-, f_i \rangle]$ (1[·] is the indicator function, for samples with modified $X'$, we require $1/N \sum_{i=1}^{N} 1[\langle f_i^+, f_i \rangle > \langle f_i^-, f_i' \rangle]$). We adopt both metrics to provide a comprehensive assessment: Recall@k evaluates performance in a realistic large-scale scenario, while the Precision score offers a more precise measure of a model's discriminative power.

## 4 THE PROPOSED FRAMEWORK: SUMMR

Our proposed framework, SUMMR, jointly extracts features from multimodal samples via five components: a vision encoder $\mathcal{E}_V$ and a language encoder $\mathcal{E}_L$ to process raw inputs; two-layer MLP adapters $(\mathcal{A}_V, \mathcal{A}_L)$ that project the unimodal features into a shared latent space; and a Vision-Language (VL) encoder $\mathcal{E}_{VL}$, composed of three attention layers, to model cross-modal interactions.

As illustrated in Fig. 3, for an input image-text pair, $X = (I, T)$, the unimodal encoders and adapters produce patch-level visual features $V$ and token-level textual features $L$, i.e., $V = \mathcal{A}_V[\mathcal{E}_V(I)]$ and $L = \mathcal{A}_L[\mathcal{E}_L(T)]$. The features excluding the original [CLS] tokens, denoted as $V'$ and $L'$, are concatenated with a new, learnable [CLS] token. This combined sequence is fed into the VL-encoder, whose output corresponding to the new [CLS] token serves as the final multimodal embedding $f = \mathcal{E}_{VL}([V', L', [CLS]])$. Here, the original [CLS] tokens from the unimodal encoders $(v_{cls}, l_{cls})$ represent unimodal global features, whereas $f$ represents the joint multimodal global feature.

SUMMR is trained via a novel two-stage self-supervised approach. The first stage is dedicated to learning a reliable "intersection mask" that can disentangle the shared semantic concepts from the unique, modality-specific details within any given image-text pair. In the second stage, this mask is leveraged to automatically generate positive and hard-negative samples for contrastive learning. This

Figure 3: Architectural overview of Stage 1: Disentanglement and Alignment. The Mask Generation module (MaskGen), guided by an alignment signal from Global-to-Local Alignment ($\mathcal{L}_{\text{GLA}}$) and Local Distillation ($\mathcal{L}_{\text{LD}}$), disentangles shared (intersection) and unique (difference) information. The resulting mask orchestrates two key objectives: Masked Image-Text Contrastive ($\mathcal{L}_{\text{ITC}}$) learning aligns the shared concepts, while Global Distillation ($\mathcal{L}_{\text{GD}}$) preserves the original unimodal information.

separation of concerns—first learning how to disentangle, then using that knowledge to perfect the final representation—allows the model to focus singularly on the retrieval task in the second stage. This process distills the complex, multi-task logic from Stage 1 into a more powerful and specialized final encoder.

## 4.1 STAGE 1: DISENTANGLE AND ALIGN MULTIMODAL INFORMATION

The primary goal of this stage is to disentangle intersection from difference by learning to generate an intersection mask for any image-text pair. Meanwhile, we also aim to train the newly initialized adapters ($\mathcal{A}_V$ and $\mathcal{A}_L$) and the VL-Encoder ($\mathcal{E}_{VL}$) from scratch, while fine-tuning the base encoders using LoRA (Hu et al., 2022), all without relying on manual labels. As shown in Fig. 3, this is achieved through a synergistic process. Global-to-Local Alignment and Local Distillation produce a reliable alignment signal, which the Mask Generation (MaskGen) module uses to derive an intersection mask via Quadratic Discriminant Analysis (Hastie et al., 2009) (QDA) and an evolutionary schedule. This mask then directs the primary training objectives: a Masked Image-Text Contrastive loss aligns the shared intersection, while a Global Distillation loss acts as a counterbalance, preserving each modality's complete, unmasked information. We detail these mechanisms below.

### 4.1.1 DISTILLATION-GUIDED LOCAL ALIGNMENT

To generate the intersection mask, the model needs to first learn a quantifiable signal to distinguish shared from unique features. We achieve this through two complementary processes.

**Global-to-Local Alignment (GLA):** We start from the core assumption that local features belonging to the intersection set should have a higher similarity to the global representation of the partner modality than features from the difference set. For instance, image patches of a "dog" should be more similar to the global text embedding of "a photo of a dog" than background patches of "grass".

To realize this, we use a margin loss on in-batch similarities. For an image-text pair, we want the average similarity between its local features and the global feature of its partner modality to be higher than the average similarity with any negative pair in the batch. We use the mean similarity because we only expect a statistical tendency; not every local feature will align in a positive pair, and some incidental overlap may occur in negative pairs. For global-text-to-local-image alignment, the loss is:

$$\mathcal{L}_{L2V} = \left[\text{mean}(\mathbb{S}_{L2V}^-) + \delta - \text{mean}(\mathbb{S}_{L2V}^+)\right]_+ , \tag{1}$$

where $\mathbb{S}_{L2V}^+$ and $\mathbb{S}_{L2V}^-$ are the sets of positive and negative pair similarities respectively, $\delta$ is the margin, and $[x]_+ = \max(0, x)$. A symmetric loss, $\mathcal{L}_{V2L}$, is computed for global-image-to-local-text similarities, and the total loss is $\mathcal{L}_{\text{GLA}} = \mathcal{L}_{L2V} + \mathcal{L}_{V2L}$. This loss teaches the model a general principle of alignment, which, when applied to a single image-text pair, naturally produces higher similarity scores for their intersection features than for the difference features.

**Local Unimodal Distillation (LD)**: The quality of the GLA signal depends entirely on the quality of the local features themselves. To ensure these features are rich and semantically structured, we introduce a distillation loss. We use powerful, pre-trained unimodal encoders (e.g., DINOv2 for vision, BGE-m3 (Chen et al., 2024) for text) as teachers and encourage our student model's local features $(\boldsymbol{L}', \boldsymbol{V}')$ to mimic the teacher's intra-modal relational structure.

Instead of matching feature vectors directly, we match the ranking of similarities between tokens, which preserves representational flexibility while focusing on what matters most—the semantic relationships. The loss minimizes the dissimilarity between the student's and teacher's local similarity rankings using Pearson correlation: $\mathcal{L}_{\text{LD}}^{L} = 1 - 1/N \sum_{k=1}^{N} \text{corr}(\boldsymbol{S}_k^{\mathcal{T}}, \boldsymbol{S}_k)$, where $\boldsymbol{S}_k$ and $\boldsymbol{S}_k^{\mathcal{T}}$ are the similarity vectors of the $k$-th token to all other tokens for the student and teacher. A symmetric loss $\mathcal{L}_{\text{LD}}^{V}$ is used for vision, with the total loss being $\mathcal{L}_{\text{LD}} = \mathcal{L}_{\text{LD}}^{L} + \mathcal{L}_{\text{LD}}^{V}$. This distillation is essential; it stabilizes training by providing a strong unimodal signal, preventing the adapters from collapsing or learning poor representations based only on noisy cross-modal signals. From this similarity signal, the MaskGen module then derives the intersection mask.

### 4.1.2 EVOLUTIONARY MASKING VIA ADAPTIVE THRESHOLDING

**Adaptive Thresholding via QDA**: The GLA process yields two distributions of similarity scores for each modality: one for positive (likely intersection) pairs and one for negative (likely difference) pairs. To find an optimal threshold $\tau$ to separate them, we model each distribution as a Gaussian and find their intersection point using the one-dimensional QDA. This is equivalent to solving for $\tau$ in $\mathcal{N}(\tau; \mu^+, (\sigma^+)^2) = \mathcal{N}(\tau; \mu^-, (\sigma^-)^2)$. This method provides a principled, data-driven threshold that adapts as the model's encoders improve during training. Using this threshold, we can generate a hard binary mask ($\hat{\boldsymbol{M}}_V = \boldsymbol{S}_{L2V} > \tau_V$, $\hat{\boldsymbol{M}}_L = \boldsymbol{S}_{V2L} > \tau_L$) that identifies the intersection set for any given image-text pair. The detailed derivation of $\tau$ is provided in App. C.1.

**Evolutionary Masking**: At the beginning of training, the encoders are not aligned and the estimated mask $\hat{\boldsymbol{M}}$ is unreliable. Applying a noisy hard mask at this stage could be catastrophic. We therefore introduce an evolutionary mask $\boldsymbol{M}$ that smoothly transitions from a non-informative, all-ones mask to the model's estimated hard mask, that is, $\boldsymbol{M} = \rho \mathbf{1} + (1 - \rho)\hat{\boldsymbol{M}}$, where $\mathbf{1}$ is an all-ones mask the annealing schedule $\rho$ decreases from 1 to 0 during training. This strategy allows the model to first learn from all features and then gradually rely on its own increasingly confident predictions, ensuring training stability, a choice validated in our ablation studies (see Sec. 5 and App. F.1). As a consequence, the evolutionary mask is a tool that can be used to achieve our two main goals: robustly aligning the intersection set while preserving the difference set.

### 4.1.3 MASKED CONTRASTIVE ALIGNMENT AND GLOBAL INFORMATION PRESERVATION

**Masked Image-Text Contrastive (ITC) Alignment**: To align the intersection set, we apply a contrastive loss, but with a critical modification. We first use the evolutionary mask to select only the local features from the intersection set. These masked features are then fed into our VL-Encoder $\mathcal{E}_{VL}$ to produce refined, cross-modal global embeddings:

$$\boldsymbol{f}_V = \boldsymbol{W}_V(\mathcal{E}_{VL}([\boldsymbol{V}', [\text{CLS}]], \boldsymbol{M}_V)), \quad \boldsymbol{f}_L = \boldsymbol{W}_L(\mathcal{E}_{VL}([\boldsymbol{L}', [\text{CLS}]], \boldsymbol{M}_L)), \tag{2}$$

where the masks $\boldsymbol{M}_V$ and $\boldsymbol{M}_L$ are applied within the self-attention layers of $\mathcal{E}_{VL}$ to prevent the [CLS] token from attending to masked-out (difference) tokens. The lightweight projection heads $(\boldsymbol{W}_V, \boldsymbol{W}_L)$ are also crucial. The mask only handles spatial alignment (which patches align with which tokens), but feature-level mismatches can remain. For example, in an image of a "red apple" paired with the text "an apple," the mask correctly aligns the apple region, but the visual feature for "red" is unaligned. The projection heads resolve this conflict by learning to filter out such unalignable feature details specifically for the contrastive loss calculation, allowing $\mathcal{E}_{VL}$ itself to produce a rich joint embedding that preserves all information from the intersection, including "red." The standard InfoNCE loss is then applied to these projected embeddings:

$$\mathcal{L}_{\text{ITC}} = -\frac{1}{2B}\left(\sum_{i}^{B} \log \frac{\exp(\langle \boldsymbol{f}_V^i, \boldsymbol{f}_L^i \rangle / \eta)}{\sum_{j}^{B} \exp(\langle \boldsymbol{f}_V^i, \boldsymbol{f}_L^j \rangle / \eta)} + \sum_{i}^{B} \log \frac{\exp(\langle \boldsymbol{f}_L^i, \boldsymbol{f}_V^i \rangle / \eta)}{\sum_{j}^{B} \exp(\langle \boldsymbol{f}_L^i, \boldsymbol{f}_V^j \rangle / \eta)}\right). \tag{3}$$

This masked alignment is the driving force of our method. It provides a direct supervision signal: "select masks $\boldsymbol{M}_V$ and $\boldsymbol{M}_L$ such that the remaining (intersection) information is sufficient to make the two modalities globally aligned." This refines both the mask generation process and trains $\mathcal{E}_{VL}$ to be an expert at summarizing shared concepts.

**Global Unimodal Distillation (GD)**: While Masked ITC focuses the model on alignment, it intentionally ignores the difference set. To prevent the model from forgetting this crucial modality-specific information, we introduce a complementary global distillation loss. This acts as a counterbalance, ensuring the complete, un-masked representations remain faithful to the original semantics. We use the same powerful teacher encoders as in LD and require our student's un-masked global embeddings to have a similar relational structure to the teacher's embeddings. As with LD, we match batch-wise similarity rankings, that is, $\mathcal{L}_{\text{GD}}^{L} = 1 - \text{corr}(\boldsymbol{S}^{\mathcal{T}}, \boldsymbol{S})$, where $\boldsymbol{S}$ and $\boldsymbol{S}^{\mathcal{T}}$ are the global similarity matrices for the student and teacher. The total loss $\mathcal{L}_{\text{GD}} = \mathcal{L}_{\text{GD}}^{L} + \mathcal{L}_{\text{GD}}^{V}$ ensures that the model preserves the information in the difference set, even though it is not used for contrastive alignment.

### 4.1.4 THE SYNERGISTIC TRAINING OBJECTIVE

The total training objective for Stage 1 combines all these components:

$$\mathcal{L} = \mathcal{L}_{\text{ITC}} + \lambda_1 \mathcal{L}_{\text{GLA}} + \lambda_2 \mathcal{L}_{\text{GD}} + \lambda_3 \mathcal{L}_{\text{LD}}. \tag{4}$$

where $\lambda_1$, $\lambda_2$ and $\lambda_3$ are the loss weights. While the alignment objective ($\mathcal{L}_{\text{ITC}}$) and preservation objective ($\mathcal{L}_{\text{GD}}$) create an intentional tension, our framework resolves it architecturally. The evolutionary mask and projection heads work in tandem to disentangle these conflicting goals at the spatial and feature levels, respectively. These components form a synergistic, self-correcting loop where each part is essential, as empirically validated by our ablation studies (Tab. 5, Fig. 4). Local Distillation ensures high-quality local features, which allows Global-to-Local Alignment to generate a clean signal separating shared and unique concepts. This signal is converted into a progressively reliable intersection mask via adaptive QDA thresholding and evolutionary annealing. The Masked ITC loss leverages this mask to perform a highly focused alignment on the intersection set, which provides a powerful supervisory signal that refines the entire mask generation process. Finally, Global Distillation acts as a crucial counterbalance, preserving the rich, modality-specific information that is temporarily masked out for alignment. By the end of this stage, the model has learned not only to generate an intersection mask but has also developed encoders trained to both align shared concepts and preserve complete unimodal information. As a result, the model is ready to produce a high-quality, comprehensive joint embedding for any image-text pair.

### 4.2 STAGE 2: SELF-SUPERVISED MULTI-MODAL REPRESENTATION LEARNING

In Stage 2 (see Fig. 7), we learn the final joint embedding via self-supervised contrastive learning, powered by automatically constructed training samples. This process leverages the disentanglement learned in Stage 1 to distinguish between shared (intersection) and unique (difference) information. A *positive sample* is created by masking the intersection (e.g., the "black bear" in Fig. 3). This preserves the pair's core semantic identity as the essential information remains recoverable from the other modality—analogous to how a cropped image is a valid positive in standard contrastive learning. Conversely, a *hard-negative sample* is formed by masking the difference (e.g., the image background or the text "to greet visitors"), which causes an irrecoverable loss of information. Specifically, for a given image-text pair, $\boldsymbol{X}^{\{i\}} = (\boldsymbol{I}^{\{i\}}, \boldsymbol{T}^{\{i\}})$, we first use the trained model from Stage 1 to calculate the local similarity scores ($\boldsymbol{S}_{L2V}$ and $\boldsymbol{S}_{V2L}$) and the adaptive thresholds ($\tau_V$ and $\tau_L$), as detailed in Sec. 4.1.2. These components guide the generation of masks for positive and negative samples.

**Text Masking**: For the text modality, a positive mask $M_L^+$ is constructed by randomly masking tokens whose similarity scores are *above* the threshold $\tau_L$ (the intersection). A negative mask $M_L^-$ is created by randomly masking tokens with scores *below* $\tau_L$ (the difference).

**Image Masking via Segmentation**: Due to high spatial redundancy in images, simply masking individual patches is often insufficient to meaningfully alter semantics (He et al., 2022). We therefore generate coarse semantic segments by applying hierarchical clustering to the local visual features $\boldsymbol{V}'$, leveraging the observation that features from the same object exhibit high similarity (Oquab et al., 2024) (Details in App. C.2). The relevance of each resulting segment $\boldsymbol{R}_k$ to the text is then scored by its mean similarity: $s_k = \sum_{p \in \boldsymbol{R}_k} \boldsymbol{S}_{L2V}(p)/|\boldsymbol{R}_k|$. Segments with $s_k > \tau_V$ are identified as the intersection set $\{M_V^+\}$, while those with $s_k < \tau_V$ form the difference $\{M_V^-\}$. These segment-based masks are then used to generate positive and negative samples for the image modality. Note that this entire sample generation pipeline, including segmentation and masking, is used exclusively during training and is completely discarded at inference time, resulting in zero computational overhead.

Table 1: Results on *sym-MM2MM* dataset, with the best-performing method marked in **bold** and the runner-up method underlined. Metric Avg. is the average of Precision and mR, which is chosen as the final metric. SUMMR-B+D represents our method with BGE-m3+DINOv2 backbone, and SUMMR-C represents our method with CLIP backbone.

| Category | Method | R@1 | R@5 | R@10 | mR | Prec. | Avg. | #Param.(B) | #Dim. | FPS |
|---|---|---|---|---|---|---|---|---|---|---|
| **Supervised**, **Encoder-based** | CLIP-SF | 55.61 | 94.86 | 97.20 | 82.55 | 73.36 | 77.96 | 0.43 | 768 | 361 |
| | BGE-VL | 29.91 | 57.94 | 61.68 | 49.84 | 64.02 | 56.93 | 0.15 | 512 | 1157 |
| **Supervised**, **VLM-based** | MM-Embed | 55.61 | 94.39 | 96.26 | 82.09 | 75.70 | 78.89 | 7.75 | 4096 | 19 |
| | VLM2Vec | 52.33 | 79.91 | 85.05 | 72.43 | 71.50 | 71.96 | 7.75 | 3584 | 19 |
| | GME | 56.07 | 91.12 | 93.93 | 80.37 | 74.77 | 77.57 | 7.75 | 3584 | 19 |
| | Unite | 52.80 | 91.59 | 95.33 | 79.91 | 75.23 | 77.57 | 7.75 | 3584 | 19 |
| | LamRA | 53.74 | 82.71 | 90.65 | 75.70 | 78.50 | 77.10 | 7.75 | 3584 | 19 |
| | UniME | 72.43 | 96.73 | 97.20 | 88.79 | 73.36 | 81.07 | 7.49 | 3584 | 18 |
| | mmE5 | 76.64 | **99.53** | **99.53** | **91.90** | 76.64 | 84.27 | 10.12 | 4096 | 4 |
| **Unsupervised**, **Encoder-based** | CLIP-SF-ZS | 53.27 | 92.99 | 94.39 | 80.22 | 71.03 | 75.62 | 0.15 | 512 | 1157 |
| | **SUMMR-B+D** | 72.90 | 93.93 | 95.79 | 87.54 | **85.51** | 86.53 | 0.71 | 768 | 520 |
| | **SUMMR-C** | **77.57** | 97.20 | 97.66 | 90.81 | 84.58 | **87.69** | 0.20 | 768 | 922 |

The original (anchor) sample and its constructed positive and negative augmentations are used for contrastive training. The negative set is comprehensive, combining three complementary strategies: 1) constructed negatives from masking the difference set, which teach sensitivity to information deletion; 2) standard in-batch negatives for diversity; and 3) offline-mined hard negatives (details in App. D.4), which introduce semantic conflicts (e.g., content modification or addition) to complement the deletion-based negatives. This entire process trains the model to produce a highly discriminative joint embedding for symmetric MM2MM retrieval.

## 5 EXPERIMENTS

In this section, we benchmark SUMMR against ten leading methods on the *sym-MM2MM* task, conduct extensive ablation studies to validate our two-stage design, and provide qualitative analyses to verify both the model's internal mechanisms and its retrieval performance.

**Experimental Setup**: We train SUMMR on 800,000 unlabeled image-text pairs from LAION-5B and evaluate all methods on our new *sym-MM2MM* benchmark. Performance is measured using Recall@k (k=1, 5, 10) on a large candidate pool, a Precision score on our curated pairs, and the average (Avg.) of mean Recall (mR) and Precision. Our baselines involve a comprehensive suite of recent universal multimodal embedding models, which are claimed to be effective across a wide range of tasks and include both lightweight encoder-based models (e.g., CLIP-SF) and powerful VLM-based systems (e.g., mmE5). Crucially, as publicly available, large-scale training datasets for the symmetric MM2MM task do not exist—a core problem our work aims to solve—these supervised baselines are necessarily fine-tuned on existing asymmetric task datasets. This experimental setup therefore provides a realistic test of their claimed universality and generalization, pitting models trained for general or asymmetric tasks against our framework designed specifically for symmetric retrieval. We evaluate two variants of our model, SUMMR-B+D (BGE-m3+DINOv2) and SUMMR-C (CLIP), to demonstrate backbone compatibility, with a zero-shot CLIP-SF-ZS model providing an unsupervised lower bound. Further details on the baselines and implementation are provided in App. D.

**Main Results**: Tab. 1 presents the quantitative results on our *sym-MM2MM* benchmark. SUMMR demonstrates a clear superiority over all baselines, with our best model, SUMMR-C, achieving an average score of **87.69**. This surpasses the strongest VLM-based method, mmE5, by 3.42 points, being over 50 times smaller in model size (0.20B vs. 10.12B) and producing an embedding vector over 5 times more compact (768 vs. 4096 dimensions). This result highlights a critical limitation of the current paradigm: even massive, SOTA VLMs, when supervised on asymmetric tasks, fail to effectively generalize to the symmetric retrieval paradigm. Our self-supervised framework, designed specifically for this task's structure, proves more effective and vastly more efficient. We also present qualitative results and analysis in App. E.

We observe that mmE5 achieves a notably strong **mR** score, likely due to the fact that it synthesizes training instances where both the query and the target are image-text pairs and partially symmetric - starting with a source image, it finds a visually similar positive image via simple image-to-image retrieval, and then uses a VLM to generate corresponding texts for both. However, this data aug-

mentation process fails to guarantee perfect symmetry. In contrast, our self-supervised approach guarantees strong semantic symmetry by constructing positive samples directly from the anchor's own multimodal context, resulting in a more discriminative feature space that helps distinguish correct positives from challenging hard negatives, explaining SUMMR's superior performance on the more challenging **R@1** and **Precision** metrics.

The lack of generalization from purely asymmetric tasks is also highlighted by the performance of the supervised CLIP-SF model, which shows only a marginal gain over its zero-shot counterpart CLIP-SF-ZS (77.96 vs. 75.62 Avg.). In stark contrast, SUMMR-C improves upon the same zero-shot backbone by over 12 points, proving the usefulness of our task-aligned self-supervision. Finally, the results highlight our framework's ability to effectively learn cross-modal alignment, not merely depend on it. The SUMMR-B+D variant, which starts with powerful but entirely separate unimodal encoders, already achieves a score of 86.53, outperforming the strongest VLM baseline. This result is crucial, as it proves our framework can bridge modality disparities from a "cold start." The superior performance of the CLIP-based SUMMR-C then demonstrates the model's versatility, showing it can also capitalize on a pre-aligned initialization to achieve an additional performance gain.

**Ablation Studies**: To attribute performance gains to specific design choices, we conduct a series of ablation studies on the BGE+DINO variant of SUMMR. App. F presents more details.

We first analyze the impact of each loss term and architectural choice in Stage 1, with results shown in Tab. 2. An important initial observation is that the primary role of Stage 1 is to learn a high-quality disentanglement capability for Stage 2, as optimal intermediate performance after Stage 1 does not always correlate with the best final performance. The necessity of our disentanglement framework is immediately apparent. A baseline trained with only a standard contrastive loss ($\mathcal{L}_{ITC}$ **only**), without our disentangle-

Table 2: Ablation studies for Stage 1.

| Setting | After Stage 1 | | | | After Stage 2 | | | |
|---|---|---|---|---|---|---|---|---|
| | R@1 | mR | Prec. | Avg. | R@1 | mR | Prec. | Avg. |
| $\mathcal{L}_{ITC}$ only | 64.0 | 81.5 | 77.6 | 79.5 | 67.8 | 84.6 | 78.5 | 81.5 |
| w/o $\mathcal{L}_{ITC}$ | 68.7 | 81.6 | 85.1 | 83.3 | 68.7 | 80.2 | 85.1 | 82.6 |
| w/o $\mathcal{L}_{GLA}$ | 66.4 | 80.7 | 80.8 | 80.8 | 68.7 | 80.8 | 80.8 | 80.8 |
| w/o $\mathcal{L}_{GD}$ | 64.5 | 85.2 | 78.0 | 81.6 | 72.0 | 87.1 | 82.7 | 84.9 |
| w/o $\mathcal{L}_{LD}$ | 65.0 | 81.2 | 79.9 | 80.5 | 68.7 | 82.6 | 82.2 | 82.4 |
| w/o $M$ | 70.1 | 83.3 | 84.6 | 84.0 | 72.0 | 88.3 | 83.6 | 86.0 |
| $\hat{M}$ | 66.4 | 84.0 | 81.3 | 82.6 | 71.5 | 88.8 | 80.8 | 84.8 |
| w/o $W$ | 66.8 | 85.5 | 84.1 | 84.8 | 69.6 | 81.2 | 84.1 | 82.6 |
| **SUMMR** | 68.7 | 82.6 | 81.8 | 82.2 | 72.9 | 87.5 | 85.5 | 86.5 |

ment mechanism, suffers a 5.0 point drop in final performance, underscoring the importance of our two-stage design (see full metrics in Tab. 5). This highlights the ineffectiveness of naively aligning all information without separating shared and unique concepts. The synergistic nature of our multi-faceted loss is also clear. Removing any of the four core loss components ($\mathcal{L}_{ITC}$, $\mathcal{L}_{GLA}$, $\mathcal{L}_{GD}$, $\mathcal{L}_{LD}$) degrades the final performance. The removal of local distillation (**w/o $\mathcal{L}_{LD}$**) is particularly detrimental (-4.1 points), as it compromises the local feature quality essential for the alignment signal. The absence of global distillation (**w/o $\mathcal{L}_{GD}$**) also causes a significant drop (-1.6 points), confirming the importance of preserving modality-specific information. The specific design of our masking strategy is also proved vital. Discarding the evolutionary mask schedule (**w/o $M$**) or applying a static hard mask from the outset ($\hat{M}$) hinders performance, demonstrating that the gradual annealing process is critical for training stability. Similarly, removing the projection heads (**w/o $W$**) results in a 3.9 point drop, confirming their role in resolving feature-level mismatches to produce a cleaner alignment signal. Further sensitivity analyses in App. F.2 and F.3 confirm that our model is not overly sensitive to the margin $\delta$ or loss weights, underscoring the robustness of our training framework.

We then proceed to validate the self-supervised sample construction strategy (via masking) in Stage 2. As shown in Tab. 3, the model trained only through Stage 1 achieves a respectable score of 82.2. However, the introduction of our contrastive sample construction pipeline in Stage 2 boosts this to 86.5. This 4.3-point gain (a 5.2% relative improvement) is not only statistically significant but is also larger than the entire 3.42-point performance gap between our SOTA model and the strongest VLM baseline. Notably, the gain is concentrated on the most challenging metrics,

Table 3: Ablation of Stage 2.

| Setting | R@1 | mR | Prec. | Avg. |
|---|---|---|---|---|
| Stage 1 | 68.7 | 82.6 | 81.8 | 82.2 |
| w/o CN | 71.5 | 88.0 | 79.9 | 84.0 |
| RP, w/o CN | 71.5 | 85.7 | 80.8 | 83.3 |
| AP, w/o CN | 65.9 | 87.2 | 77.6 | 82.4 |
| w/o Seg. | 67.8 | 80.7 | 82.2 | 81.5 |
| SAM | 70.6 | 84.0 | 83.4 | 83.7 |
| **SUMMR** | 72.9 | 87.5 | 85.5 | 86.5 |

with R@1 increasing from 68.7 to 72.9 and the Precision score rising from 81.8 to 85.5. The benefit is even more pronounced for our best-performing SUMMR-C model, where, as detailed in App. F.5 (Table 9), Stage 2 delivers a massive 8.5-point improvement. This highlights the purpose of our design: Stage 1 learns to disentangle information, while Stage 2 is a focused, lightweight fine-tuning step (200 steps) that distills this capability into a more specialized and powerful final encoder, free from the auxiliary mask-generation objectives. Removing the constructed negatives (**w/o CN**) leads to a 2.5 point drop, demonstrating their value in teaching the model robustness to information loss. The superiority of our positive sampling strategy is also evident. When we replace our method of masking the semantic intersection with alternatives like random patch/token masking (**RP, w/o CN**) or standard data augmentation (such as image crop, rotation and text rewriting via LLM, denoted as "**AP, w/o CN**"), performance further declines by 0.7 and 1.6 points compared to "w/o CN", respectively. This proves that our unique approach, which constructs positive samples by removing the intersection set of both modalities, is particularly effective. Additionally, as detailed in App. F.4, removing only the offline-mined hard negatives causes a significant 5.45-point performance drop, highlighting their critical role. Finally, the necessity of a segmentation approach to image masking is confirmed by a 5.0 point drop when replacing semantic segmentation with simple patch-level masking (**w/o Seg.**), which is insufficient for altering image semantics meaningfully. Furthermore, as shown in the "SAM" ablation (Tab. 3), our simple clustering-based segmentation outperforms using a SOTA model like SAM, justifying our efficient design choice.

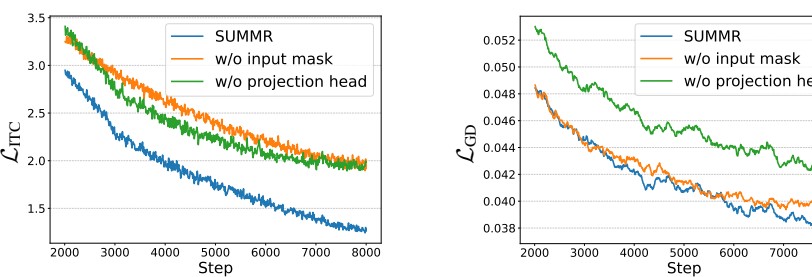

Figure 4: Training loss curves for different settings in Stage 1.

**Visualization**: To qualitatively verify that SUMMR learns the intended disentanglement, we visualize its internal mechanisms. Training loss curves in Fig. 4 confirm our model's stability, demonstrating that the evolutionary mask and projection heads effectively disentangle the alignment and distillation objectives. This learning process is further illustrated in Fig. 12, where similarity score distributions for positive and negative pairs progressively diverge, validating the adaptive QDA thresholding's ability to find a reliable signal separating shared from unique features. The result is evident in the global-to-local similarity heatmaps (Fig. 13 and Fig. 14), which offer qualitative proof of successful disentanglement: the model correctly highlights shared concepts (the "child" region and text) while ignoring modality-specific details like background elements or dates. Furthermore, analysis in App. G.3 shows the model is robust to geometric transformations while remaining sensitive to semantic changes like color, confirming it learns meaningful attributes.

## 6 CONCLUSION

In this paper, we introduce SUMMR, a novel self-supervised framework designed for the challenging and underexplored task of symmetric MM2MM retrieval. By learning to disentangle shared and unique information from unlabeled image-text pairs, SUMMR automatically generates its own training data for contrastive learning. On a new benchmark we develop for this task, SUMMR substantially outperforms ten state-of-the-art models, including a VLM-based system over 50 times its size. Our work demonstrates that task-aligned self-supervision is a more effective and efficient paradigm than costly supervised approaches for symmetric retrieval.

## 7 REPRODUCIBILITY STATEMENT

We are committed to ensuring the reproducibility of our work. To this end, the source code for our SUMMR framework and all experiments will be made publicly available upon publication, as mentioned in the abstract. Our newly introduced *sym-MM2MM* benchmark, including its detailed

construction pipeline, evaluation protocol, and additional examples, is described in Section 3 and App. B. The complete *sym-MM2MM* benchmark is available in the anonymous repository. The core technical details of our method are fully elaborated in the appendices: the derivation of the adaptive QDA threshold ($\tau$) is provided in App. C.1; the iterative hierarchical clustering algorithm for image segmentation is detailed in App. C.2. A comprehensive description of the experimental setup, including training data, model backbones, hyperparameters (e.g., training steps, batch sizes, loss weights), LoRA fine-tuning strategy, masking strategy for constructed positive/negative, hard negative mining strategy, and details on all baseline models, can be found in App. D. Furthermore, App. F presents more results and analysis of ablation studies and more visualization results, which demonstrate the effectiveness and reveal the internal mechanisms of our method further. We believe these resources provide sufficient support for reproducing the main findings of this paper.

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

Table 4: Meanings of Notations

| Notation | Type | Meaning |
|---|---|---|
| $\boldsymbol{X}$ | Input | The input image-text sample |
| $\boldsymbol{I}$ | Input | The input image in a image-text sample |
| $\boldsymbol{T}$ | Input | The input text in a image-text sample |
| $\mathcal{E}$ | Module | The complete joint representation encoder |
| $\mathcal{E}_V$ | Module | The visual encoder |
| $\mathcal{E}_L$ | Module | The language encoder |
| $\mathcal{A}_V$ | Module | The MLP adapter for vision |
| $\mathcal{A}_L$ | Module | The MLP adapter for language |
| $\mathcal{E}_{VL}$ | Module | The Vision-Language encoder |
| $\boldsymbol{W}_V$ | $d \times d$ | The projection head for vision |
| $\boldsymbol{W}_L$ | $d \times d$ | The projection head for language |
| $V$ | $P \times d$ | The patch-level visual features |
| $V'$ | $P \times d$ | The patch-level visual features excluding the original [CLS] tokens |
| $L$ | $T \times d$ | The token-level textual features |
| $L'$ | $T \times d$ | The token-level textual features excluding the original [CLS] tokens |
| $\boldsymbol{M}_V$ | $P$ | The input mask for vision |
| $\boldsymbol{M}_L$ | $T$ | The input mask for language |
| $\hat{\boldsymbol{M}}$ | $P$ or $L$ | The hard input mask of image or text |
| $\boldsymbol{R}_k$ | $T$ | The $k$-th segmentation generated by image segmentation generator |
| $\boldsymbol{f}_V$ | $d$ | The refined, cross-modal global visual embedding |
| $\boldsymbol{f}_L$ | $d$ | The refined, cross-modal global language embedding |
| $\boldsymbol{f}$ | $d$ | The joint representation of an image-text sample |
| $\boldsymbol{S}_{L2V}$ | $P$ | The global-text-to-local-image similarity |
| $\boldsymbol{S}_{V2L}$ | $T$ | The global-image-to-local-text similarity |
| $\mathbb{S}_{L2V}^{+}$ | Set | The positive set of global-text-to-local-image similarity |
| $\mathbb{S}_{V2L}^{-}$ | Set | The negative set of global-image-to-local-text similarity |
| $\mathcal{L}_{\text{ITC}}$ | Loss | The Image-Text Contrastive loss in Stage 1 |
| $\mathcal{L}_{\text{GLA}}$ | Loss | The Global-to-Local Alignment loss in Stage 1 |
| $\mathcal{L}_{\text{GD}}$ | Loss | The Global Distillation loss in Stage 1 |
| $\mathcal{L}_{\text{LD}}$ | Loss | The Local Unimodal Distillation loss in Stage 1 |
| $\delta$ | Constant | The margin of Global-to-Local Alignment |
| $\tau$ | Variable | The adaptive threshold for intersection mask |
| $\eta$ | Variable | The temperature in contrastive loss |
| $\rho$ | Variable | The schedule for evolutionary masking |
| $\lambda_i$ | Constant | The loss weights |
| $B$ | Constant | The batch size |
| $\boldsymbol{M}_V^{+}$ | $P$ | The positive mask for image in Stage 2 |
| $s^k$ | Variable | The average global-text-to-local-image similarity for $k$-th image part |

## A  A DETAILED DISCUSSION ON RELATED WORKS

In this section, we first provide a taxonomy of multimodal retrieval tasks, and then review methods for learning joint multimodal representations.

### A.1  MULTIMODAL RETRIEVAL

Based on the modality of the query and retrieved content, multimodal retrieval can be broadly categorized into three groups:

**Unimodal-to-Multimodal (UM2MM)**: These systems use a single-modality query (e.g., text) to retrieve multimodal results (e.g., images with accompanying text). Examples include EDIS (Liu et al., 2023), which retrieves news headline images based on text queries, and WebQA (Chang et al., 2022), which addresses open-domain question answering by retrieving answers that may be textual or visual.

**Multimodal-to-Unimodal (MM2UM)**: These systems use a multimodal query to retrieve results in a single modality. This includes OVEN (Hu et al., 2023) and InfoSeek (Chen et al., 2023) in the context of open-domain visual question answering. Additionally, examples focused on composed image retrieval, such as FashionIQ (Wu et al., 2021), CIRR (Liu et al., 2021), and MagicLens (Zhang et al., 2024), retrieve a target image based on a reference image and a text description that modifies or refines the image.

**Multimodal-to-Multimodal (MM2MM)**: These systems use a multimodal query to retrieve results that are also multimodal. Some configurations of OVEN and InfoSeek fall into this category.

UniIR and VLM2Vec further summarize these datasets from different perspectives and have developed the M-BEIR (Multimodal Benchmark for Information Retrieval) and MMEB (Massive Multimodal Embedding Benchmark).

Notably, within the last MM2MM category, existing approaches like OVEN and InfoSeek primarily address asymmetric search tasks, where the query and result have distinct roles. However, as discussed in Sec. 1, many real-world scenarios demand symmetric MM2MM retrieval, where the query and retrieved content are semantically interchangeable. To address this gap, we introduce a novel benchmark dataset, *sym-MM2MM*, specifically designed for symmetric MM2MM retrieval and present SUMMR, our approach tailored to this problem.

## A.2 MULTIMODAL JOINT EMBEDDING METHODS

Existing multimodal joint embedding methods can be classified into two main groups based on their architectural approach:

**Score Fusion**: These methods, such as BGE-VL (Zhou et al., 2025) and CLIP-SF (as employ in UniIR (Wei et al., 2024)), typically leverage cross-modal bi-encoders (e.g., CLIP) as a foundation. They independently extract image and text embeddings and then compute a multimodal joint embedding as a weighted average of the unimodal embeddings. The resulting similarity score is then calculated as a weighted average of the cosine similarities between all possible pairings of the image and text embeddings from the query and candidate items. However, relying solely on a weighted average may not effectively capture the complex interplay and complementary information between modalities.

**Feature Fusion**: These methods aim to more directly integrate information from different modalities. UniIR (Wei et al., 2024) and MagicLens (Zhang et al., 2024) employ a late fusion approach that integrates features extracted by CLIP using an additional vision-language (VL) encoder. More recent techniques, such as MM-Embed (Lin et al., 2025) and VLM2Vec (Jiang et al., 2025), utilize multimodal large language models (VLMs) to fuse features, often using the hidden state of the [END] token as the multimodal joint embedding. While promising, the high computational complexity of VLMs makes them less practical for real-world retrieval systems that require rapid throughput; for example, Google processes approximately 6.3 million search queries per minute[1], necessitating embedding models with fast inference speeds. Additionally, since VLMs are primarily trained for generation tasks, their features may not be optimally suited for discriminative embedding tasks and often require extensive supervised fine-tuning (SFT) to achieve competitive performance. Indeed, all of the aforementioned methods rely on supervised learning. However, obtaining high-quality labeled data is challenging, and recent studies (Chu et al., 2025) suggest that SFT can lead to memorization and poor generalization to unseen queries, especially in asymmetric search contexts like open question answering.

In contrast to these approaches, our work focuses on the symmetric MM2MM search paradigm, allowing us to adopt a self-supervised learning strategy based on readily available image-text pairs from datasets like LAION-5B. This eliminates the need for expensive manual annotation. Furthermore, we employ a computationally efficient late fusion architecture that leverages a VL-encoder to integrate features extracted separately by text and image encoders, resulting in a small and practical model suitable for real-world deployment. By focusing on symmetric search and self-supervision, we aim to overcome the limitations of existing multimodal retrieval methods.

---

[1] https://clictadigital.com/how-many-google-searches-per-day-are-there/

| $X$ | $X^+$ | $X'$ | $X^-$ |
|---|---|---|---|

Obama visits Berlin, Germany.

The american president waved to the crowd for greeting in Berlin.

Obama's speech in Berlin, Germany.

Obama's speech in Berlin, Germany, calls for strategic nuclear weapons reduction.

In Chiang Mai, even during traffic jams, you won't hear the sound of car horns.

The streets of Chiang Mai are quiet, even though there are many cars

Chiang Mai's congested streets are deafening with the sound of car horns due to traffic jams.

Boy's white T-shirt.

Boy's white T-shirt.

Boy's white T-shirt.

In the kitchen, on the left side, there is a cabinet full of life, adorned with some decorative small items. Various cooking utensils hang on the wall, next to a curtain with yellow patterns.

In the kitchen, there is a cabinet on the left side. Some cooking utensils and yellow patterned curtains are hanging on the wall. A person is cooking.

In the kitchen, there is a cabinet on the left side. Some cookware and yellow patterned curtains hang on the wall.

The picture shows a spotted kitten walking on an indoor wooden floor, with a white wall in the background.

The picture shows a spotted kitten walking on a wooden floor indoors. In the background, a bicycle is parked against the wall next to the kitten.

The picture shows a spotted kitten walking on a wooden floor indoors, with a white wall in the background.

Figure 5: More examples of our *sym-MM2MM* dataset.

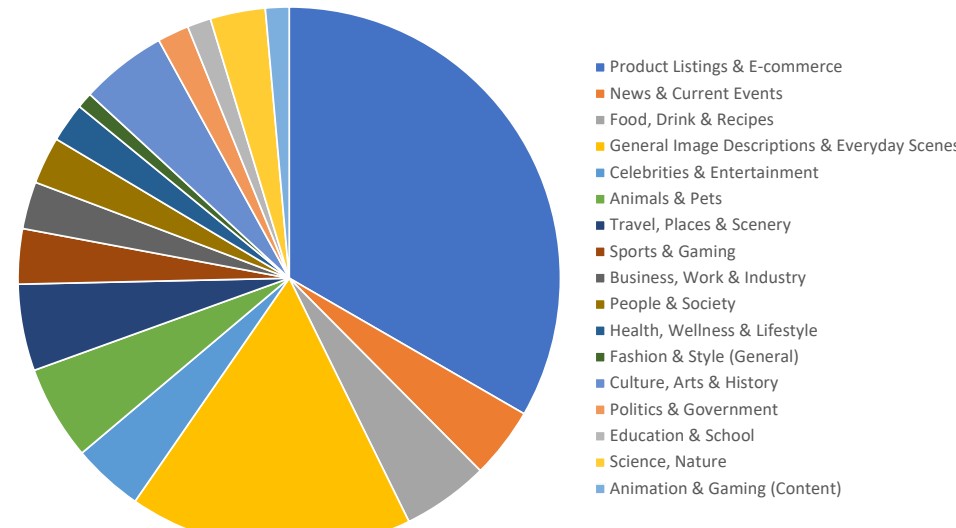

Figure 6: The distribution of categories for *sym-MM2MM*.

## B *sym-MM2MM* DATASET

We show the distribution of categories for *sym-MM2MM* in Fig. 6, it consists of 17 categories and these categories cover a broad spectrum of topics, indicating the diversity and generalization of *sym-MM2MM*. Fig. 5 presents additional examples from our *sym-MM2MM* dataset. In each row, $X$ is the original sample, $X^+$ and $X^-$ are the positive and negative sample corresponding to $X$, respectively. Note that to construct more challenging test cases, we occasionally introduce a variant $X'$ for some $X$, which is created by slightly modifying the text of the original sample. For these instances, the evaluation criterion for precision is adjusted: a model is considered correct only if the similarity of the positive pair is greater than the similarity of this hard-negative pair, i.e., $\langle f^+, f \rangle > \langle f^-, f' \rangle$, in which $f'$ is the joint representation of $X'$. This modification significantly increases the difficulty of the benchmark, requiring the model to discern more subtle differences. Notably, the images in the fourth and fifth examples are generated using our proposed pipeline.

The rationale for designating positive and negative samples in each example is as follows:

1. **Specific Detail:** The text of $X^-$ contains an additional detail (the "theme of the speech") not present in $X$, $X^+$ and $X'$.

2. **Quiet Street vs. Car Horns:** Both $X$ and $X^+$ indicate a quiet street, whereas $X^-$'s text mentions "car horns."

3. **Plain vs. Lettered T-shirt:** The images in $X$ and $X^+$ show a plain white T-shirt back, while the image in $X^-$ has a letter "S" on it.

4. **Cross-Modal "Cooking":** The concept of "cooking" is shared between the image of $X$ and the text of $X^+$, but is absent in $X^-$.

5. **Cross-Modal "Bicycle":** The concept of a "bicycle" connects the image of $X$ and the text of $X^+$, but is missing in $X^-$.

For more instances, please visit the anonymous repository of *sym-MM2MM*.

## C METHOD DETAILS

### C.1 DERIVATION OF THE THRESHOLD $\tau$

GLA yields two distributions of similarity scores—one for positives, one for negatives. We model the positive similarities as $\mathcal{N}(\mu^+, (\sigma^+)^2)$, negatives as $\mathcal{N}(\mu^-, (\sigma^-)^2)$. The optimal threshold $\tau$

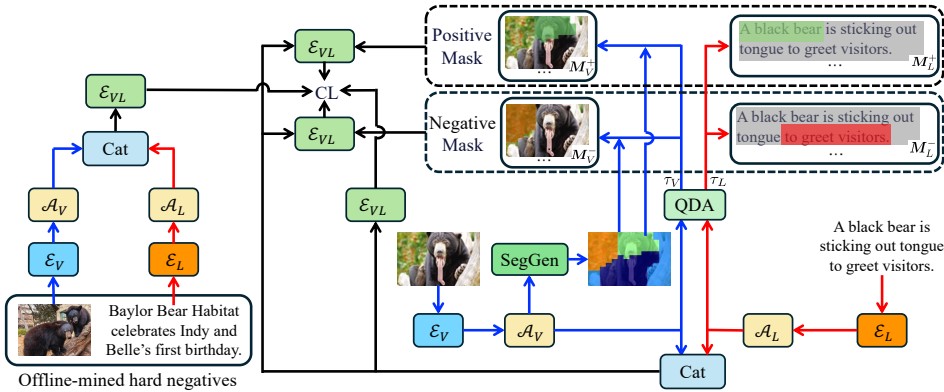

Figure 7: Architectural overview of Stage 2: Self-supervised Representation Learning. The SegGen module segments the image, and the QDA module provides thresholds to discriminate between the intersection and difference sets. Positive samples are created by masking the intersection set, and negative samples by masking the difference set. A contrastive loss (CL) is then computed using original sample, constructed samples, in-batch negatives, and offline-mined hard negatives.

separating intersection from difference is then obtained by finding where these Gaussians intersect, via one-dimensional Quadratic Discriminant Analysis (QDA):

$$\mathcal{N}(\tau; \mu^+, (\sigma^+)^2) = \mathcal{N}(\tau; \mu^-, (\sigma^-)^2), \tag{5}$$

This has a closed-form solution:

$$\tau = \frac{b \pm \sqrt{b^2 - 4ac}}{2a}, \tag{6}$$

$$a = (\sigma^+)^2 - (\sigma^-)^2, \tag{7}$$

$$b = 2(\mu^+ \sigma^{-2} - \mu^- \sigma^{+2}), \tag{8}$$

$$c = (\sigma^+ \mu^-)^2 - (\sigma^- \mu^+)^2 + 2(\sigma^+ \sigma^-)^2 \ln \frac{\sigma^-}{\sigma^+}. \tag{9}$$

The relevant $\tau$ lies between $\mu^+$ and $\mu^-$. In the case $\sigma^+ = \sigma^-$, we have $\tau = (\mu^+ + \mu^-)/2$. QDA's decision boundary is theoretically the Bayes optimal classifier if class-conditional distributions are Gaussians; It minimizes the sum of type-I (false positive) and type-II (false negative) errors under these conditions.

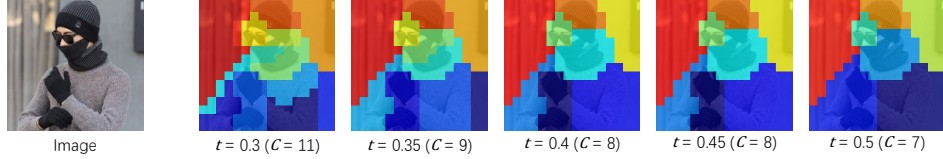

Figure 8: Impact of the distance threshold $t$ on hierarchical clustering for image segmentation, $C$ is the number of resulting clusters

## C.2 ITERATIVE HIERARCHICAL CLUSTERING FOR IMAGE SEGMENTATION GENERATION

For image masking, we first partition the image into semantically coherent segments using hierarchical clustering on the patch embeddings. We adopt the mean linkage criterion, where the decision to merge two clusters is based on the average distance between all pairs of elements, one from each cluster. The clustering process is governed by a distance threshold $t$.

The choice of $t$ is critical to the quality of the segmentation, as illustrated in Fig. 8. A small $t$ leads to over-segmentation, where a single semantic region is fragmented into multiple spurious parts. Conversely, an excessively large $t$ results in under-segmentation, incorrectly merging distinct objects. As the optimal threshold varies across different images, we propose a simple yet effective iterative

hierarchical clustering with an adaptive thresholding method, detailed in Algorithm 1. The initial threshold is set to $t = 0.45$. It is iteratively adjusted based on two conditions: if any resulting segment is excessively large (e.g., containing over 87% of total patches), $t$ is decreased; if the number of segments is too high (e.g., greater than 5), $t$ is increased. The process terminates when neither of these conditions is met or after a maximum of 5 iterations.

---

**Algorithm 1** Iterative Hierarchical Clustering

---

**Require:** Image features $V'$ with number of patches $P$, initial threshold $t_{init} = 0.45$, step $\Delta_t = 0.05$, max iterations $N_{iter} = 5$, max segment size ratio $r_{max} = 0.87$, max segments $K_{max} = 5$.
1: $t \leftarrow t_{init}$
2: **for** $i = 1$ to $N_{iter}$ **do**
3:     Perform hierarchical clustering on $V'$ with threshold $t$ to obtain segments $\{\boldsymbol{R}\}$.
4:     Let $K = |\{\boldsymbol{R}\}|$ be the number of segments.
5:     Let $r_{largest} = \max_{\boldsymbol{R}_k \in \{\boldsymbol{R}\}} |\boldsymbol{R}_k|/P$.
6:     **if** $r_{largest} > r_{max}$ **then**
7:         $t \leftarrow t - \Delta_t$                    ▷ Decrease threshold to prevent overly large segments
8:     **else if** $K > K_{max}$ **then**
9:         $t \leftarrow t + \Delta_t$                      ▷ Increase threshold to merge small segments
10:     **else**
11:         **break**                                      ▷ Conditions met, terminate.
12:     **end if**
13: **end for**
14: **return** Segments $\{\boldsymbol{R}\}$.

---

## C.3   TRAINING LOSS OF STAGE 2

The contrastive loss in Stage 2 is calculated as following:

$$\mathcal{L} = \frac{1}{N} \sum_i \log \left( \frac{\sum_{j \in \mathbb{D}^{+i}} \exp(\langle \boldsymbol{f}^i, \boldsymbol{f}^j \rangle / \eta)}{\sum_{k \in \mathbb{D}^{+i} \cup \mathbb{D}^{-i}} \exp \langle \boldsymbol{f}^i, \boldsymbol{f}^k \rangle / \eta} \right) , \tag{10}$$

in which $\mathbb{D}^{+i}$ and $\mathbb{D}^{-i}$ are the positive set and negative set of $\boldsymbol{X}^i$, respectively. The positive set is composed of the constructed positive. While negative set is composed of constructed negative samples, offline-mined hard negatives and in-batch negatives.

## C.4   DISCUSSION ON THE MASKING AND DISENTANGLEMENT MECHANISM

Our masking and disentanglement mechanism was designed to be both effective and robust. Here, we address common questions regarding its technical implementation.

**On Differentiability and Learning with QDA**: A key design choice is the use of QDA to derive the adaptive threshold $\tau$, which is a non-differentiable step. However, the overall learning process remains end-to-end in practice. The training objective drives the fully differentiable encoders ($\mathcal{E}_V$, $\mathcal{E}_L$) to produce feature embeddings that are increasingly separable by the QDA classifier. In this way, the model learns through the non-differentiable boundary by continuously improving the features fed into it. This design pattern provides a principled, statistical approach to separating signal from noise without requiring a fully differentiable path.

**On Robustness to Domain Shift and Inference Overhead**: The adaptive threshold $\tau$ is inherently robust to domain shifts because it is not a fixed value. It is recalculated on-the-fly for each training batch using QDA, allowing it to adjust to the statistics of new data. Most importantly, the entire thresholding and masking mechanism—including QDA and segmentation—is part of the training-time only self-supervised data generation pipeline. At inference time, these components are discarded. The model performs a single, efficient forward pass to generate an embedding, with zero overhead from any clustering, masking, or thresholding computations.

## D    EXPERIMENTAL SETUP

### D.1    DATASET AND EVALUATION METRICS

We train SUMMR on a subset of 800,000 image-text pairs randomly sampled from the LAION-5B dataset. We then evaluate all methods on our newly introduced *sym-MM2MM* benchmark, with a retrieval candidate pool comprising 1 million LAION pairs plus our benchmark pairs. As detailed in Section 3, we report Recall@k (k=1, 5, 10) and its mean (mR) to assess performance in a large-scale setting, alongside a fine-grained Precision score to measure discriminative power on our curated positive/hard-negative pairs. The final metric, Avg., is the average of mR and Precision.

### D.2    BASELINES

Our baselines include a comprehensive suite of recent universal multimodal embedding models. It is crucial to note that these supervised methods are typically fine-tuned on existing large-scale datasets such as M-BEIR and MMEB. The reliance on these datasets is a necessity, not a choice, as publicly available, large-scale datasets for symmetric MM2MM training do not exist—a data scarcity problem our work directly addresses. Consequently, this experimental setup provides a realistic test of generalization: it pits models trained on asymmetric tasks against SUMMR, a framework specifically designed for symmetric retrieval from the ground up. The baselines fall into two categories: lightweight **Encoder-based Supervised** methods such as **CLIP-SF** (Wei et al., 2024) and **BGE-VL** (Zhou et al., 2025), and powerful but computationally intensive **VLM-based Supervised** models including **MM-Embed** (Lin et al., 2025), **VLM2Vec** (Jiang et al., 2025), **GME** (Zhang et al., 2025), **Unite** (Kong et al., 2025), **LamRA** (Liu et al., 2025), **UniME** (Gu et al., 2025), and **mmE5** (Chen et al., 2025). To establish a strong **unsupervised** lower bound, we also include a zero-shot version of **CLIP-SF-ZS**, which uses the original pre-trained CLIP model without any task-specific fine-tuning. Further details are provided below:

- **CLIP-SF** (Wei et al., 2024) is an encoder-based model that utilizes CLIP (ViT-Large) as its backbone. It generates a joint representation by summing the individual text and image embeddings. The model is trained on the M-BEIR benchmark, achieving an average score of 48.9. The authors also explore alternative fusion strategies (e.g., BLIP or CLIP feature-level fusion) but found simple summation to be the most effective.

- **CLIP-SF-ZS** serves as our zero-shot encoder-based baseline. It employs the same summation strategy as CLIP-SF but uses a pre-trained, un-fine-tuned CLIP (ViT-Base) model to generate embeddings.

- **BGE-VL** (Zhou et al., 2025) employs a similar architecture to CLIP-SF-ZS. It is trained on a subset of the M-BEIR benchmark, augmented with a large volume of synthetic data.

- **MM-Embed** (Lin et al., 2025) is a VLM-based model that employs Qwen2-VL-7B as its backbone. It uses the hidden state (last-layer) of the final token as the multimodal joint embedding. The model undergoes a two-stage fine-tuning process: first on M-BEIR for universal multimodal retrieval, and then on text-to-text retrieval tasks. During fine-tuning, only the vision-language projector and LoRA modules are updated. It achieves an average score of 52.7 on M-BEIR.

- **VLM2Vec** (Jiang et al., 2025) also extracts embeddings from a VLM (Qwen2-VL-7B) using the final token's hidden state. It is fine-tuned on the MMEB benchmark, which, in addition to retrieval, includes tasks such as Classification, VQA, and Visual Grounding. We compare against the version with the Qwen2-VL-7B backbone, which secures the best retrieval performance (69.9) on MMEB.

- **GME** (Zhang et al., 2025) follows the same embedding extraction strategy and uses the same Qwen2-VL-7B backbone as MM-Embed but is trained on a more extensive dataset and synthetic data to enhance its generalization capabilities.

- **Unite** (Kong et al., 2025), also based on Qwen2-VL-7B and trained on MMEB, introduces a *Modal-Aware Masked Contrastive Learning* objective. This technique enables focused contrastive learning by restricting comparisons to samples sharing the same target modality, thereby reducing inter-modal interference. It achieves SOTA performance on the MMEB retrieval tasks with an average score of 71.6.

- **LamRA** (Liu et al., 2025) utilizes the Qwen2-VL-7B backbone and is fine-tuned on the M-BEIR benchmark, achieving an average score of 56.6 on M-BEIR. The authors also propose a VLM-based reranker, which further boosts the performance to 63.7.

- **UniME** (Gu et al., 2025) extracts embeddings from a LLaVA-OneVision-7B backbone. A key feature is a preliminary *Textual Discriminative Knowledge Distillation* step, where the VLM's language module is trained using a specialized language embedding model (NV-Embed V2) as a teacher, prior to fine-tuning on MMEB. It achieves an average score of 70.6 on the MMEB retrieval tasks.

- **mmE5** (Chen et al., 2025) leverages a Llama-3.2-11B-Vision backbone and is trained on both MMEB and a synthetically generated dataset, achieving an average score of 70.9 on MMEB. The data synthesis process is particularly relevant to our work: it samples a source image, retrieves similar images from LAION using CLIP features to form positive and hard-negative pairs, and then uses a VLM to generate captions. This process creates partially symmetric positive pairs and is highly relevant to our *sym-MM2MM* task, as the source images are also drawn from the LAION corpus.

For all VLM-based methods, we employ an adaptation strategy to make them suitable for the symmetric MM2MM retrieval task. As these models are instruction-tuned, we apply a consistent, task-appropriate prompt to both the query and candidate pairs during inference: 'Represent the given image with related text information:'. This instruction encourages the models to generate a holistic joint embedding rather than focusing on asymmetric question-answering or captioning. This adaptation is crucial for fair evaluation; for instance, it improves the average performance of VLM2Vec from 67.76 to 71.96 on our benchmark. The results reported for all VLM baselines in Table 1 reflect this improved performance.

### D.3 IMPLEMENTATION DETAILS

To show that SUMMR is compatible with different backbones, we evaluate two variants of our model: **SUMMR-B+D**, which uses separate unimodal encoders (BGE-m3 for text, DINOv2 for vision), and **SUMMR-C**, which employs the aligned vision and text encoders in CLIP as the backbone. The default learning rate is $1e-5$. For the distillation in Stage 1, the teacher models are DINOv2 for vision, and a combination of BGE-m3 (for global features) and BGE-m3-retromae (for local features) for text, selected for their respective strengths. In Stage 1, the batch size is 4096 and all loss weights ($\lambda_1$, $\lambda_2$, $\lambda_3$ in Eq. (4)) are set to 1. For SUMMR-B+D, we train 8k steps. For SUMMR-C, we train for only 2k steps, since CLIP already possesses a coarse-grained cross-modal alignment from its extensive pre-training. $\mathcal{E}_{VL}$, $\mathcal{A}_V$, $\mathcal{A}_L$, $\mathcal{W}_V$ and $\mathcal{W}_L$ are fully trained while $\mathcal{E}_V$ and $\mathcal{E}_L$ are fine-tuned using LORA (rank=16 and alpha=32). In Stage 2, training proceeds for 200 steps with a batch size of 512. $\mathcal{A}_V$, $\mathcal{A}_L$ are fully trained while $\mathcal{E}_{VL}$, $\mathcal{E}_V$ and $\mathcal{E}_L$ are fine-tuned via LORA (rank=16 and alpha=32). The loss of each p sample is calculated with one constructed positive (by masking the intersection in either the image or the text), three constructed negatives (two by masking the difference in image or text, and one by masking the intersection in both), two offline mined hard negatives and in-batch negatives. Below, we illustrate the details of image/text masking.

**Text masking**: Departing from the conventional approach of using a fixed masking probability in language model pre-training, our method employs a dynamic probability that spans the full range of 0% to 100%. This strategy is designed to simulate various degrees of information loss, thereby significantly enhancing the diversity of the constructed training samples and improving the model's robustness to incomplete textual information.

**Image masking**: To enrich our training data, we employ a **multi-segment masking** technique. After generating initial segments, instead of masking a single segment, we stochastically select and mask multiple segments simultaneously. This selection is constrained: all chosen segments must reside entirely within either the shared area (to construct positive) or a unique area (to construct negative). This approach generates more complex and diverse training exemplars compared to single-segment masking.

Table 5: Complete metrics for ablation studies of Stage 1.

| Setting | After Stage 1 | | | | | | After Stage 2 | | | | | |
|---|---|---|---|---|---|---|---|---|---|---|---|---|
| | R@1 | R@5 | R@10 | mR | Prec. | Avg. | R@1 | R@5 | R@10 | mR | Prec. | Avg. |
| $\mathcal{L}_{\text{ITC}}$ only | 64.0 | 89.7 | 90.7 | 81.5 | 77.6 | 79.5 | 67.8 | 92.5 | 93.5 | 84.6 | 78.5 | 81.5 (-5.0) |
| w/o $\mathcal{L}_{\text{ITC}}$ | 68.7 | 86.9 | 89.3 | 81.6 | 85.1 | 83.3 | 68.7 | 85.5 | 86.5 | 80.2 | 85.1 | 82.6 (-3.9) |
| w/o $\mathcal{L}_{\text{GLA}}$ | 66.4 | 87.4 | 88.3 | 80.7 | 80.8 | 80.8 | 68.7 | 86.0 | 87.9 | 80.8 | 80.8 | 80.8 (-5.7) |
| w/o $\mathcal{L}_{\text{GD}}$ | 64.5 | 94.4 | 96.7 | 85.2 | 78.0 | 81.6 | 72.0 | 94.4 | 94.9 | 87.1 | 82.7 | 84.9 (-1.6) |
| w/o $\mathcal{L}_{\text{LD}}$ | 65.0 | 88.3 | 90.2 | 81.2 | 79.9 | 80.5 | 68.7 | 88.8 | 90.2 | 82.6 | 82.2 | 82.4 (-4.1) |
| w/o $M$ | 70.1 | 88.3 | 91.6 | 83.3 | 84.6 | 84.0 | 72.0 | 95.8 | 97.2 | 88.3 | 83.6 | 86.0 (-0.5) |
| $\hat{M}$ | 66.4 | 91.6 | 93.9 | 84.0 | 81.3 | 82.6 | 71.5 | 97.2 | 97.7 | 88.8 | 80.8 | 84.8 (-1.7) |
| w/o $W$ | 66.8 | 93.9 | 95.8 | 85.5 | 84.1 | 84.8 | 69.6 | 86.5 | 87.4 | 81.2 | 84.1 | 82.6 (-3.9) |
| **SUMMR** | 68.7 | 89.3 | 89.7 | 82.6 | 81.8 | 82.2 | 72.9 | 93.9 | 95.8 | 87.5 | 85.5 | 86.5 |

## D.4 HARD NEGATIVE MINING FOR STAGE 2

To ensure the model learns fine-grained distinctions, we employ a comprehensive hard negative mining strategy. The process is as follows:

1. **Corpus Construction:** We first assemble a large-scale candidate corpus comprising 4 million image-text pairs from the LAION dataset.

2. **Multi-Modal Feature Extraction:** We utilize a suite of powerful pre-trained models to extract features for all samples in the corpus. The models include: a text embedding model (BGE-m3), a visual feature extractor (DINOv2), a cross-modal model (CLIP), and our own model from Stage 1.

3. **Candidate Retrieval:** For each training sample (anchor), we retrieve hard negatives based on feature similarity.

   - For BGE-m3, DINOv2, and our Stage 1 model, we identify the top-10 samples with the highest embedding similarity to the anchor as hard-negative candidates.

   - For CLIP, we leverage its multimodal alignment capabilities by computing four distinct similarity scores: text-to-image, image-to-text, text-to-text, and image-to-image. For each of these four metrics, we retrieve the top-10 most similar samples.

4. **Final Set Aggregation:** The final hard-negative set for each anchor is formed by the union of all candidates retrieved from the different models and similarity metrics.

During training, for each anchor sample, we randomly sample two hard negatives from its aggregated set to be used in the contrastive loss computation. This approach provides a diverse and challenging set of negatives, compelling the model to learn a more discriminative and robust embedding space.

## E CASE STUDY

To qualitatively evaluate the performance of each method, we present the top-5 retrieval results for sample queries from *sym-MM2MM* in Fig. 9. As illustrated, SUMMR successfully ranks the positive sample first, demonstrating its ability to distinguish fine-grained textual semantics (e.g., "car horn"). We attribute this success to our global text distillation. In contrast, we observe that baselines like CLIP-SF and MM-Embed tend to over-focus on proper nouns (e.g., "Chiang Mai"). Consequently, some of their retrieved images match the location but are irrelevant to the query's primary subject, "street" (see ranks 4–5 for CLIP-SF and 3–4 for MM-Embed). Finally, the results from mmE5 show that some retrieved samples (ranks 4–5) are only tangentially related to the query, which may indicate a less continuous feature space. We hypothesize that this is due to the model overfitting to the synthesis training data from LAION dataset.

**Query**

In Chiang Mai, even during traffic jams, you won't hear the sound of car horns.

| Methods | Rank 1 | Rank 2 | Rank 3 | Rank 4 | Rank 5 |
|---|---|---|---|---|---|
| SUMMR-B+D | ✓ The streets of Chiang Mai are quiet, even though there are many cars. | ✗ Chiang Mai's congested streets are deafening with the sound of car horns due to traffic jams. | ✗ Life in North America: How Chinese Americans won the right to go to school in San Francisco more than 130 years ago. | ✗ Black smoke is rushing! There is a fire in a house on Guoan Street, Tainan, and emergency irrigation is underway | ✗ tricycle |
| SUMMR-C | ✓ The streets of Chiang Mai are quiet, even though there are many cars. | ✗ Chiang Mai's congested streets are deafening with the sound of car horns due to traffic jams. | ✗ What to do in San Francisco in 2020? Fisherman's Wharf, riding the electric rail bus system, and renting bicycles in the city for instruction @Gina Lin | ✗ Walking mariachi bass | ✗ tricycle |
| CLIP-SF | ✗ Chiang Mai's congested streets are deafening with the sound of car horns due to traffic jams. | ✓ The streets of Chiang Mai are quiet, even though there are many cars. | ✗ Chiang Mai - Thailand Royalty Free Stock Photos | ✗ Chiang Mai | ✗ Chiang Mai |
| MM-Embed | ✗ Chiang Mai's congested streets are deafening with the sound of car horns due to traffic jams. | ✓ The streets of Chiang Mai are quiet, even though there are many cars. | ✗ Chiang Mai | ✗ Chiang Mai | ✗ Chiang Mai |
| mmE5 | ✗ Chiang Mai's congested streets are deafening with the sound of car horns due to traffic jams. | ✓ The streets of Chiang Mai are quiet, even though there are many cars. | ✗ Museum Island Pass in Berlin, Germany (including 72 hours of free transportation in Berlin and Berlin Welcome Card)-Berlin Attraction Tickets-Hopetrip Travel Net | ✗ Imported Mercedes-Benz SLS-class AMG driver's headrest | ✗ Xianju. Jingxingyan travels to feel the paradise |

Figure 9: Top-5 retrieval results for a query from the *sym-MM2MM* benchmark. The positive sample is marked with ✓ and the negative with ✗.

**Query**

 The bananas are vividly colored, numerous, and appear very fresh.

| Methods | Rank 1 | Rank 2 | Rank 3 | Rank 4 | Rank 5 |
|---|---|---|---|---|---|
| SUMMR-B+D | ✓ A person is carrying a large square basket of green bananas, set against the backdrop of a bustling market environment where other people are active. | ✗ A person is carrying a large square basket of green bananas on his shoulder. The bananas are brightly colored, numerous, and look very fresh. | ✗ Beautiful woman holding banana bunch | ✗ On the farm, people carry buckets full of green produce | ✗ Millet Banana Net Red Banana |
| SUMMR-C | ✓ A person is carrying a large square basket of green bananas, set against the backdrop of a bustling market environment where other people are active. | ✗ A person is carrying a large square basket of green bananas on his shoulder. The bananas are brightly colored, numerous, and look very fresh. | ✗ Human World Vegetable Vendor: It took 10 years without a break to earn a family's life Author: Source: NetEase Human World | ✗ On July 11, at the cherry tomato planting base in Minzhu Village, Zhongjian Township, Qianxi County, Bijie City, Guizhou Province... | ✗ Woman buying fruits and vegetables at farmers outdoor market Young woman shopping portrait for healthy lifestyle |
| CLIP-SF | ✗ A person is carrying a large square basket of green bananas on his shoulder. The bananas are brightly colored, numerous, and look very fresh. | ✓ A person is carrying a large square basket of green bananas, set against the backdrop of a bustling market environment where other people are active. | ✗ Beautiful woman holding banana bunch | ✗ On the farm, people carry buckets full of green produce | ✗ Millet Banana Net Red Banana |
| MM-Embed | ✗ A person is carrying a large square basket of green bananas on his shoulder. The bananas are brightly colored, numerous, and look very fresh. | ✓ A person is carrying a large square basket of green bananas, set against the backdrop of a bustling market environment where other people are active. | ✗ Harvested bananas for caring Royalty Free Stock Photo | ✗ On the farm, people carry buckets full of green produce | ✗ Banana |
| mmE5 | ✗ A person is carrying a large square basket of green bananas on his shoulder. The bananas are brightly colored, numerous, and look very fresh. | ✓ A person is carrying a large square basket of green bananas, set against the backdrop of a bustling market environment where other people are active. | ✗ Millet Banana Net Red Banana | ✗ "Philippine Banana Going North Remember" | ✗ Banana |

Figure 10: Top-5 retrieval results for a query from the *sym-MM2MM* benchmark. The positive sample is marked with ✓ and the negative with ✗.

Table 6: Complete metrics for ablation studies of Stage 2.

| Setting | R@1 | R@5 | R@10 | mR | Prec. | Avg. |
|---|---|---|---|---|---|---|
| Stage 1 | 68.7 | 89.3 | 89.7 | 82.6 | 81.8 | 82.2 (-4.3) |
| w/o CN | 71.5 | 95.8 | 96.7 | 88.0 | 79.9 | 84.0 (-2.5) |
| RP, w/o CN | 71.5 | 92.1 | 93.5 | 85.7 | 80.8 | 83.3 (-3.2) |
| AP, w/o CN | 65.9 | 97.7 | 98.1 | 87.2 | 77.6 | 82.4 (-4.1) |
| w/o Seg. | 67.8 | 86.9 | 87.4 | 80.7 | 82.2 | 81.5 (-5.0) |
| **SUMMR** | 72.9 | 93.9 | 95.8 | 87.5 | 85.5 | 86.5 |

Table 7: Ablation study of the offline-mined hard negatives in Stage 2

| Setting | R@1 | R@5 | R@10 | mR | Prec. | Avg. |
|---|---|---|---|---|---|---|
| w/o hard negatives | 65.89 | 86.45 | 87.38 | 79.91 | 82.24 | 81.07 (-5.45) |
| w/o BGE-m3 &DINOv2 | 72.43 | 95.79 | 96.73 | 88.32 | 84.11 | 86.21 (-0.31) |
| w/o CLIP | 71.50 | 91.59 | 94.39 | 85.83 | 84.11 | 84.97 (-1.55) |
| w/o Stage 1 | 72.43 | 92.99 | 93.93 | 86.45 | 83.64 | 85.05 (-1.47) |
| **SUMMR** | 72.90 | 93.93 | 95.79 | 87.54 | 85.51 | 86.53 |

# F ABLATION STUDIES

## F.1 DETAILED ANALYSIS FOR ABLATION STUDIES OF STAGE 1

We present the complete results with all the metrics in Tab. 5. We also give the specific explanation of each setting, as well as the analysis below.

- **$\mathcal{L}_{\text{ITC}}$ only**: We only use $\mathcal{L}_{\text{ITC}}$ to train, without evolutionary mask. It is similar to the feature fusion baseline with CLIP as backbone in UniIR (Wei et al., 2024). As seen, the performance significantly degrades (5.0 points drop after Stage 2).

- **w/o $\mathcal{L}_{\text{ITC}}$**: Removing $\mathcal{L}_{\text{ITC}}$ from training objective. The performance after Stage 2 drops by 3.9 points. This is because $\mathcal{L}_{\text{ITC}}$ provides the primary supervision for aligning the shared (intersection) concepts between modalities. Without it, the evolutionary mask $M$ becomes inaccurate, which in turn compromises the quality of the constructed training samples for Stage 2.

- **w/o $\mathcal{L}_{\text{GLA}}$**: Removing $\mathcal{L}_{\text{GLA}}$ from training objective. The performance after Stage 2 drops 5.7 points. The reason is that without $\mathcal{L}_{\text{GLA}}$, the global feature and local feature are not well aligned across modalities, which degrade the accuracy of input mask.

- **w/o $\mathcal{L}_{\text{GD}}$**: Removing $\mathcal{L}_{\text{GD}}$ from training objective. The performance after Stage 2 drops 1.6 points. The reason is that without $\mathcal{L}_{\text{GD}}$, $\mathcal{E}_V$ and $\mathcal{E}_L$ fail to preserve modality-specific information that belongs to the different set.

- **w/o $\mathcal{L}_{\text{LD}}$**: Removing $\mathcal{L}_{\text{LD}}$ from training objective. The performance after Stage 2 drops 4.1 points. Since without $\mathcal{L}_{\text{LD}}$, the feature $V'$ and $L'$ is not ensured to be well aligned with the input, thus degrade the generalizability of input mask.

- **w/o $M$**: Removing the input mask of $\mathcal{E}_{VL}$ when extracting cross-modal feature for alignment leads to 0.5 points drop. Without the mask, the model is forced to align all features, including those belonging to the difference set, which causes a direct conflict with the optimization of $\mathcal{L}_{\text{GD}}$, this is also verified in Fig. 4.

- **$\hat{M}$**: Replacing the evolutionary mask with static hard mask leads to 1.7 points drop. Since at the beginning, the hard mask is inaccurate, applying a noisy hard mask at this stage is catastrophic.

- **w/o $W$**: Removing the lightweight projection heads before cross-modal alignments leads to 3.9 points drop. These projection heads are designed to disentangle the feature spaces for alignment and distillation. Without them, the same representation is used for both conflicting tasks—aligning shared concepts via $\mathcal{L}_{\text{ITC}}$ and preserving unique ones via $\mathcal{L}_{\text{GD}}$—leading to optimization instability, as also observed in Fig. 4.

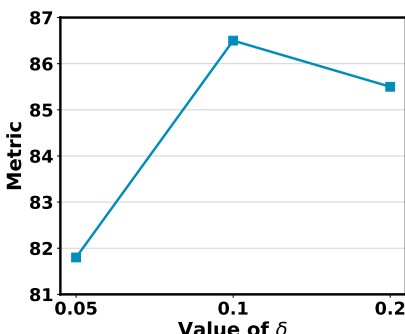

Figure 11: Impact of the margin hyperparameter $\delta$ in GLA

Table 8: Sensitivity analysis of loss weights (1, 2, 3) for Stage 1 training. Final performance is reported after Stage 2, demonstrating that the default setting (all weights set to 1) is a robust choice.

| $\lambda_1$ | $\lambda_2$ | $\lambda_3$ | R@1 | R@5 | R@10 | mR | Prec. | Avg. |
|---|---|---|---|---|---|---|---|---|
| 2 | 1 | 1 | 75.70 | 98.13 | 98.13 | 90.65 | 82.71 | 86.68 |
| 1 | 2 | 1 | 73.83 | 95.79 | 96.73 | 88.79 | 84.11 | 86.45 |
| 1 | 1 | 2 | 71.03 | 97.20 | 97.20 | 88.47 | 79.91 | 84.19 |
| 1 | 1 | 1 | 72.90 | 93.93 | 95.79 | 87.54 | 85.51 | 86.53 |

## F.2 SENSITIVITY ANALYSIS FOR THE MARGIN

We first investigate the impact of the margin $\delta$ in $\mathcal{L}_{\text{GLA}}$. Fig. 11 shows the performance after Stage 2 with different $\delta$ values. As seen, when $\delta = 0.1$, the model achieves the best performance. The larger $\delta$ will force the model to align the difference set, thus lead to over alignment, while a smaller $\delta$ will lead to insufficient alignment. Furthermore, we hypothesize that the optimal $\delta$ is dataset-dependent. Specifically, a larger margin may be beneficial for datasets with higher intrinsic semantic consistency between image-text pairs.

## F.3 SENSITIVITY ANALYSIS FOR LOSS WEIGHTS

We then investigate the impact of the loss weights $\lambda_1$, $\lambda_2$ and $\lambda_3$, as Tab. 8 shows. During training, we find the magnitude of $\mathcal{L}_{\text{ITC}}$ is significantly greater than that of the rest losses. To test if the balance of loss could improve the performance, we increase the weight of a single loss to 2 at a time. Tab. 8 shows the performance after Stage 2 with different loss weights, when increasing $\lambda_1$ or $\lambda_2$, the metric changes slightly, which suggests that the performance is not sensitive to the weight of $\mathcal{L}_{\text{GLA}}$ and $\mathcal{L}_{\text{GD}}$, when increasing $\lambda_3$, the metric drops 2.33 points. The results suggest that $\lambda_1 = \lambda_2 = \lambda_3 = 1$ is a proper setting.

## F.4 OFFLINE-MINED HARD NEGATIVES IN STAGE 2

We perform a series of ablation studies to systematically analyze the contribution of the offline-mined hard negative. As presented in Tab. 7, a key observation is that the complete ablation of the offline-mined hard negatives ('w/o hard negatives') results in a substantial performance degradation of 5.45 points. This sharp decline can be attributed to the inherent limitations of our constructed negative samples, which are constructed solely by deleting information and thus cannot simulate modifications within the original content. The introduction of hard negatives is crucial to compensate for this deficiency. To further dissect its effectiveness, we also evaluate the model by ablating one category of hard negatives at a time. The consistent, albeit smaller, performance drops observed in these settings underscore the individual contribution of each type of hard negatives.

Table 9: Ablation study validating the two-stage training process for the SUMMR-C.

| Setting | R@1 | R@5 | R@10 | mR | Prec. | Avg. |
|---|---|---|---|---|---|---|
| Stage 1 only | 63.6 | 83.6 | 85.5 | 77.6 | 80.8 | 79.2 (-8.5) |
| Stage 2 only | 67.8 | 95.8 | 96.2 | 86.6 | 75.7 | 81.2 (-6.5) |
| **SUMMR-C** | 77.6 | 97.2 | 97.7 | 90.8 | 84.6 | 87.7 |

### F.5 ABLATION STUDY ON CLIP BACKBONE

Given that the CLIP backbone, unlike the BGE-m3+DINOv2 combination, already possesses a coarse-grained cross-modal alignment from its extensive pre-training, we conduct a focused ablation study to validate the necessity of our two-stage approach for this architecture. The results, presented in Table 9, compare the complete SUMMR-C against two simplified settings:

- **Stage 1 Only**: The model is trained exclusively with our Stage 1 and evaluated directly, completely omitting the Stage 2 fine-tuning.
- **Stage 2 Only**: We bypass Stage 1 entirely. The Stage 2 is applied directly to the off-the-shelf CLIP backbone, while the auxiliary modules ($\mathcal{E}_{VL}$, $\mathcal{A}_V$, $\mathcal{A}_L$) are trained from scratch.

The results reveal two critical insights. First, progressing from the "Stage 1 Only" result to the full two-stage model yields a substantial performance gain of 8.5 points, underscoring the vital role of Stage 2. Second, the "Stage 2 Only" configuration underperforms the complete model by a significant margin of 6.5 points. This drop demonstrates that our Stage 1 is crucial for establishing a proper foundation for Stage 2, even for a pre-aligned backbone like CLIP. Collectively, these findings confirm the synergistic and indispensable nature of our two-stage methodology.

### F.6 IMPACT OF IMAGE SEGMENTATION METHOD IN STAGE 2

To justify our choice of a lightweight, clustering-based segmentation method for image masking in Stage 2, we conducted an ablation study comparing it against a state-of-the-art general-purpose segmentation model, Segment Anything Model (SAM) (Kirillov et al., 2023). While integrating a powerful model like SAM might seem like a straightforward path to improvement, our results in Table 10 demonstrate that this is not the case. Our simpler, iterative hierarchical clustering method not only outperforms the SAM-based approach across all key metrics but is also significantly more efficient, reducing per-step training time by over $3\times$ (2.84s vs. 9.24s).

We attribute this seemingly counter-intuitive result to the specific goal of segmentation within our framework. The purpose is not to achieve pixel-perfect segmentation, but rather to generate coarse, semantically coherent regions that can be manipulated to construct meaningful positive and negative training samples. Here, SAM's primary strength—its ability to produce highly detailed, fine-grained masks—becomes a liability. It often fragments a single semantic object (e.g., a "bear") into multiple distinct parts (e.g., head, torso, paws). Masking just one of these small fragments is often insufficient to meaningfully alter the image's core semantics, leading to the creation of weak or ambiguous training signals.

In contrast, our clustering-based approach naturally groups visually and semantically similar patches, producing coarser segments that better align with whole objects or large object parts. This makes it far more effective for our purpose: masking an entire "bear" segment creates a strong positive sample (as the concept is recoverable from the text), whereas masking a background segment creates a clear hard negative.

Therefore, this ablation confirms that our lightweight segmentation method is not merely a compromise for efficiency; it is functionally superior for the specific task of self-supervised sample generation in our framework.

### F.7 ANALYSIS OF THE EVOLUTIONARY MASK SCHEDULE

The evolutionary mask schedule ($\rho$ annealing from 1 to 0) is a critical component for ensuring training stability. We provide its motivation and empirical validation below.

Table 10: Ablation study on the segmentation method of stage 2.

| Method | R@1 | R@5 | R@10 | mR | Prec. | Avg. | Per-step Training Time (s) |
|--------|-----|-----|------|-----|-------|------|----------------------------|
| SAM | 70.6 | 90.2 | 91.1 | 84.0 | 83.4 | 83.7 | 9.24 |
| Ours | 72.9 | 93.9 | 95.8 | 87.5 | 85.5 | 86.5 | 2.84 |

Table 11: Impact of schedule for evolutionary mask in stage 1.

| Setting | R@1 | R@5 | R@10 | mR | Prec. | Avg. |
|---------|-----|-----|------|-----|-------|------|
| Linear decay | 68.7 | 89.3 | 89.7 | 82.6 | 81.8 | 82.2 |
| Cosine decay | 67.2 | 88.8 | 91.1 | 82.4 | 82.2 | 82.3 |

At the beginning of Stage 1, the model's encoders are not yet aligned, making the signal used to generate the mask noisy and unreliable. Applying a hard binary mask from the outset would force the model to learn from these potentially incorrect disentanglement signals, which could be detrimental. The evolutionary schedule mitigates this by transitioning smoothly from a non-informative, all-ones mask (where the model learns from all features) to relying solely on the learned binary mask. This allows the model to first establish a coarse alignment and then gradually trust its own increasingly confident predictions to perform fine-grained disentanglement.

We conducted two key experiments to validate this design. We first tested if the specific functional form of the decay matters. As shown in Tab. 11, the performance difference between a linear decay and a cosine decay schedule is negligible after Stage 1. This demonstrates that our framework is not sensitive to the precise form of the schedule, as long as the transition is gradual. More importantly, we validated the necessity of a gradual schedule itself. The $\hat{M}$ ablation in Tab. 5 evaluates a model trained with a static hard mask from the beginning of training ($\rho = 0$ throughout). This resulted in a significant 1.7-point performance drop in the final model (84.8 vs. 86.5 Avg.). This result confirms that an abrupt masking strategy is harmful and that the gradual annealing process is vital for achieving optimal performance.

Regarding convergence, it is important to note that training continues for a significant number of steps after $\rho$ has decayed to 0. During this final phase, the model operates with a stable masking policy (i.e., using the hard mask $\hat{M}$ directly). However, $\hat{M}$ itself is not fixed; it remains dynamic and is re-computed for each batch based on the evolving feature similarities. This ensures the model fully converges while perfecting its representation based on its own continuously refined, adaptive disentanglement capability.

## G  VISUALIZATIONS

To further elucidate the internal mechanisms of our method, we present additional visualizations.

### G.1  ANALYSIS OF SIMILARITY SCORE DISTRIBUTIONS AND QDA ROBUSTNESS

To empirically validate the core assumption of our QDA-based thresholding, Fig. 12 visualizes the evolution of the global-text-to-local-image similarity score distributions during Stage 1 training. This analysis offers two key insights into our model's learning dynamics.

First, the primary finding is that the model successfully learns to create a separable feature space. Initially, the distributions for positive (in-pair) and negative (out-of-pair) scores overlap significantly, indicating that the model cannot distinguish between relevant and irrelevant features. As training progresses, they progressively diverge into two highly separable populations. This separation provides strong quantitative evidence that the model is successfully learning to distinguish shared (intersection) features from unique (difference) ones, thereby generating a reliable signal for the MaskGen module.

Second, a closer inspection reveals a notable nuance: while the negative score distribution consistently forms a clean, unimodal Gaussian, the positive distribution occasionally exhibits minor deviations from this ideal shape. We attribute this asymmetry to the significant disparity in sample sizes used for estimation within each training batch. The negative distribution is estimated from $\mathcal{O}(B_s^2)$ ($B_s$=64, which is the per-GPU batch-size) out-of-pair comparisons, benefiting from the Law of Large Numbers

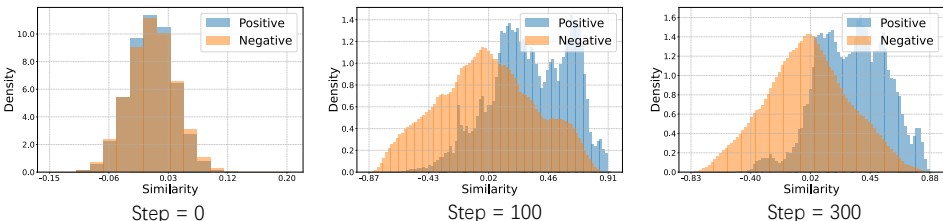

Figure 12: The evolution of similarity score distributions for positive (in-pair) and negative (out-of-pair) text-global-to-image-local comparisons during Stage 1 training.

to form a stable Gaussian. In contrast, the positive distribution is derived from only $\mathcal{O}(B_s)$ in-pair comparisons, making it more susceptible to batch-specific variance and slight multimodalities.

Crucially, this observation does not undermine the method's efficacy. The divergence between the two distributions is substantial enough that even with minor imperfections in the positive distribution's shape, QDA can robustly identify an optimal decision boundary (threshold $\tau$) that effectively separates the intersection from the difference. This demonstrates that our framework is resilient to slight violations of the perfect Gaussian assumption, a key factor in its practical effectiveness.

While the current approach proves robust, these observations also suggest clear avenues for future refinement. Two potential enhancements could further mitigate the impact of this distributional variance: 1) **Cross-Device Score Aggregation**: A simple yet effective engineering improvement would be to aggregate similarity scores across all GPUs before computing the QDA parameters. Our current implementation calculates these statistics on a per-GPU basis for computational efficiency. By synchronizing scores, we would substantially increase the sample size for the positive distribution, yielding a more stable estimate that better approximates a unimodal Gaussian. 2) **Adoption of Mixture Discriminant Analysis (MDA):** A more theoretically robust solution would be to replace QDA with a more flexible modeling approach, such as Mixture Discriminant Analysis (MDA). By modeling each class distribution as a Gaussian Mixture Model (GMM) instead of a single Gaussian, MDA could explicitly capture any observed multimodality. This would allow the thresholding mechanism to adapt to more complex data distributions, providing a more principled and potentially more accurate method for disentanglement.

## G.2 VISUALIZATION OF DISENTANGLED INTERSECTION

To demonstrate the model's ability to disentangle shared concepts from modality-specific information, we visualize the global-to-local similarity scores learned in Stage 1 on some image-text pairs. Fig. 13 illustrates cases with a single object in the semantic intersection. Each triplet, consisting of the original image and two cross-modal similarity heatmaps, shows that warmer image regions (red) and darker text tokens correspond to higher similarity. Our model successfully identifies the core semantic concept (e.g., "children") while assigning low similarity to irrelevant background elements and temporal metadata (e.g., "2020-08-20").

In contrast, Fig. 14 presents scenarios with multiple objects. Notably, our method highlights all concepts described in the text (e.g., the boy, dog, and ball) rather than focusing solely on the primary subject (e.g., the boy). This is because our approach considers all entities, including both subjects and objects, as integral contributors to the final joint representation.

## G.3 ANALYSIS OF SENSITIVITY TO EDGE SEMANTICS

To analyze how the joint embedding responds to low-level visual variations—or "edge semantics"—we conducted a targeted case study on color, orientation, and occlusion. As shown in Fig. 15, we anchored our analysis on the text "Tennis player Nadal in a match" and its corresponding image. We then generated variants of this image by applying common transformations: horizontal flipping (orientation), cropping (occlusion), and altering the background (color).

We computed the cosine similarity between the joint embedding of the original pair and that of each transformed pair. The results reveal a clear pattern: the model demonstrates remarkable robustness

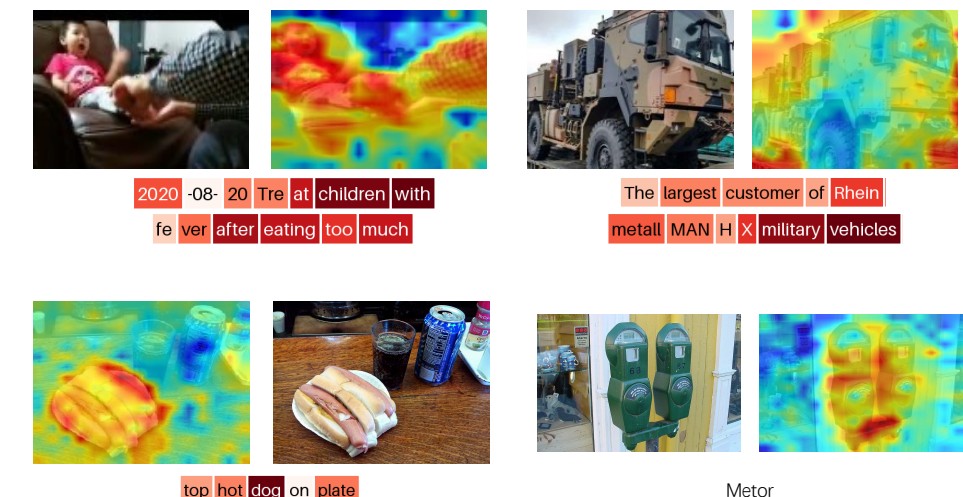

Figure 13: Visualization the global-to-local similarity of single-object case

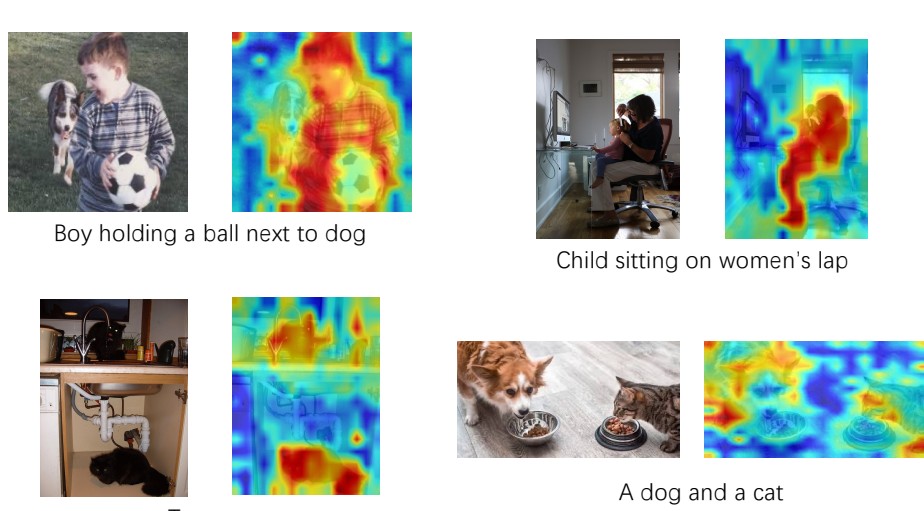

Figure 14: Visualization the global-to-local similarity of multiple-object case

to geometric transformations like orientation (sim: 0.98) and occlusion (sim: 0.97). In contrast, a significant change in background color leads to a substantial drop in similarity (sim: 0.77).

This behavior is insightful. The robustness to orientation and occlusion is a desirable trait, likely inherited from the extensive data augmentation (e.g., random flips and crops) used to pre-train the vision backbone, which teaches geometric invariance. Conversely, the sensitivity to color is also logical, as color is a powerful semantic attribute frequently described in text. A drastic color change can alter the image's context, and the model correctly identifies this as a larger semantic shift than a simple change in perspective or framing.

## H   EVALUATION ON OTHER BENCHMARK

### H.1   REFERRING IMAGE SEGMENTATION EVALUATION FOR STAGE 1

To quantitatively verify that Stage 1 successfully learns to identify the semantic 'intersection' between an image and text, we evaluated its disentanglement capability on the standard Referring Image

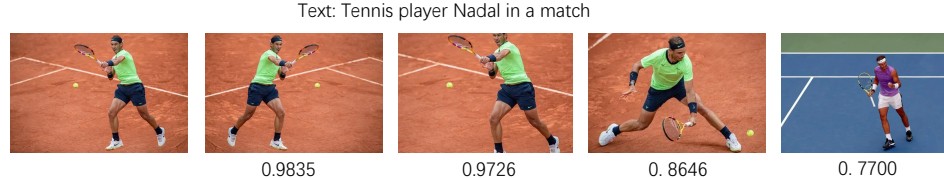

Text: Tennis player Nadal in a match

| 0.9835 | 0.9726 | 0. 8646 | 0. 7700 |

Figure 15: The influence of edge semantics

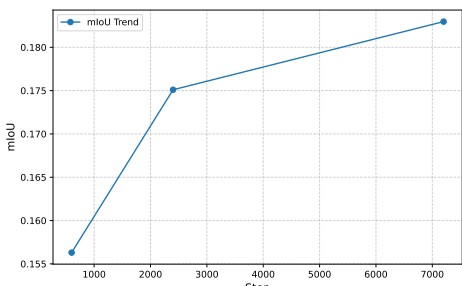

Figure 16: MIoU trend for stage 1.

Segmentation (RIS) benchmark, RefCOCO (Nagaraja et al., 2016). The goal of RIS is to segment the specific image region described by a natural language expression.

To adapt our model for this task, we generate a segmentation mask by thresholding the global-text-to-local-image similarity scores produced by the model at different training checkpoints. Given that RefCOCO images often contain 'distractors' (e.g., other objects of the same class as the target), we used a stricter, fixed threshold for this evaluation rather than the adaptive threshold $\tau_V$ used during training. This ensures we only segment regions with the highest confidence of matching the text description.

As shown in Fig. 16, the mean Intersection over Union (mIoU) score steadily improves as training progresses. This trend provides strong quantitative evidence that our model progressively learns a more accurate representation of the semantic intersection.

While the absolute mIoU is modest compared to state-of-the-art RIS models, this result is expected and highlights a crucial distinction in objectives: 1) Our model is trained on 0.8M noisy, web-crawled image-text pairs from LAION. In contrast, specialized RIS models are typically fine-tuned on the clean, curated COCO dataset itself, giving them a significant in-domain advantage. 2) Most importantly, the tasks have different goals. RIS aims to isolate a single *subject* entity described in the text (e.g., segmenting only 'the dog on the left'). In contrast, our framework is designed to identify the *entire shared semantic space*. For a phrase like "a boy holding a ball," our model correctly identifies both the 'boy' and the 'ball' as part of the intersection, as both contribute to the joint representation. This broader definition of 'intersection' naturally leads to a lower mIoU on a task that rewards segmenting a single, specific object.

Therefore, this evaluation serves not as a direct performance comparison against specialized models, but as a robust validation of Stage 1's core mechanism: its ability to learn and progressively refine the semantic intersection between modalities, entirely without direct supervision.

## H.2    EVALUATION ON MBEIR BENCHMARK

To better understand the of objective of stage 1. We conduct ablation on the UM2UM subset of MBEIR benchmark, including single NIGHTS (image2image), COCO (cross-modal) and WebQA (text2text), as Tab. 12 shows. For COCO, we report the average metric of image2text and text2image task. In which CLIP-ZS means zero-shot CLIP, while CLIP-FT mean CLIP that is trained and tested on each task, respectively. As seen, SUMMR-B+D stage 1 achieve the best balance on three tasks

Table 12: Ablation and comparison on subset of MBEIR.

| Model | WebQA | NIGHTS | COCO | Avg. |
|---|---|---|---|---|
| BGE | 84.2 | - | - | 29.0 |
| DINO | - | 31.8 | - | 10.6 |
| SUMMR-B+D Stage 1 | 72.6 | 31.1 | 30.3 | 44.7 |
| SUMMR-B+D Stage 1 w/o $W$ | 65.9 | 30.7 | 31.5 | 42.7 |
| SUMMR-B+D Stage 2 | 71.9 | 22.7 | 1.1 | 31.9 |
| CLIP-ZS (ViT-B) | 23.1 | 25.7 | 55.3 | 34.7 |
| CLIP-ZS (ViT-L) | 36.2 | 26.1 | 70.0 | 44.1 |
| CLIP-FT (ViT-L) | 81.7 | 33.5 | 85.1 | 66.8 |

among zero-shot models. Specifically, SUMMR-B+D outperforms BGE and DINO on COCO, and outperform CLIP-ZS on single-modal task (NIGHTS and WebQA), which demonstrate that stage 1 success to disentangle and alignment. We also observed that: 1) by removing the lightweight projection head, the average metric drop by 2.0, which demonstrate its advantage on disentangle. 2) we find that stage 2 lead to performance drop on UM2UM task, we argue the reason is that stage 2 is trained on MM2MM data only, thus its UM2UM ability is degraded. Finally, we observe that finetuning leads to great improvement on the corresponding task (i.e., CLIP-ZS (ViT-L) vs CLIP-FT (ViT-L)), which prove that in-domain training data is crucial for the performance.

## I  ANALYSIS OF A STRAIGHTFORWARD VLM ADAPTATION FOR SYMMETRIC RETRIEVAL

To investigate why a "straightforward extension" of existing VLMs is insufficient for the symmetric MM2MM task, we analyzed the impact of training data synthesis. The most relevant prior work, mmE5 (Chen et al., 2025), synthesizes a large dataset of partially symmetric pairs by finding a visually similar positive image for a source image and then generating captions for both. While this approach appears suitable, it underperforms SUMMR on our benchmark.

To isolate the effect of this data generation strategy, we conducted a new experiment. We fine-tuned a VLM (Llama-3.2-11B-Vision, consistent with mmE5) exclusively on the publicly available synthesized symmetric data from mmE5. The model was then evaluated on our sym-MM2MM benchmark.

The model trained solely on mmE5's data achieved an average score of 69.39. This is substantially lower than SUMMR-C (87.69) and even underperforms several baselines trained on the more general MMEB benchmark, such as VLM2Vec (71.96). This result strongly suggests that generating symmetric pairs based on simple visual similarity is not enough. Such a strategy may create pairs that are topically related but does not enforce the strict semantic equivalence and interchangeability that define the symmetric MM2MM task. For example, two different images of "a dog on grass" are visually similar but may represent entirely different entities with unique, unstated attributes. In contrast, SUMMR's self-supervised framework learns to disentangle the shared (intersection) and unique (difference) information within a single multimodal entity, enabling the construction of truly symmetric positive samples. This finding validates that a specialized, disentanglement-based approach like SUMMR is necessary to master the nuances of this task, which cannot be solved by simply scaling up naively synthesized symmetric data.

## J  TRAINING TIME

We show the per-step training time of previous work and SUMMR (stage 1 and stage 2, short as S1, S2, respectively) in Tab. 13. For fair comparison, the batch-size are all set as 32, the VLM-based methods all use LORA finetuning with rank 8. For VLM-based methods, since they are use contrastive loss to train, thus the only factor that influence training time is backbone, thus we divide them into 3 categories based on their backbone: 1) Qwen2-VL-7B (shorted as Qwen), including MM-Embed, VLM2Vec, GME, Unite and LamRA. 2) LLaVA-OneVision-7B, including UniME

Table 13: Per step training time

| Model | CLIP-SF | BGE-VL | Qwen | Llama | SUMMR-B+D S1 | SUMMR-B+D S2 | SUMMR-C S1 | SUMMR-C S2 |
|---|---|---|---|---|---|---|---|---|
| Training time (s) | 1.26 | 0.30 | 7.12 | 10.83 | 0.79 | 0.53 | 1.29 | 0.30 |

and 3) Llama-3.2-11B-Vision (shorted as Llama), including mmE5. For convenient, we only show training time for each category, we choose Qwen2-VL-7B and Llama-3.2-11B-Vision for comparison.

As can be observed in Tab. 13, Stage 1 is intentionally more computationally intensive, primarily driven by the $\mathcal{O}(n^2)$ pairwise local similarity calculation required by the $\mathcal{L}_{LD}$ loss and the overhead of its multi-objective framework involving teacher models. In sharp contrast, Stage 2 is significantly more efficient, functioning as a lightweight fine-tuning step with a single contrastive loss; its rapid segmentation process is far outweighed by the removal of the expensive distillation objectives. This two-stage approach embodies a strategic trade-off: investing computational effort upfront to learn a robust disentanglement mechanism, which then enables a highly efficient final tuning phase. Furthermore, even the more complex Stage 1 is 5-13× faster than the VLM baselines, highlighting the fundamental efficiency of our approach, which stems from two key design choices. First, our model is architecturally far smaller, utilizing dedicated encoders with hundreds of millions of parameters (e.g., 0.7B for SUMMR-B+D) instead of monolithic VLMs with over 7 billion. Second, we employ LoRA to fine-tune only a small fraction of these encoder parameters, which dramatically reduces the computational cost of the backward pass. This combination of a lightweight architecture and an efficient training strategy is what allows SUMMR to be not only more accurate but also significantly more practical for large-scale training and real-world deployment.

## K  LIMITATIONS AND FUTURE WORK

### K.1  LIMITATIONS

While our experimental results demonstrate the effectiveness of the proposed method, our constructed samples have a notable limitation: they primarily simulate information deletion but not modification or addition. This makes offline-mined hard negatives crucial for compensating for this deficiency by introducing more diverse and challenging examples. Furthermore, the construction of our high-quality *sym-MM2MM* dataset currently relies on manual annotation, which inherently limits its scale.

### K.2  FUTURE WORK

Our future research will focus on developing automated methods to construct more sophisticated training samples that simulate a broader spectrum of transformations, including content modification and addition. The recent emergence of Native Multimodal Models (i.e., NextGPT (Wu et al., 2024) and AnyGTP (Zhan et al., 2024)), which can generate images and text simultaneously, presents a promising direction for automating the synthesis of both training and testing data, potentially overcoming the scalability bottleneck of manual annotation. Furthermore, we plan to investigate how performance scales by training SUMMR on much larger uncurated web datasets (>10M pairs) and with larger backbone models (>2B parameters).

A second direction is to explore methodological refinements, such as making the mask-generation module fully differentiable. This would replace the current non-differentiable QDA thresholding and could allow for more direct, end-to-end optimization of the disentanglement process.

Another promising direction for future work is to extend our framework to other pairings, such as text-audio and text-video, by leveraging large-scale unlabeled datasets (e.g., AudioSet (Gemmeke et al., 2017), HowTo100M (Miech et al., 2019)) and strong unimodal encoders (e.g., wav2vec2 (Baevski et al., 2020), VideoMAE (Tong et al., 2022)). This would involve adapting the masking mechanism to the temporal nature of audio and video data but would validate SUMMR's potential as a general paradigm for complex retrieval tasks, making it a strong candidate for emerging benchmarks like a potential MMEB-v2.

## L    THE USAGE OF LARGE LANGUAGE MODELS (LLMs)

In this paper, we primarily utilize large language models to assist with two main tasks: refining the manuscript's writing and enhancing the layout and presentation of tables.

