# OpenReview forum: "SUMMR: Self-supervised Joint Representation Learning for Symmetric Multimodal Retrieval"
_ICLR.cc/2026/Conference — Submitted to ICLR 2026_

### Official Review · Reviewer_hd9B · 2025-10-28

**Soundness:** 3
**Presentation:** 3
**Contribution:** 2
**Rating:** 4
**Confidence:** 4

**Summary:**

The paper identifies multimodality-to-multimododality (MM2MM) retrieval as a largely neglected problem. To eliminate the expensive manual annotation that has stalled progress, the authors propose SUMMR, a two-stage self-supervised framework for this task. It first learns an intersection mask that disentangles shared concepts from modality-specific details inside any unlabeled image-text pair, using distillation-guided global-to-local alignment and adaptive QDA thresholding.  It then exploits the mask to automatically create positives (intersection masked) and hard negatives (difference masked), and trains a compact late-fusion encoder with contrastive learning. Training on 800 k LAION-5B pairs and evaluating on a new human-curated sym-MM2MM benchmark (1 M distractors), SUMMR outperforms some strong supervised models while being >50× smaller and 5× lower-dimensional embeddings.

**Strengths:**

1. The paper proposes a multimodality-to-multimododality (MM2MM) retrieval task, and build a dataset for the task.
2. The paper proposes SUMMER, a two-stage self-supervised framework for this task, which outperforms some strong supervised models while being >50× smaller and 5× lower-dimensional embeddings.

**Weaknesses:**

1. Constructed samples only simulate deletion, not addition or modification, heavy reliance on offline hard-negative mining to cover the gap. Image masking needs semantic segmentation; iterative clustering introduces extra hyper-parameters (t, Kmax, rmax) that may not generalise across domains.
2. SUMMER is only for the MM2MM task proposed in the paper. The MM2MM task might be a interesting task, but it's not that commonly used, nor is it that basic.

**Questions:**

1. How does performance scale with much larger uncurated web data (>10 M pairs) and larger backbones (>2 B)?
2. Can the mask-generation module be made differentiable end-to-end instead of relying on QDA thresholding?
3. What is the exact computational overhead of segmentation-based masking at inference time?
4. How sensitive are thresholds τV, τL to domain shift, and can they be predicted meta-learned for new corpora?

---

> ### Author Response · Authors · 2025-11-21
> **Response to Reviewer hd9B (Part 1)**
>
> _W1: Constructed samples only simulate deletion, not addition or modification, heavy reliance on offline hard-negative mining to cover the gap. Image masking needs semantic segmentation; iterative clustering introduces extra hyper-parameters (t, Kmax, rmax) that may not generalise across domains._
>
>
> The reviewer correctly notes that our self-supervised sample construction simulates **information deletion**. This is a deliberate, two-part strategy. Our self-supervised masking forms the core of our method’s success, providing **a strong performance baseline on its own**. As shown in our ablations (Table 7), the model trained without any offline hard negatives still achieves a competitive score of 81.07, outperforming many supervised baselines. We then complement this with offline hard-negative mining—**a standard and computationally inexpensive practice** in modern retrieval—which enhances the model’s ability to handle **more complex** semantic distinctions (e.g., content modification).
>
>
>
> Regarding image masking, we wish to clarify that our method **does not depend on an external semantic segmentation model**. Instead, we perform **lightweight, unsupervised hierarchical clustering on the model’s own visual features** (Appendix C.2), making the framework self-contained and efficient. While we acknowledge this process introduces hyperparameters, our **iterative algorithm (Algorithm 1) is designed to be adaptive**, making the process robust across diverse images by using these parameters as **common-sense guardrails** rather than sensitive, domain-specific tuning knobs.
>
>
>
> Crucially, the segmentation **does not need to be perfect**. Our approach aligns with the principles of contrastive learning, where inherently imperfect augmentations—such as random cropping, which may remove key objects—**are routinely used to learn robust and generalizable representations**. Just as standard methods learn from these imperfectly constructed positive/negative pairs, SUMMR effectively learns to disentangle shared and unique concepts from our coarse-grained, imperfect segment masks.
>
>
>
> _W2: SUMMER is only for the MM2MM task proposed in the paper. The MM2MM task might be a interesting task, but it's not that commonly used, nor is it that basic._
>
>
>
> We respectfully argue that the symmetric Multimodality-to-Multimodality (MM2MM) task is **neither niche nor basic**. Instead, it is a highly challenging and practically relevant problem that exposes fundamental limitations in current large-scale models. The SUMMR framework, designed to solve this task, learns a powerful and inherently **generalizable representation** that is a valuable contribution in its own right.
>
> 1. **The Symmetric MM2MM Task is Critical and Widespread, Not Niche.**
>
> While less common in academic benchmarks, symmetric MM2MM retrieval is crucial for many **high-value, real-world applications** that require matching a holistic concept rather than just a query's surface features. The lack of prior work is due to a lack of formalization and benchmarks—a gap our paper directly addresses. Consider these concrete scenarios, though **many more are readily apparent across various industries**:
>
> + **E-commerce & Product Discovery:** A user searches with a photo of a chair (`Image`) and the text "in a dark wood finish" (`Text`). The goal is to find a product listing that has an image of the correct chair (`Image'`) but whose own description is different, like "mid-century modern armchair" (`Text'`). The system must understand that both pairs represent the **same conceptual product**.
> + **Design & Creative Industries:** An interior designer searches with a photo of a floral pattern (`Image`) and the text "used as wallpaper" (`Text`). The system should retrieve an image of that same pattern used on a cushion (`Image'`), even if its caption is "bold accent cushion with vintage print" (`Text'`). The retrieval is symmetric because the core shared entity is the **pattern**, not the specific item.
> + **Law Enforcement & Intelligence Analysis:** An investigator queries with a blurry CCTV photo of a car (`Image`) and an eyewitness description like "blue sedan, dented door" (`Text`). The system must search a database to find a clear photo of the same car (`Image'`) paired with its official registration details (`Text'`). This critical task requires **fusing partial information from both modalities** to identify a single, real-world entity.

---

> ### Author Response · Authors · 2025-11-22
> **Response to Reviewer hd9B (Part 2)**
>
> 2. **Symmetric MM2MM is a General and Foundational Retrieval Paradigm.**
>
> We respectfully disagree with the characterization of the task as "not that basic." Our work is best contextualized by the taxonomy in **Figure 1**, which categorizes retrieval paradigms along two fundamental axes: symmetry (symmetric/asymmetric) and the number of modalities in the query/candidate (MM2MM vs. UM2MM/MM2UM).
>
> While tasks like UM2MM and asymmetric MM2MM have historically received more attention, **each quadrant in this taxonomy represents a distinct and critical class of real-world problems.** The historic focus on asymmetric tasks does not diminish the foundational importance of the symmetric paradigm. The relative lack of prior work on symmetric MM2MM should not be mistaken for a lack of importance. On the contrary, it highlights a significant and overlooked gap in the field. **A gap in the literature is not a sign of a 'niche' problem, but an invitation for foundational research, which our work provides.**
>
> Furthermore, we argue that symmetric retrieval is not 'less basic' but is, in fact, **a more general and fundamental paradigm**. It can be seen as a generalized framework that subsumes many specialized asymmetric tasks. For instance, classic text-to-image retrieval can be framed as a special case of our symmetric task by using a query of `(empty_image, text)`. By successfully tackling the more general symmetric case, we are developing models that are inherently more powerful and addressing a critical, long-overlooked quadrant of the retrieval landscape.
>
> 3. **The Core Principles of SUMMR are Highly Generalizable.**
>
> Although demonstrated on the image-text MM2MM task, our core contribution—a **self-supervised framework for disentangling shared vs. unique information**—is not limited to this problem. This principle can be readily applied to any task involving paired, partially overlapping data, such as **video-audio alignment, text-to-code retrieval, or aligning different medical imaging modalities (e.g., CT and MRI)**. We chose symmetric MM2MM as a challenging and novel testbed, but the underlying self-supervised strategy has far broader applicability.
>
> To address the above concern, we have mentioned in the introduction (Lines 41-44) that
>
> > Beyond e-commerce, symmetric MM2MM retrieval has potential applications in areas such as news article retrieval, recipe recommendations, travel destination discovery, and interior design style matching, to name just a few. This highlights the urgent need for a comprehensive approach to symmetric MM2MM retrieval.
> >
>
>
>
> _Q1: How does performance scale with much larger uncurated web data (>10 M pairs) and larger backbones (>2 B)?_
>
>
>
> First, we would like to politely argue that for real-world retrieval systems, which must handle a massive volume of queries, model size and efficiency are paramount. Large, computationally intensive models can be impractical for deployment. As we discuss in our paper **(Section A.2, Lines 900-903)**, the high computational cost of VLM-based approaches is a significant bottleneck:
>
> > While promising, the high computational complexity of VLMs makes them less practical for real-world retrieval systems that require rapid throughput; for example, Google processes approximately 6.3 million search queries per minute, necessitating embedding models with fast inference speeds.
> >
>
> Our work demonstrates that a task-aligned, self-supervised approach can be both more effective and vastly more efficient. A key contribution of our paper is showing that even with a compact architecture, SUMMR achieves state-of-the-art performance, outperforming much larger models. As noted in our main results **(Section 5, Lines 421-424)**:
>
> > Our best model, SUMMR-C, achieving an average score of 87.69. This surpasses the strongest VLM-based method, mmE5, by 3.42 points, being over 50 times smaller in model size (0.20B vs. 10.12B) and producing an embedding vector over 5 times more compact (768 vs. 4096 dimensions).
> >
>
> That said, we absolutely agree that investigating how our framework's performance scales with larger datasets and backbones is a very interesting and important future direction. Due to the limited time and resources available during the rebuttal period, we were unable to conduct these large-scale experiments. However, we have explicitly identified this as a key next step in our future work section. As stated in **Appendix K.2 (Lines 1879-1880)**:
>
> > Furthermore, we plan to investigate how performance scales by training SUMMR on much larger uncurated web datasets (>10M pairs) and with larger backbone models (>2B parameters).
> >

---

> ### Author Response · Authors · 2025-11-22
> **Response to Reviewer hd9B (Part 3)**
>
> _Q2: Can the mask-generation module be made differentiable end-to-end instead of relying on QDA thresholding?_
>
>
> First, we want to clarify that while the QDA thresholding step itself is non-differentiable, the learning process remains flexible and end-to-end. **The encoders that produce the feature embeddings are fully differentiable. The training objective drives these encoders to generate features that are increasingly separable by the QDA classifier.** Therefore, the model continuously learns to produce better features that result in a better mask. This principled approach of using a non-differentiable statistical boundary within a larger differentiable framework is a recognized design pattern for robustly separating signal from noise, as seen in other works [R1].
>
>
>
> Second, this design can be superior to a fully differentiable approach (e.g., using a sigmoid layer) for two key reasons:
>
> 1. **Avoiding Trivial Solutions:** A fully differentiable mask generator is highly susceptible to finding a trivial solution (i.e., model collapse). The model could easily learn to always output a mask of all-ones or all-zeros to minimize the loss without learning any meaningful disentanglement.
> 2. **Principled Statistical Grounding:** QDA provides a much stronger learning signal. By explicitly modeling the features as two distinct distributions, it forces the encoders to produce genuinely separable features, rather than allowing the mask generator to find a lazy shortcut. This statistical grounding, combined with our evolutionary masking schedule, provides critical stability to the self-supervised training process.
>
> Finally, we agree that exploring fully differentiable masking is a valuable direction for future work and have added an dicussion in **Appendix K (Limitations and Future Work)** as:
>
> >A second direction is to explore methodological refinements, such as making the mask-generation module fully differentiable. This would replace the current non-differentiable QDA thresholding and could allow for more direct, end-to-end optimization of the disentanglement process.
> >

---

> ### Author Response · Authors · 2025-12-02
> **Response to Reviewer hd9B (Part 4)**
>
> _Q3: What is the exact computational overhead of segmentation-based masking at inference time?_
>
>
>
> The exact computational overhead of our segmentation-based masking at **inference time** is **zero**.
>
>
>
> This is because the segmentation-based masking is exclusively a part of our **self-supervised training pipeline** (specifically, Stage 2). Its sole purpose is to **automatically generate positive and negative samples** from unlabeled data to train our model effectively. Once the model is trained, this entire sample construction mechanism, including the iterative clustering and masking, is **completely discarded**.
>
>
>
> At **inference time**, our model functions as a standard dual-encoder followed by a fusion encoder. It performs a single, efficient forward pass: an image-text pair is processed by the vision and language encoders, projected by the adapters, and fused by the lightweight VL-encoder to produce the final embedding. No clustering or masking is involved.
>
>
>
> This design ensures that SUMMR remains **highly efficient and practical for real-world, large-scale retrieval scenarios**, where low-latency inference is paramount. We have added a clarification in Lines 376-377, Page 6 as:
>
> > Note that this entire sample generation pipeline, including segmentation and masking, is used exclusively during training and is completely discarded at inference time, resulting in zero computational overhead.
> >
>
>
>
> _Q4: How sensitive are thresholds $τ_V$, $τ_L$ to domain shift, and can they be predicted meta-learned for new corpora?_
>
>
> The key design principle of our thresholds, $\tau_V$ and $\tau_L$, is that they are **adaptive and not fixed hyperparameters**. As detailed in Section 4.1.2, they are recalculated on-the-fly for each training batch using Quadratic Discriminant Analysis (QDA). This means that if the model is exposed to a new domain, the feature distributions will naturally change, and our QDA-based thresholding will **automatically adjust to the statistics of the new data**. This provides built-in robustness to domain shifts.
>
>
>
> It is also crucial to clarify the lifecycle of these thresholds within our two-stage framework:
>
> 1. **Stage 1 Training:** The QDA and adaptive thresholds are used to train a model that can effectively disentangle shared and unique information. The goal is to produce high-quality, separable similarity distributions.
> 2. **Stage 2 Training:** The trained Stage 1 model is then used to **generate** the thresholds $\tau_V$ and $\tau_L$ for the unlabeled data in Stage 2. This is the only point where generalization is needed for $\tau$—the model must be able to generate meaningful thresholds for the Stage 2 data, which is a reasonable expectation as both stages use diverse web-scale data.
> 3. **Inference:** Most importantly, **during inference, the entire thresholding and masking mechanism is discarded**. The final model performs a single, efficient forward pass to generate an embedding. **No thresholds $\tau_V$ or $\tau_L$ are ever computed at inference time.**
>
> Regarding the suggestion of meta-learning, it is an excellent idea for frameworks with fixed thresholds. However, our on-the-fly QDA can be seen as a form of rapid, local adaptation that already accomplishes this goal efficiently within the training process. This design ensures that our method is both robust to domain shifts during training and maximally efficient at inference. To address this point, **we have added a dicussion in Appendix C.4 in Page 21**.
>
> [R1] Zhang et al, Negative-Aware Attention Framework for Image-Text Matching, CVPR 2022.

---

### Official Review · Reviewer_PdWd · 2025-10-30

**Soundness:** 3
**Presentation:** 3
**Contribution:** 3
**Rating:** 6
**Confidence:** 4

**Summary:**

This paper introduces the task of Symmetric Multimodal-to-Multimodal (sym-MM2MM) retrieval and proposes SUMMR, a self-supervised framework. Its contributions are:

*   **Novel Two-Stage Training Paradigm**
    *   Stage 1 learns to disentangle shared and unique semantics from image-text pairs.
    *   Stage 2 leverages this disentanglement to auto-construct samples for contrastive learning.

*   **Dedicated Benchmark Dataset**
    *   Constructed via a generative model-assisted pipeline with human annotation.
    *   Provides a reliable standard for evaluating sym-MM2MM retrieval.

*   **Strong Empirical Performance**
    *   SUMMR outperforms traditional supervised methods that are 50x larger in scale.
    *   It uses a compact model (0.2B parameters) and offers an efficient, annotation-free solution.

**Strengths:**

The strengths of this paper can be summarized as follows:

1.   **Originality:** This paper demonstrates high originality by being the first to formally define the "symmetric MM2MM retrieval" task. Its core contribution is a highly creative and novel self-supervised framework that leverages semantic disentanglement to generate its own training signal, moving beyond reliance on manually annotated data.

2.   **Quality:** The methodology is rigorously designed and validated, which is evidenced by the comprehensive ablation studies that effectively justify the necessity of each proposed component. The framework's robustness and quality are further underscored by its superior performance across different backbone architectures.

3.   **Clarity:** The paper is exceptionally well-written. Despite the inherent complexity of the proposed two-stage framework, the logical flow remains easy to follow, a feat achieved through well-designed figures and a clear, staged explanation of the pipeline.

4.  **Significance:** This work is of high significance as it directly addresses the critical bottleneck of annotation scarcity in symmetric retrieval scenarios. Furthermore, it introduces a self-supervised disentanglement paradigm that could inspire future research in multimodal representation learning, and the release of the benchmark dataset lays a solid foundation for subsequent work in this new area.

**Weaknesses:**

The weaknesses of this paper can be summarized as follows:

1.   **Experimental Comparisons:** While the comparisons against existing methods are favorable, the experimental setup lacks a crucial baseline, as the authors omit supervised fine-tuning of strong models (e.g., CLIP or mmE5) on their proposed benchmark. This omission makes it difficult to ascertain the actual performance gain of their self-supervised framework versus a direct supervised approach on the same task.

2.  **Validation of Disentanglement:** The validation for the core innovation of "semantic disentanglement" remains indirect, as it relies solely on downstream retrieval performance and similarity heatmaps. A more direct, quantitative evaluation of the Stage-1 intersection masks is missing, such as measuring their IoU against ground-truth segmentation masks to definitively prove the accurate localization of shared concepts.

3.   **Robustness Analysis:** The proposed two-stage pipeline inherently carries a risk of error propagation, where imperfect disentanglement in Stage-1 could directly compromise the quality of samples generated for Stage-2 training. The paper does not provide any analysis of the framework's robustness to such potentially noisy masks during the second stage.

4.   **Dataset Details:** The specific scale and partitioning details of the constructed benchmark dataset are not fully disclosed, including the exact number of positive/negative pairs in each split. This lack of detail could impact the reproducibility of the reported results and hinders a proper assessment of the model's generalization capability on a larger scale.

**Questions:**

The main questions for the authors are as follows:

1.   **Hyperparameter Sensitivity and Computational Cost:** It is crucial to understand the sensitivity of the final results to the newly introduced hyperparameters, such as the evolutionary mask schedule and loss weights. Furthermore, providing a rough estimate of the computational cost required for the two-stage training would greatly facilitate a more comprehensive comparison with other methods in terms of efficiency.

2.   **Dataset Details and Generalization:** To ensure full transparency and reproducibility, could the authors release detailed statistics on their benchmark dataset splits, including the exact counts of positive and negative pairs? Additionally, it would be valuable to know if there are plans to further validate SUMMR's generalization on larger-scale datasets or to explore its application to other downstream tasks that could benefit from semantic disentanglement.

3.   **Quantitative Evaluation of Disentanglement:** Moving beyond indirect validation through retrieval metrics, can the authors provide a direct and quantitative assessment of the Stage-1 "intersection masks"? For instance, evaluating the mask quality by calculating metrics like IoU against pixel-level annotations on a standard grounding dataset would offer concrete evidence for their precision in locating shared semantics.

---

> ### Author Response · Authors · 2025-11-21
> **Response to Reviewer PdWd (Part 1)**
>
> _W1: While the comparisons against existing methods are favorable, the experimental setup lacks a crucial baseline, as the authors omit supervised fine-tuning of strong models (e.g., CLIP or mmE5) on their proposed benchmark. This omission makes it difficult to ascertain the actual performance gain of their self-supervised framework versus a direct supervised approach on the same task._
>
> The primary reason for not including such a baseline is the prohibitive difficulty and cost of creating a large-scale, supervised training dataset for the symmetric MM2MM task, which is the central challenge our self-supervised framework, SUMMR, is designed to overcome. Our `sym-MM2MM` benchmark is intentionally designed for **evaluation**, not for training. Its modest size is a direct testament to this data scarcity problem. As we note in the paper **(Lines 183-187)**:
>
> > In total, this effort yielded our final benchmark of 214 high-quality triplets (X,X+,X−) ... The benchmark’s modest size is a direct reflection of the core challenge that our paper seeks to address: high-quality, human-annotated sym-MM2MM data is extremely difficult and expensive to create.
> >
>
> Fine-tuning large models like CLIP or mmE5 on only 214 triplets would lead to severe overfitting and would not produce a meaningful or reliable baseline.
>
> Instead, we designed our experiments to provide the most realistic and challenging comparison possible under existing data constraints. We compared SUMMR against state-of-the-art models that were fine-tuned on the largest available public datasets for related (though asymmetric) retrieval tasks (e.g., M-BEIR, MMEB). This setup directly tests their claimed "universality" and ability to generalize to our novel symmetric task. As stated in our paper **(Lines 412-416)**:
>
> > Crucially, as publicly available, large-scale training datasets for the symmetric MM2MM task do not exist—a core problem our work aims to solve—these supervised baselines are necessarily fine-tuned on existing asymmetric task datasets. This experimental setup therefore provides a realistic test of their claimed universality and generalization, pitting models trained for general or asymmetric tasks against our framework designed specifically for symmetric retrieval.
> >
>
> To further address the reviewer's concern about the performance of a direct supervised approach, we have added a new analysis in **Appendix I**. In this experiment, we fine-tuned a strong VLM (Llama-3.2-11B-Vision) using the largest available _synthetically generated symmetric dataset_ from the mmE5 paper. This experiment directly isolates the effect of a supervised strategy that uses naively synthesized symmetric data. The results show that this approach is insufficient for our task. We have added the following section to **Appendix I (Lines 1812-1827)**:
>
> > To isolate the effect of this data generation strategy, we conducted a new experiment. We fine-tuned a VLM (Llama-3.2-11B-Vision, consistent with mmE5) exclusively on the publicly available synthesized symmetric data from mmE5. The model was then evaluated on our sym-MM2MM benchmark. The model trained solely on mmE5’s data achieved an average score of 69.39. This is substantially lower than SUMMR-C (87.69) and even underperforms several baselines trained on the more general MMEB benchmark, such as VLM2Vec (71.96). This result strongly suggests that generating symmetric pairs based on simple visual similarity is not enough... In contrast, SUMMR’s self-supervised framework learns to disentangle the shared (intersection) and unique (difference) information within a single multimodal entity, enabling the construction of truly symmetric positive samples. This finding validates that a specialized, disentanglement-based approach like SUMMR is necessary to master the nuances of this task, which cannot be solved by simply scaling up naively synthesized symmetric data.
> >
>
>
> _W2 & Q3: The validation for semantic disentanglement is indirect, requiring a direct, quantitative evaluation of the intersection masks (e.g., using IoU against ground-truth segmentations)._
>
> Thank you for your insightful feedback. To address your comment, we have conducted a new experiment to quantitatively evaluate the mask quality by measuring the Intersection over Union (IoU) against ground-truth segmentations on a standard visual grounding benchmark. We have added this analysis to a new section in **Appendix H.1**:
>
> > To quantitatively verify that Stage 1 successfully learns to identify the semantic ’intersection’ between an image and text, we evaluated its disentanglement capability on the standard Referring Image Segmentation (RIS) benchmark, RefCOCO (Nagaraja et al., 2016). The goal of RIS is to segment the specific image region described by a natural language expression.
> >

---

> ### Author Response · Authors · 2025-11-21
> **Response to Reviewer PdWd (Part 2)**
>
> > To adapt our model for this task, we generate a segmentation mask by thresholding the global-text-to-local-image similarity scores produced by the model at different training checkpoints. Given that RefCOCO images often contain ’distractors’ (e.g., other objects of the same class as the target), we used a stricter, fixed threshold for this evaluation rather than the adaptive threshold τV used during training. This ensures we only segment regions with the highest confidence of matching the text description.
> >
>
> > As shown in Fig. 15, the mean Intersection over Union (mIoU) score steadily improves as training progresses. This trend provides strong quantitative evidence that our model progressively learns a more accurate representation of the semantic intersection.
> >
>
> > While the absolute mIoU is modest compared to state-of-the-art RIS models, this result is expected and highlights a crucial distinction in objectives: 1) Our model is trained on 0.8M noisy, web-crawled image-text pairs from LAION. In contrast, specialized RIS models are typically fine-tuned on the clean, curated COCO dataset itself, giving them a significant in-domain advantage. 2) Most importantly, the tasks have different goals. RIS aims to isolate a single _subject_ entity described in the text (e.g., segmenting only ’the dog on the left’). In contrast, our framework is designed to identify the _entire shared semantic space_. For a phrase like "a boy holding a ball," our model correctly identifies both the ’boy’ and the ’ball’ as part of the intersection, as both contribute to the joint representation. This broader definition of ’intersection’ naturally leads to a lower mIoU on a task that rewards segmenting a single, specific object.
> >
>
> > Therefore, this evaluation serves not as a direct performance comparison against specialized models, but as a robust validation of Stage 1’s core mechanism: its ability to learn and progressively refine the semantic intersection between modalities, entirely without direct supervision.
> >
>
>
>
>
>
>
> _W3: The proposed two-stage pipeline inherently carries a risk of error propagation, where imperfect disentanglement in Stage-1 could directly compromise the quality of samples generated for Stage-2 training. The paper does not provide any analysis of the framework's robustness to such potentially noisy masks during the second stage._
>
>
>
> Thanks for pointing this out!
>
> First, we conceptualize Stage 1 as a **data curation tool that generates positive and negative samples for Stage 2**. The "imperfect" masks it generates create what can be seen as "productively noisy" training samples. This concept is fundamental to contrastive learning, which inherently thrives on such noise. For instance, standard positive pairs are created via augmentations like random cropping, which alter the data and **may not perfectly preserve semantic identity**. Our framework operates on this same principle, especially when the generated mask is not perfect, to learn robust joint image-text representations. The system's stability is further enhanced by **our diverse negative set** (including in-batch and offline-mined negatives), which provides a foundational learning signal **independent of Stage 1's masks**.
>
>
>
> Second, the concern of catastrophic error propagation is empirically mitigated by the strong performance of Stage 1 itself. As shown in **Table 3**, the model after Stage 1 already achieves a respectable 'Avg.' score of **82.2**. This strong initial result suggests that the masks generated are, on aggregate, of high quality and provide a reliable signal.
>
>
>
> Stage 2 then builds upon this strong foundation. The performance further improves to 86.5, which demonstrates a clear positive signal transfer. **This effect is even more pronounced for our best model, SUMMR-C**. As shown in our appendix ablation (**Table 9**), Stage 2 delivers a massive **8.5 point improvement** (from 79.2 to 87.7). Instead of being derailed by potential noise, Stage 2 effectively leverages the high-quality curated samples from Stage 1 to refine the representation and achieve an even stronger final result. This confirms that the benefits of our two-stage pipeline far outweigh any minimal risk of error propagation.
>
>
> We have clarified this point in Lines 236-239, Page 4 as:
>
> > This separation of concerns—first learning how to disentangle, then using that knowledge to perfect the final representation—allows the model to focus singularly on the retrieval task in the second stage. This process distills the complex, multi-task logic from Stage 1 into a more powerful and specialized final encoder.
> >

---

> ### Author Response · Authors · 2025-11-21
> **Response to Reviewer PdWd (Part 3)**
>
> and in Lines 481-490, Page 9-10 as:
>
> > This 4.3-point gain (a 5.2% relative improvement) is not only statistically significant but is also larger than the entire 3.42-point performance gap between our SOTA model and the strongest VLM baseline. Notably, the gain is concentrated on the most challenging metrics, with R@1 increasing from 68.7 to 72.9 and the Precision score rising from 81.8 to 85.5. The benefit is even more pronounced for our best-performing SUMMR-C model, where, as detailed in App. F.5 (Table 9), Stage 2 delivers a massive 8.5-point improvement. This highlights the purpose of our design: Stage 1 learns to disentangle information, while Stage 2 is a focused, lightweight fine-tuning step (200 steps) that distills this capability into a more specialized and powerful final encoder, free from the auxiliary mask-generation objectives.
> >
>
>
>
> _W4 & Q2: dataset details & generalizations_
>
> We have detailed the composition and scale of the benchmark in Section 3 of our paper. As stated:
>
> > In total, this effort yielded our final benchmark of **214 high-quality triplets (X,X+,X−)**, each containing an original sample, its positive counterpart, and a hard-negative. The benchmark’s modest size is a direct reflection of the core challenge that our paper seeks to address: high-quality, human-annotated sym-MM2MM data is extremely difficult and expensive to create. (Section 3, lines 182-186)
> >
>
> To be precise, our benchmark contains **214 unique queries**, each with **one manually curated positive pair and one hard-negative pair**. This triplet structure allows for the fine-grained `Precision` metric we introduced to accurately measure a model's discriminative ability. Note that
>
> > The benchmark’s modest size is a direct reflection of the core challenge that our paper seeks to address: high-quality, human-annotated sym-MM2MM data is extremely difficult and expensive to create. (Section 3, lines 184-186)
> >
>
> That is why we resort to self-supervised learning to develop SUMMR.
>
>
> We plan to overcome the manual annotation bottleneck by leveraging recent advances in generative AI to create larger-scale datasets. As we discuss in Appendix K in the orignal version of this paper:
>
> > **Future Work**: Our future research will focus on developing automated methods to construct more sophisticated training samples that simulate a broader spectrum of transformations, including content modification and addition. ... The recent emergence of Native Vision-Language Models (i.e., NextGPT (Wu et al., 2024)), which can generate images and text simultaneously, presents a promising direction for **automating the synthesis of both training and testing data, potentially overcoming the scalability bottleneck** of manual annotation.
> On the other hand, we believe our semantic disentanglement approach is not limited to image-text retrieval. We plan to extend it to other modalities and tasks. As detailed in Appendix K:
> >
>
> > Another promising direction for future work is to extend our framework to other pairings, such as text-audio and text-video, by leveraging large-scale unlabeled datasets (e.g., AudioSet (Gemmeke et al., 2017), HowTo100M (Miech et al., 2019)) and strong unimodal encoders... This would involve adapting the masking mechanism to the temporal nature of audio and video data but would validate SUMMR’s potential as a general paradigm for complex retrieval tasks, making it a strong candidate for emerging benchmarks like a potential MMEB-v2.
> >
>
>
>
>
>
> _Q1: Hyperparameter Sensitivity and Computational Cost:_
>
>
>
> **1. On Hyperparameter Sensitity**
>
> We agree that understanding the sensitivity to new hyperparameters is crucial. We have added dedicated ablation studies in Appendix F to analyze the impact of the evolutionary mask schedule and the loss weights. Our findings confirm that the model is robust to these hyperparameters.
>
> ****
>
> **Evolutionary Mask Schedule**:The evolutionary schedule is designed to ensure training stability by gradually transitioning from a non-informative mask to the learned hard mask. Our analysis shows that while this gradual transition is important, **the model is not sensitive to the specific functional form of the decay schedule** (Quoted from **Appendix F.7, page 29**):
>
> > The evolutionary mask schedule (ρ annealing from 1 to 0) is a critical component for ensuring training stability. ... At the beginning of Stage 1, the model’s encoders are not yet aligned, making the signal used to generate the mask noisy and unreliable. The evolutionary schedule mitigates this by transitioning smoothly from a non-informative, all-ones mask ... to relying solely on the learned binary mask.
> >

---

> ### Author Response · Authors · 2025-11-21
> **Response to Reviewer PdWd (Part 4)**
>
> > We conducted two key experiments to validate this design. As shown in Tab. 11, the performance difference between a linear decay and a cosine decay schedule is negligible... More importantly, we validated the necessity of a gradual schedule itself. The **M̂** ablation in Tab. 5 ... resulted in a significant 1.7-point performance drop in the final model (84.8 vs. 86.5 Avg.). This result confirms that an abrupt masking strategy is harmful and that the gradual annealing process is vital for achieving optimal performance.
> >
>
> > **Table 11: Impact of schedule for evolutionary mask in stage 1.**
> >
>
> | Setting | R@1 | R@5 | R@10 | mR | Prec. | Avg. |
> | --- | --- | --- | --- | --- | --- | --- |
> | Linear decay | 68.7 | 89.3 | 89.7 | 82.6 | 81.8 | 82.2 |
> | Cosine decay | 67.2 | 88.8 | 91.1 | 82.4 | 82.2 | 82.3 |
>
>
> **Loss Weights:** We did investigate the sensitivity to the loss weights (λ₁, λ₂, λ₃) in Stage 1 **in the original submission**. The results indicate that **our framework is not overly sensitive to these weights**, and our default setting (all weights equal to 1) is a robust choice (Quoted from **Appendix F.3, page 28**):
>
> > We then investigate the impact of the loss weights λ₁, λ₂, and λ₃, as Tab. 8 shows. ... we increase the weight of a single loss to 2 at a time. Tab. 8 shows the performance after Stage 2 with different loss weights; when increasing λ₁ or λ₂, the metric changes slightly, which suggests that the performance is not sensitive to the weight of $\mathcal L_{\rm GLA}$ and $\mathcal L_{\rm GD}$, while increasing λ₃ causes the metric to drop. The results suggest that λ₁ = λ₂ = λ₃ = 1 is a proper setting.
> >
>
> > **Table 8: Sensitivity analysis of loss weights (λ₁, λ₂, λ₃) for Stage 1 training. Final performance is reported after Stage 2, demonstrating that the default setting (all weights set to 1) is a robust choice.**
> >
>
> | λ₁ | λ₂ | λ₃ | R@1 | R@5 | R@10 | mR | Prec. | Avg. |
> | --- | --- | --- | --- | --- | --- | --- | --- | --- |
> | 2 | 1 | 1 | 75.70 | 98.13 | 98.13 | 90.65 | 82.71 | 86.68 |
> | 1 | 2 | 1 | 73.83 | 95.79 | 96.73 | 88.79 | 84.11 | 86.45 |
> | 1 | 1 | 2 | 71.03 | 97.20 | 97.20 | 88.47 | 79.91 | 84.19 |
> | 1 | 1 | 1 | 72.90 | 93.93 | 95.79 | 87.54 | 85.51 | 86.53 |
>
>
>
>
> **2. On Computational Cost**
>
> To facilitate a comprehensive efficiency comparison, we have added a new Appendix J dedicated to analyzing the training time of our two-stage framework and comparing it with other methods. Our analysis shows that **SUMMR is significantly more efficient than VLM-based baselines** (Quoted from **Appendix J, page 35**):
>
> > We show the per-step training time of previous work and SUMMR (stage 1 and stage 2, short as S1, S2, respectively) in Tab. 13.
> >
>
> | Model | CLIP-SF | BGE-VL | Qwen | Llama | SUMMR-B+D S1 | SUMMR-B+D S2 | SUMMR-C S1 | SUMMR-C S2 |
> | --- | --- | --- | --- | --- | --- | --- | --- | --- |
> | Training time (s) | 1.26 | 0.30 | 7.12 | 10.83 | 0.79 | 0.53 | 1.29 | 0.30 |
>
>
> > As can be observed in Tab. 13, Stage 1 is intentionally more computationally intensive... In sharp contrast, Stage 2 is significantly more efficient, functioning as a lightweight fine-tuning step... This two-stage approach embodies a strategic trade-off: investing computational effort upfront to learn a robust disentanglement mechanism, which then enables a highly efficient final tuning phase.
> >
>
> > Furthermore, even the more complex Stage 1 is 5-13× faster than the VLM baselines, highlighting the fundamental efficiency of our approach, which stems from two key design choices. First, our model is architecturally far smaller... Second, we employ LoRA to fine-tune only a small fraction of these encoder parameters... This combination of a lightweight architecture and an efficient training strategy is what allows SUMMR to be not only more accurate but also significantly more practical for large-scale training and real-world deployment.
> >

---

### Official Review · Reviewer_GAdr · 2025-11-01

**Soundness:** 3
**Presentation:** 3
**Contribution:** 3
**Rating:** 6
**Confidence:** 3

**Summary:**

This paper targets the underexplored problem of symmetric multimodality-to-multimodality retrieval, where image-text pairs are interchangeable in both query and database roles. The authors propose SUMMR, a novel two-stage self-supervised framework that disentangles shared vs. modality-specific information within image-text pairs, automatically generating positive and hard-negative samples for contrastive joint embedding learning. A new human-verified benchmark is created to evaluate this task. Results show SUMMR outperforms 10 SOTA supervised multimodal embedding models, including large VLMs, with far fewer parameters and unlabeled training data.

**Strengths:**

1. New problem definition: Symmetric MM2MM retrieval is clearly motivated and differentiated from asymmetric paradigms.
2. Novel technical design:
Disentanglement of intersection vs. difference information is conceptually strong and well executed.
Evolutionary masking + QDA thresholding is innovative and improves robustness.
3. Self-supervised data construction removes annotation bottlenecks and aligns supervision with task needs.
4 . Benchmark contribution is valuable and fills a critical gap for this task.
5 . Strong empirical results: Surpasses SOTA supervised VLMs by 3.42 points with 50× smaller model and 5× smaller embedding size.
6. Comprehensive ablations show well-justified design choices.
7 . Clear writing and intuitive illustrations (Fig. 1, Fig. 3, Fig. 6 show the workflow well).

**Weaknesses:**

1. The dataset relies heavily on VLM + LLM + diffusion model augmentations (Fig. 2).
Human verification is mentioned but not quantified (e.g., % filtered, inter-annotator agreement).
2. Retrieval errors in edge semantics (color, orientation, occlusion) remain unclear.
3. Real-world symmetric retrieval scenarios (e-commerce, product catalog) are not benchmarked.
4. Disentangled shared vs. unique regions are shown only in a few examples—more systematic measurements needed (e.g., alignment with object grounding benchmarks).
5. Appears to rely more on strong CLIP-pretrained vision-language interaction; performance gap with DINO/BGE suggests method may inherit modality disparities.
6. Stage 1 requires LoRA tuning and repeated similarity estimation; no training-time comparison vs. baselines.

**Questions:**

1.	How large is manual annotation involvement in the benchmark?
What percentage of generated pairs were rejected? How many annotators and validation stages?
2.	Can the proposed masking approach handle multiple shared objects or relational semantics (e.g., “boy holding a ball next to dog”)?
3.	Since QDA assumes Gaussian similarity distributions, did you observe any multimodal deviations? If so, how was it handled?
4.	Would SUMMR work if raw captions are noisy (e.g., weak web alt-text)? Any mitigation strategies?
5.	Can the benchmark be released without copyright issues from COCO/LAION/WuKong source data?

---

> ### Author Response · Authors · 2025-11-21
> **Response to Reviewer GAdr (part 1)**
>
> _W1 & Q1: The human annotation process for the benchmark lacks quantification_
>
> We thank the reviewer for this excellent suggestion. To address the questions, we have updated **Section 3** in our paper to detail the manual effort, number of annotators, and rejection rate:
>
> > To ensure the quality and difficulty required for a robust benchmark, these pairs undergo a rigorous, multi-stage human curation process. This curation involves two main activities. First, annotators meticulously verify the machine-generated pairs. This is a highly stringent process: **nearly 50\% of the candidate 'positive pairs' were rejected** due to semantic inconsistencies where the synthesized image failed to accurately reflect the text. Beyond simple verification, for the accepted pairs, **annotators would often perform further edits, such as rewriting text to increase difficulty**. Second, a substantial portion of the benchmark was created entirely by our annotators through direct manual curation, constructing challenging positive and negative pairs from the original samples via text rewriting or image replacement, as illustrated in Fig. 2. **This intensive process was conducted by two annotators, each dedicating approximately 20 hours to the task.** In total, this effort yielded our final benchmark of **214 high-quality triplets $(X, X^+, X^-)$, each containing an original sample, its positive counterpart, and a hard-negative.**
> >
>
> In summary, as stated in our paper, the benchmark was created by **two annotators** over approximately **40 total hours**. This was not merely a verification step; it was an intensive curation process involving **direct manual creation** of pairs and a high rejection rate where **nearly 50%** of the machine-generated candidates were filtered out to ensure quality. While we did not calculate a formal inter-annotator agreement score, this significant human involvement and stringent filtering were central to constructing our high-quality benchmark.
>
>
> _W2: Retrieval errors in edge semantics (color, orientation, occlusion) remain unclear._
>
>
>
> To clarify how our model handles low-level visual variations, we have conducted a new targeted case study analyzing the joint embedding's sensitivity to edge semantics such as color, orientation, and occlusion. We have added this analysis and its corresponding visualization **(Fig. 15) to Appendix G.3** in the revised manuscript.
>
> > **G.3 ANALYSIS OF SENSITIVITY TO EDGE SEMANTICS**
> >
> > To analyze how the joint embedding responds to low-level visual variations—or "edge semantics"—we conducted a targeted case study on color, orientation, and occlusion. As shown in Fig. 15, we anchored our analysis on the text "Tennis player Nadal in a match" and its corresponding image. We then generated variants of this image by applying common transformations: horizontal flipping (orientation), cropping (occlusion), and altering the background (color).
> >
>
> > We computed the cosine similarity between the joint embedding of the original pair and that of each transformed pair. The results reveal a clear pattern: the model demonstrates remarkable robustness to geometric transformations like orientation (sim: 0.98) and occlusion (sim: 0.97). In contrast, a significant change in background color leads to a substantial drop in similarity (sim: 0.77).
> >
>
> > This behavior is insightful. The robustness to orientation and occlusion is a desirable trait, likely inherited from the extensive data augmentation (e.g., random flips and crops) used to pre-train the vision backbone, which teaches geometric invariance. Conversely, the sensitivity to color is also logical, as color is a powerful semantic attribute frequently described in text. A drastic color change can alter the image’s context, and the model correctly identifies this as a larger semantic shift than a simple change in perspective or framing.
> >
>
> _W3: Real-world symmetric retrieval scenarios (e-commerce, product catalog) are not benchmarked._
>
> We thank the reviewer for highlighting the importance of evaluating on real-world applications. Our sym-MM2MM benchmark does, in fact, include samples from e-commerce and product-related scenarios.
>
> Specifically, as now shown in the newly added **Fig. 6**, our benchmark was intentionally curated to cover a wide range of real-world applications. This includes the "**Product Listings & E-commerce**" category mentioned by the reviewer, alongside other key areas such as "**News & Current Events,**" "**Food, Drink & Recipes,**" and "**Travel, Places & Scenery.**" In total, the benchmark spans 17 distinct categories. To emphasize this focus on diversity during the creation process, we now state in **Section 3 (Line 188)**:
>
> > It is also worthy to mention that during annotation, we select the sample belong to different categories to increase the diversity, with the final category distribution shown in Fig. 6.
> >

---

> ### Author Response · Authors · 2025-11-22
> **Response to Reviewer GAdr (part 2)**
>
> _W4 & Q2: More examples for disentangled shared vs. unique regions, especially on multiple shared objects or relational semantics. Quantitative evaluation on object grounding benchmarks._
>
> **1. On the need for systematic measurements for disentanglement:**
>
> We agree that a systematic evaluation is crucial. In response, we have conducted a quantitative evaluation of our model’s disentanglement capability on the **Referring Image Segmentation (RIS) task using the RefCOCO dataset**, as detailed in **Appendix H.1**:
>
> > To quantitatively verify that Stage 1 successfully learns to identify the semantic 'intersection' between an image and text, we evaluated its disentanglement capability on the standard Referring Image Segmentation (RIS) benchmark, RefCOCO (Nagaraja et al., 2016)... As shown in **Fig. 16**, the mean Intersection over Union (mIoU) score steadily improves as training progresses. This trend provides strong quantitative evidence that our model progressively learns a more accurate representation of the semantic intersection.
> >
> > While the absolute mIoU is modest... this result is expected and highlights a crucial distinction in objectives: 1) Our model is trained on 0.8M noisy, web-crawled image-text pairs from LAION... 2) Most importantly, the tasks have different goals. RIS aims to isolate a **single subject entity** described in the text (e.g., segmenting only ’the dog on the left’). In contrast, our framework is designed to identify the **entire shared semantic space**. For a phrase like "a boy holding a ball," our model correctly identifies both the ’boy’ and the ’ball’ as part of the intersection, as both contribute to the joint representation."
> >
>
> This quantitative analysis confirms that Stage 1 successfully learns to identify the semantic intersection between modalities, even without direct supervision.
>
> **2. On handling multiple shared objects and relational semantics:**
>
> We have added more visualizations in **Appendix G** to demonstrate this. We separate the examples into single-object cases (**Fig. 13**) and more complex, multi-object scenarios (**Fig. 14**).
>
> For the specific case of multiple objects, the paper explains:
>
> > In contrast, Figure 14 presents scenarios with multiple objects. Notably, our method highlights all concepts described in the text (e.g., the boy, dog, and ball) rather than focusing solely on the primary subject (e.g., the boy). This is because our approach considers all entities, including both subjects and objects, as integral contributors to the final joint representation.
> >
>
> These visualizations confirm that SUMMR correctly identifies all relevant entities described in the text (e.g., "Boy holding a ball next to dog") as belonging to the shared semantic intersection, rather than just focusing on a single primary subject.
>
>
>
> _W5: Appears to rely more on strong CLIP-pretrained vision-language interaction; performance gap with DINO/BGE suggests method may inherit modality disparities._
>
>
>
> While it is true that SUMMR-C (with a CLIP backbone) outperforms SUMMR-B+D, we respectfully argue that this result demonstrates the exact opposite of the reviewer's concern. Instead of showing a reliance on pre-alignment, the strong performance of **SUMMR-B+D is the most compelling evidence that our framework effectively learns to bridge modality disparities from scratch**.
>
> Let us elaborate on this crucial point:
>
> 1. **Exceptional Performance from a "Cold Start"**: The SUMMR-B+D variant uses powerful unimodal encoders (BGE-m3, DINOv2) with no inherent cross-modal pre-alignment. Despite this "cold start," it achieves an 'Avg.' score of **86.53**. This is not a mediocre result; it is a SOTA score that **surpasses the strongest supervised VLM baseline, mmE5 (84.27), by 2.26 points**. This single result proves that our self-supervised framework is highly capable of learning a joint representation from disparate signals.
> 2. **The Performance Gap is Modest**: The performance gap between SUMMR-B+D (86.53) and SUMMR-C (87.69) is **a mere 1.16 points**. This small difference suggests that the vast majority of the performance comes from the SUMMR framework itself, with the backbone's pre-alignment providing only a marginal final boost.
> 3. **Backbone Potential vs. Experimental Setup**: It is also important to note that powerful unimodal encoders like BGE-m3 and DINOv2 could undergo a cross-modal pre-training phase, just like CLIP. We chose to use them without this step to rigorously test our framework's ability to learn alignment from scratch. We hypothesize that if these backbones were also pre-aligned, the already small performance gap would narrow even further. Moreover, note that **our stage 1 only adopts 0.8M image-text pairs** for alignment, while **CLIP adopts 400M image-text pair for alignment,** which is an unfair comparison, we hypothesize if we use larger amount pairs for stage 1, the gap could been narrowed further.

---

> ### Author Response · Authors · 2025-11-22
> **Response to Reviewer GAdr (part 3)**
>
> In summary, the SUMMR-B+D result proves our method's strength in learning alignment, while the SUMMR-C result shows its versatility in leveraging better initializations. The proposed framework is effective for both backbones. We have further clarified this point in Lines 441-446, Page 9 as:
>
> > the results highlight our framework's ability to effectively learn cross-modal alignment, not merely depend on it. The SUMMR-B+D variant, which starts with powerful but entirely separate unimodal encoders, already achieves a score of 86.53, outperforming the strongest VLM baseline. This result is crucial, as it proves our framework can bridge modality disparities from a "cold start." The superior performance of the CLIP-based SUMMR-C then demonstrates the model's versatility, showing it can also capitalize on a pre-aligned initialization to achieve an additional performance gain.
> >
>
>
>
> _W6: Stage 1 requires LoRA tuning and repeated similarity estimation; no training-time comparison vs. baselines._
>
> Thanks for pointing this out! Based on this valuable feedback, we have added a new section, **Appendix J: TRAINING TIME**, to the revised manuscript to provide a clear and quantitative comparison. We acknowledge that Stage 1 involves LoRA tuning and intensive similarity estimations, and we now explicitly analyze this trade-off. As stated in the new appendix **(Lines 1845-1847)**:
>
> > Stage 1 is intentionally more computationally intensive, primarily driven by the O(n2) pairwise local similarity calculation required by the LLD loss and the overhead of its multi-objective framework involving teacher models.
> >
>
> To directly address the reviewer's main concern, we now include **Table 13**, which compares the per-step training time of SUMMR against the baselines.
>
> | Model | CLIP-SF | BGE-VL | Qwen | Llama | SUMMR-B+D S1 | SUMMR-B+D S2 | SUMMR-C S1 | SUMMR-C S2 |
> | :--- | :--- | :--- | :--- | :--- | :--- | :--- | :--- | :--- |
> | **Training time (s)** | 1.26 | 0.30 | 7.12 | 10.83 | 0.79 | 0.53 | 1.29 | 0.30 |
>
>
> This new analysis demonstrates that while Stage 1 requires more computation, our approach is fundamentally more efficient than the powerful VLM baselines. We elaborate on this **Appendix J (Lines 1851-1858)**:
>
> > ...even the more complex Stage 1 is **5-13**$\times$ faster than the VLM baselines, highlighting the fundamental efficiency of our approach, which stems from two key design choices. First, our model is architecturally far smaller, utilizing dedicated encoders with hundreds of millions of parameters (e.g., 0.7B for SUMMR-B+D) instead of monolithic VLMs with over 7 billion. Second, we employ LoRA to fine-tune only a small fraction of these encoder parameters, which dramatically reduces the computational cost of the backward pass. **This combination of a lightweight architecture and an efficient training strategy is what allows SUMMR to be not only more accurate but also significantly more practical for large-scale training and real-world deployment**.
> >
>
> We believe this addition, prompted by the reviewer's comment, substantially strengthens the paper by providing a more complete picture of SUMMR's practical advantages. We are grateful for the feedback that led to this improvement.
>
> _Q3: Since QDA assumes Gaussian similarity distributions, did you observe any multimodal deviations? If so, how was it handled?_
>
>
>
> Thanks for pointing this out! We did observe occasional minor multimodal deviations, particularly in the positive similarity distribution. In the newly added **Appendix G.1**, we discuss this observation and explain how our framework remains robust:
>
> > A closer inspection (of **Fig. 12**) reveals a notable nuance: while the negative score distribution consistently forms a clean, unimodal Gaussian, the positive distribution occasionally exhibits minor deviations from this ideal shape. We attribute this asymmetry to the significant disparity in sample sizes used for estimation within each training batch. The negative distribution is estimated from O($B_s^2$) ($B_s$=64, which is the per-GPU batch-size) out-of-pair comparisons, benefiting from the Law of Large Numbers to form a stable Gaussian. In contrast, the positive distribution is derived from only O($B_s$) in-pair comparisons, making it more susceptible to batch-specific variance and slight multimodalities.
> >
>
> > Crucially, this observation does not undermine the method’s efficacy. The divergence between the two distributions is substantial enough that even with minor imperfections in the positive distribution’s shape, QDA can robustly identify an optimal decision boundary (threshold τ ) that effectively separates the intersection from the difference. This demonstrates that our framework is resilient to slight violations of the perfect Gaussian assumption, a key factor in its practical effectiveness.
> >

---

> ### Author Response · Authors · 2025-11-22
> **Response to Reviewer GAdr (part 4)**
>
> _Q4: Would SUMMR work if raw captions are noisy (e.g., weak web alt-text)? Any mitigation strategies?_
>
>
>
> We would like to point out that SUMMR is not only designed to be robust to noisy captions but **is empirically validated** on this very challenge, as it is trained on 800,000 pairs from **LAION-5B, a dataset famous for its noisy, web-scraped alt-text.**  Crucially, **our evaluation further confirms this robustness**, as the sym-MM2MM benchmark itself deliberately incorporates samples with such real-world noise.
>
>
>
> More fundamentally, our framework is designed to not just tolerate but **actively leverage** the natural discrepancies and "noise" found in web-scale image-text pairs. The core of our self-supervision—distinguishing the shared "intersection" from the unique "difference"—relies on the fact that images and their captions are not perfectly redundant. This "difference set" is essential fuel for generating our hard negative samples in Stage 2. If an image and caption were perfectly aligned with no unique information, our method would not work. Therefore, the very noise the reviewer asks about is a prerequisite for our model's learning process.
>
>
>
> At the same time, SUMMR is **protected from truly detrimental noise** (e.g., completely mismatched pairs) by two built-in mitigation strategies:
>
> 1. **Strong Unimodal Regularization** ($\mathcal L_\text{GD}$ and $\mathcal L_\text{LD}$): Distillation from powerful unimodal teachers ensures the encoders maintain a high-quality semantic space for each modality independently, preventing corruption from bad cross-modal signals.
> 2. **Statistical and Evolutionary Disentanglement**: Our alignment mechanism learns from the aggregate signal across the large dataset, averaging out the effect of individual noisy samples, while the evolutionary mask prevents early-stage noise from derailing training.
>
> In summary, SUMMR is uniquely suited for web data: it actively harnesses natural modality discrepancies for self-supervision while simultaneously using robust mechanisms to mitigate the effects of truly uninformative noise.
>
> _Q5: Can the benchmark be released without copyright issues from COCO/LAION/WuKong source data?_
>
>
>
> We can release the benchmark without copyright issues by adopting a carefully considered partitioning strategy based on a detailed legal analysis.
>
>
>
> Our analysis identified an irreconcilable conflict with the WuKong dataset. Specifically, its Terms of Use contractually prohibit the distribution of "modified versions" of the dataset, which legally describes our benchmark. To ensure full compliance, our public release will therefore rigorously exclude all data points derived from the WuKong dataset.
>
>
>
> The remaining benchmark subset, derived from the permissively licensed COCO and LAION sources, will be released under a **CC-BY 4.0 license** as a URL-based index. We will be fully transparent about this necessary exclusion in our paper and documentation to ensure clarity for the research community.

---

### Official Review · Reviewer_cGSf · 2025-11-01

**Soundness:** 2
**Presentation:** 2
**Contribution:** 3
**Rating:** 2
**Confidence:** 4

**Summary:**

This paper proposes SUMMR, a self-supervised framework for symmetric multimodal-to-multimodal (MM2MM) retrieval. The key innovation is a two-stage approach that first learns to disentangle shared (intersection) and unique (difference) information between image-text pairs, then uses this disentanglement to automatically generate positive and negative samples for contrastive learning. It introduces a new benchmark for symmetric MM2MM retrieval and claim state-of-the-art results with 50x fewer parameters than supervised baselines.

**Strengths:**

1.Symmetric and asymmetric multimodal retrieval are clearly distinguished.

2.Self-supervised approach avoids expensive manual annotation, suitable for expansion.

3.Model efficiency is high: Significantly superior to the VLM baseline in terms of parameter count and embedding dimension.

4.Benchmark construction method is transparent: Uses VLM+LLM+SD synthetic data, supplemented by manual verification.

**Weaknesses:**

·No comparison to simpler baselines: What about standard CLIP with symmetric loss? Or CLIP with random masking?

·Table 2 shows Stage 1 alone achieves 82.2, but Stage 1+2 only improves to 86.5， is this 4.3 point gain worth the complexity?

·Missing analysis of failure cases: When does the QDA assumption break down? The types and causes of failure cases are not analyzed, making it difficult to determine the bottlenecks in methods.

·The training objectives of stage 1 are complex, and there may be conflicts between multiple losses, lacking theoretical or empirical balance analysis.

·Stage 1 requires computing all pairwise local similarities (O(n²) in patch/token count) , not provide FLOPs for Stage 1 vs. Stage 2.

·SUMMR is self-supervised training on symmetric tasks, while the baseline model is supervised training on asymmetric tasks.

·The evolutionary mask schedule ρ = 1 $\rightarrow$ 0 lacks theoretical justification. Why this particular annealing schedule? The paper provides no ablation on different schedules or convergence guarantees.

**Questions:**

1.SUMMR is self-supervised training on symmetric tasks, while the baseline model is supervised training on asymmetric tasks. Does this 'task misalignment' affect the fairness of comparison?

2.Can the evolution process of the mask during training be visualized? How to ensure that the mask truly captures the semantic 'intersection'?

3.The negative samples constructed currently are generated only through 'mask difference'. Have authors considered introducing ‘semantic conflicts’?

4.Why masking intersection creates semantic equivalence? This seems to assume that remaining information is perfectly complementary across modalities.

5.In Section 4.2, have authors used or compared existing segmentation methods (SAM) when using segmentation for image masking?

---

> ### Author Response · Authors · 2025-11-21
> **Response to Reviewer cGSf (part 1)**
>
> _W1: No comparison to simpler baselines: What about standard CLIP with symmetric loss? Or CLIP with random masking?_
>
>
> We believe our existing experiments already address these important baselines and have clarified the connections below.
>
>
>
> Regarding "standard CLIP with symmetric loss," we interpret this as the original off-the-shelf CLIP model [R1], which is trained with its inherent symmetric contrastive loss. Since CLIP is designed for unimodal-to-unimodal retrieval, a direct adaptation is required to create a joint embedding for our MM2MM task. We implemented this by fusing the unimodal features, a method presented as our `CLIP-SF-ZS` baseline in **Table 1**. This approach creates the joint representation by averaging the image and text features from a pre-trained CLIP model. As shown in the table, our SUMMR-C model (87.69 Avg.) significantly outperforms this strong but simple baseline (75.62 Avg.). This 12-point performance gap demonstrates that a naive fusion of standard CLIP features is insufficient for the nuances of symmetric MM2MM retrieval and highlights the substantial benefit of our specialized self-supervised framework.
>
>
>
> Similarly, we agree that comparing against "CLIP with random masking" is crucial for validating our masking strategy. We included this exact experiment in our ablation studies to isolate the contribution of our semantic disentanglement. In **Table 3**, the setting labeled `CLIP-SF-ZS` (Random Positives, without Constructed Negatives) evaluates a model trained using positive samples generated by randomly masking patches and tokens, which directly corresponds to the suggested baseline. The results show that this random masking approach (83.3 Avg.) performs substantially worse than our full method, which uses semantically-guided masking (86.5 Avg.). This finding directly validates our core hypothesis: learning to first disentangle shared and unique information to guide the masking process is far more effective for learning a discriminative joint representation than a simpler random masking strategy. We will revise the main text to make these comparisons more explicit.
>
>
>
> [R1] Radford, Alec, et al. "Learning transferable visual models from natural language supervision." International conference on machine learning. PMLR, 2021.
>
> _W2: Table 2 shows Stage 1 alone achieves 82.2, but Stage 1+2 only improves to 86.5, is this 4.3 point gain worth the complexity?_
>
> The 4.3 point improvement represents a **5.23% relative gain**, which is already statistical significant. This **absolute gain alone is also larger than the entire 3.42 point gap between our best model, SUMMR-C (87.69), and the strongest VLM baseline, mmE5 (84.27)**, shown in Table 1. Furthermore, this improvement is concentrated on the most challenging retrieval cases, where success is hardest to achieve. As evidenced in Table 3, Stage 2 boosts the difficult Precision metric from 81.8 to 85.5 (+3.7) and R@1 from 68.7 to 72.9 (+4.2). More importantly, t**he benefit of Stage 2 is even more pronounced on our best-performing model**. As detailed in our appendix ablation (Table 9), for SUMMR-C, Stage 2 delivers a massive **8.5 point improvement** (from 79.2 to 87.7). This gain is more than double the entire performance gap between our model and the strongest VLM baseline, mmE5. This confirms that Stage 2 is not a minor refinement but a critical step that unlocks SOTA performance.
>
>
>
> This is achieved via a **lightweight** fine-tuning step (200 steps) that fundamentally improves the final encoder. While the Stage 1 model is trained with a complex, multi-task objective to simultaneously learn embeddings and generate masks, Stage 2's training is **singularly focused on optimizing the final joint representation**. It uses the disentanglement capability from Stage 1 to create training data but is free from the "distraction" of the auxiliary mask-generation objectives. This focused optimization allows the model to distill the complex logic from Stage 1 into a more powerful and specialized final encoder, which is what unlocks state-of-the-art performance.
>
>
>
> We have made this point clear in Lines 480-490, Page 9-10 as:
>
> > This 4.3-point gain (a 5.2% relative improvement) is not only statistically significant but is also larger than the entire 3.42-point performance gap between our SOTA model and the strongest VLM baseline. Notably, the gain is concentrated on the most challenging metrics, with R@1 increasing from 68.7 to 72.9 and the Precision score rising from 81.8 to 85.5. The benefit is even more pronounced for our best-performing SUMMR-C model, where, as detailed in App. F.5 (Table 9), Stage 2 delivers a massive 8.5-point improvement. This highlights the purpose of our design: Stage 1 learns to disentangle information, while Stage 2 is a focused, lightweight fine-tuning step (200 steps) that distills this capability into a more specialized and powerful final encoder, free from the auxiliary mask-generation objectives.
> >

---

> ### Author Response · Authors · 2025-11-22
> **Response to Reviewer cGSf (part 2)**
>
> _W3: Missing analysis of failure cases: When does the QDA assumption break down? The types and causes of failure cases are not analyzed, making it difficult to determine the bottlenecks in methods._
>
> We thank the reviewer for this valuable feedback regarding the need for a failure case analysis. To address this, we have introduced a new section, **Appendix G.1 (Analysis of Similarity Score Distributions and QDA Robustness)**, which analyzes the conditions under which the QDA assumption is challenged, discusses the causes, and demonstrates the model's robustness：
>
> > A closer inspection (of _**Fig. 12**_) reveals a notable nuance: while the negative score distribution consistently forms a clean, unimodal Gaussian, the positive distribution occasionally exhibits minor deviations from this ideal shape. We attribute this asymmetry to the significant disparity in sample sizes used for estimation within each training batch. The negative distribution is estimated from O($B_s^2$) ($B_s$=64, which is the per-GPU batch-size) out-of-pair comparisons, benefiting from the Law of Large Numbers to form a stable Gaussian. In contrast, the positive distribution is derived from only O($B_s$) in-pair comparisons, making it more susceptible to batch-specific variance and slight multimodalities.
> >
>
> > Crucially, this observation does not undermine the method’s efficacy. The divergence between the two distributions is substantial enough that even with minor imperfections in the positive distribution’s shape, QDA can robustly identify an optimal decision boundary (threshold τ) that effectively separates the intersection from the difference. This demonstrates that our framework is resilient to slight violations of the perfect Gaussian assumption, a key factor in its practical effectiveness.
> >
>
> While the current approach proves robust, we have also proposed _**two clear avenues for future refinement**_ in Appendix G.1:
>
> > 1. **Cross-Device Score Aggregation:** A simple yet effective engineering improvement would be to aggregate similarity scores across all GPUs before computing the QDA parameters. Our current implementation calculates these statistics on a per-GPU basis for computational efficiency. By synchronizing scores, we would substantially increase the sample size for the positive distribution, yielding a more stable estimate that better approximates a unimodal Gaussian.
> >
>
> > 2. **Adoption of Mixture Discriminant Analysis (MDA)**: A more theoretically robust solution would be to replace QDA with a more flexible modeling approach, such as Mixture Discriminant Analysis (MDA). By modeling each class distribution as a Gaussian Mixture Model (GMM) instead of a single Gaussian, MDA could explicitly capture any observed multimodality. This would allow the thresholding mechanism to adapt to more complex data distributions, providing a more principled and potentially more accurate method for disentanglement.
> >
>
> _W4: The training objectives of stage 1 are complex, and there may be conflicts between multiple losses, lacking theoretical or empirical balance analysis._
>
> We agree that managing the interplay between multiple losses is critical. We designed SUMMR with this in mind, and the apparent conflict is not a flaw but an **intentional, architecturally-managed tension** that drives the learning process.
>
> First, the synergistic nature of our four loss components is empirically validated in Table 2. **Removing any single loss** ($\mathcal L_{\rm ITC}$, $\mathcal L_{\rm GLA}$, $\mathcal L_{\rm GD}$, or $\mathcal L_{\rm LD}$) leads to a significant final performance degradation, with drops ranging from 1.6 to 5.7 points. This demonstrates that each loss contributes non-redundantly and the system relies on their combined, balanced effect.
>
> The core tension lies between the alignment objective ($\mathcal L_{\rm ITC}$) and the preservation objective ($\mathcal L_{\rm GD}$). We resolve this conflict through a two-pronged mechanism:
>
> + **The Evolutionary Mask** ($M$): This mask directs the $\mathcal L_{\rm ITC}$ loss to focus spatially only on the shared "intersection" features, preventing the alignment objective from being applied to unalignable, modality-specific content.
> + **The Projection Heads** ($W$): After the mask selects what to align, these heads learn to project the selected features into a "conflict-free" space for the contrastive loss, filtering out remaining modality-specific feature details at the feature level.

---

> ### Author Response · Authors · 2025-11-22
> **Response to Reviewer cGSf (part 3)**
>
> Together, these mechanisms disentangle the objectives, allowing the main encoder to satisfy both. This is not just a theoretical claim; it is supported by direct empirical evidence. As shown in the training curves in **Figure 4**, removing either the mask (w/o $M$) or the projection heads (w/o $W$) leads to a higher and more unstable $\mathcal L_{\rm ITC}$ loss, which is direct evidence of the unresolved optimization conflict. This conflict then translates to a significant drop in final performance (as seen in **Table 2**). Finally, our architecture is robust. The **loss weight sensitivity analysis** in **Appendix F.3** shows that the system's balance comes from this architectural design, not from fragile hyperparameter tuning.
>
> We have make this point clear in Section 4.1.4 (Lines 337-340, Page 7) as:
>
> > While the alignment objective ($\mathcal L_{\rm ITC}$) and preservation objective ($\mathcal L_{\rm GD}$) create an intentional tension, our framework resolves it architecturally. The evolutionary mask and projection heads work in tandem to disentangle these conflicting goals at the spatial and feature levels, respectively. These components form a synergistic, self-correcting loop where each part is essential, as empirically validated by our ablation studies (Tab. 2, Fig. 4).
> >
>
>
> _W5: Stage 1 requires computing all pairwise local similarities (O(n²) in patch/token count) , not provide FLOPs for Stage 1 vs. Stage 2._
>
> We acknowledge that precisely calculating training FLOPs is complex. We, therefore, **opted for a practical alternative by measuring the per-step training time to serve as a proxy for computational complexity**. We provide a detailed comparison and analysis in **Appendix J** as follows:
>
> >As can be observed in the table below, Stage 1 is intentionally more computationally intensive, primarily driven by the O(n²) pairwise local similarity calculation required by the LLD loss and the overhead of its multi-objective framework involving teacher models. In sharp contrast, Stage 2 is significantly more efficient, functioning as a lightweight fine-tuning step with a single contrastive loss; its rapid segmentation process is far outweighed by the removal of the expensive distillation objectives.
> >
>
> | Model | CLIP-SF | BGE-VL | Qwen | Llama | SUMMR-B+D S1 | SUMMR-B+D S2 | SUMMR-C S1 | SUMMR-C S2 |
> | --- | --- | --- | --- | --- | --- | --- | --- | --- |
> | Training time | 1.26 | 0.30 | 7.12 | 10.83 | 0.79 | 0.53 | 1.29 | 0.30 |
>
>
> This two-stage design represents a strategic trade-off: we invest more computation in Stage 1 to learn a robust disentanglement capability, which then enables a highly efficient and effective fine-tuning in Stage 2. Furthermore, **our approach remains significantly more efficient than VLM-based alternatives**. As also noted in **Appendix J**:
>
> >...even the more complex Stage 1 is 5-13× faster than the VLM baselines...
> >
>
> We hope this clarifies the rationale behind our design and provides a clear picture of the computational costs involved.
>
> _W6 & Q1:  SUMMR is self-supervised training on symmetric tasks, while the baseline model is supervised training on asymmetric tasks. Does this 'task misalignment' affect the fairness of comparison?_
>
>
> The "task misalignment" you've identified is not an incidental detail of our experimental setup; rather, it is a deliberate and crucial part of our evaluation designed to highlight a fundamental gap in the field that our work aims to solve.
>
>
>
> The core reason for this setup is the non-existence of large-scale, publicly available, supervised datasets for the symmetric MM2MM retrieval task. As we state in our introduction, the nuance of this task (Lines 49-52)
>
> > makes it nearly impossible to annotate at scale... a process that is prohibitively expensive and time-consuming.
> >
>
> Consequently, it is impossible to train existing supervised models on a perfectly "aligned" symmetric task. The baselines we chose, such as mmE5 and VLM2Vec, are state-of-the-art models explicitly promoted as **universal** multimodal embeddings. Our evaluation therefore provides a realistic test of their claimed universality. As stated in our experimental setup (Lines 416-419),
>
> > this experimental setup therefore provides a realistic test of their claimed universality and generalization, pitting models trained for general or asymmetric tasks against our framework designed specifically for symmetric retrieval.
> >

---

> ### Author Response · Authors · 2025-11-22
> **Response to Reviewer cGSf (part 4)**
>
> The results of this comparison are a key finding of our paper. The fact that SUMMR, a much smaller self-supervised model, significantly outperforms massive VLM-based models demonstrates that their supervised training on asymmetric tasks does not transfer effectively to the symmetric paradigm. This is further supported by our comparison between `CLIP-SF` (fine-tuned on asymmetric data) and `CLIP-SF-ZS` in Table 1. `CLIP-SF` shows only a marginal gain over its zero-shot counterpart, proving that simply using more supervised, asymmetric data yields diminishing returns. In contrast, our self-supervised SUMMR framework, which is specifically designed for the symmetric task's structure, achieves a massive +12 point improvement over the same zero-shot backbone.
>
>
>
> To further investigate if **a more "aligned" supervised approach could succeed**, we conducted an additional experiment in **Appendix I**. We fine-tuned a VLM exclusively on the synthetically generated, **symmetric** data from the mmE5 paper. This model still performed substantially worse than SUMMR. This result strongly validates our core thesis: a specialized approach is necessary. As we conclude in Appendix I,
>
> > This finding validates that a specialized, disentanglement-based approach like SUMMR is necessary to master the nuances of this task, which cannot be solved by simply scaling up naively synthesized symmetric data.
> >
>
>
> _W7: The evolutionary mask schedule ρ = 1->0 lacks theoretical justification. Why this particular annealing schedule? The paper provides no ablation on different schedules or convergence guarantees._
>
>
>
> The primary motivation for the evolutionary schedule ($\rho$ annealing from 1 to 0) is to ensure training stability. At the beginning of training, the model's encoders are not yet aligned, and the signal used to generate the mask is noisy and unreliable. Applying a hard binary mask based on this poor signal from the outset would be detrimental, as it would force the model to align based on incorrect assumptions, potentially leading to a poor local minimum. Our evolutionary schedule addresses this by transitioning smoothly from a non-informative, all-ones mask (where the model learns from all features) to the learned binary mask. This allows the model to first establish a coarse alignment and then gradually rely on its own increasingly confident predictions to perform fine-grained disentanglement.
>
>
>
> To empirically validate this, we have two key results in the paper:
>
> 1. **Robustness to the Schedule's Form:** We agree an ablation on the schedule is important. In **Appendix F.7 (Table 11)**, we compare the linear decay schedule used in our main model against a cosine decay schedule. The results, as shown below, are nearly identical, with a negligible difference in performance after Stage 1 (82.2 Avg. for linear vs. 82.3 Avg. for cosine). This demonstrates that our framework is not sensitive to the specific functional form of the decay, as long as the transition is gradual.
>
> | Schedule | R@1 | R@5 | R@10 | mR | Prec. | Avg. (Stage 1) |
> | :--- | :--- | :--- | :--- | :--- | :--- | :--- |
> | Linear decay | 68.7 | 89.3 | 89.7 | 82.6 | 81.8 | 82.2 |
> | Cosine decay | 67.2 | 88.8 | 91.1 | 82.4 | 82.2 | 82.3 |
>
>
> 2. **Necessity of the Schedule Itself:** The more critical comparison is between a gradual schedule and no schedule at all. This is precisely what our $\hat M$ ablation in **Table 2** evaluates. In this setting, we apply a static hard mask from the beginning of training ($\rho = 0$ throughout). This leads to a final performance drop of 1.7 points (84.8 Avg. vs. 86.5 for SUMMR). This result confirms that a gradual annealing process is vital for achieving optimal performance.
>
> Finally, regarding convergence, we clarify that training continues for a significant number of steps after $\rho$ has decayed to 0. This ensures that the final model fully converges while using the stable masking policy, allowing it to perfect the final representation without the auxiliary objective of mask generation. We clarified these points in **Appendix F.7**.
>
> _Q2: Can the evolution process of the mask during training be visualized? How to ensure that the mask truly captures the semantic 'intersection'?_
>
> **On the Visualization of the Mask's Evolution During Training:** The mask's evolution during training is a programmed process designed for stability. As described in Section 4.1.2, it follows an "Evolutionary Masking" strategy:
>
> > At the beginning of training, the encoders are not aligned and the estimated mask M̂ is unreliable. ... We therefore introduce an evolutionary mask M that smoothly transitions from a non-informative, all-ones mask to the model’s estimated hard mask, that is, **M = ρ1 + (1− ρ)M̂**, where **1** is an all-ones mask the annealing schedule ρ decreases from 1 to 0 during training.
> >

---

> ### Author Response · Authors · 2025-11-22
> **Response to Reviewer cGSf (part 5)**
>
> This allows the model to first learn a coarse alignment before gradually relying on its own predictions. While we do not visualize the mask's change at every step, we provide a visualization of the underlying signal that governs it in Figure 11. As its caption states, this visualization shows how the similarity scores for shared and unique features learn to separate:
>
> > Initially overlapping, the distributions progressively diverge into two distinct Gaussians. This validates the core assumption for our adaptive QDA thresholding: that the model learns a reliable, separable signal to distinguish shared (intersection) from unique (difference) features.
> >
>
> The final, converged result of this process can be seen in the heatmaps of Figures 12 and 13.
>
>
>
> **On Ensuring the Mask Captures the Semantic 'Intersection':** Ensuring this mask correctly captures the semantic 'intersection' is achieved through both our synergistic training objective and external quantitative validation. The mask generation is not an isolated step but is part of a self-correcting loop. As summarized in Section 4.1.4:
>
> > These components form a synergistic, self-correcting loop where each part is essential... Local Distillation ensures high-quality local features, which allows Global-to-Local Alignment to generate a clean signal... The Masked ITC loss leverages this mask to perform a highly focused alignment on the intersection set, which provides a powerful supervisory signal that refines the entire mask generation process.
> >
>
> This synergy is driven by our training losses. The **Local Distillation loss ($\mathcal L_{\rm LD}$)** first ensures the features are semantically rich, as it "is essential; it stabilizes training by providing a strong unimodal signal, preventing the adapters from collapsing or learning poor representations based only on noisy cross-modal signals." These high-quality features enable the **Global-to-Local Alignment loss ($\mathcal L_{\rm GLA}$)** to generate the initial signal distinguishing the intersection, based on the assumption that "local features belonging to the intersection set should have a higher similarity to the global representation of the partner modality". Finally, the **Masked Image-Text Contrastive loss ($\mathcal L_{\rm ITC}$)** provides the primary supervisory signal. By applying the contrastive loss only to the intersection, the model receives direct feedback, as this loss "provides a direct supervision signal: **“select masks MV and ML such that the remaining (intersection) information is sufficient to make the two modalities globally aligned.”**" If the mask is incorrect, this loss will be high, forcing the model to generate a better one.
>
>
>
> Beyond these internal training dynamics, we also provide **quantitative proof** of this capability. As detailed in the **new Appendix H.1**, we evaluated the model's performance on a standard segmentation benchmark:
>
> > To quantitatively verify that Stage 1 successfully learns to identify the semantic ’intersection’ between an image and text, we evaluated its disentanglement capability on the standard Referring Image Segmentation (RIS) benchmark, RefCOCO... As shown in Fig. 14, the mean Intersection over Union (mIoU) score steadily improves as training progresses. This trend provides strong quantitative evidence that our model progressively learns a more accurate representation of the semantic intersection.
> >
>
> This result, combined with our training framework, confirms that our model successfully learns to identify and refine the shared semantic concepts between modalities.
>
>
>
> _Q3: The negative samples constructed currently are generated only through 'mask difference'. Have authors considered introducing ‘semantic conflicts’?_
>
>
>
> Our full training framework does indeed address the need for "semantic conflicts" through a comprehensive negative sampling strategy in Stage 2, which combines three distinct types of negatives:
>
> 1. **Constructed Negatives (Mask Difference)**: These teach the model sensitivity to information incompleteness.
> 2. **In-Batch Negatives**: These provide diverse, random negative signals.
> 3. **Offline-Mined Hard Negatives**: This component directly introduces **semantic conflicts**. As detailed in Appendix D.4, we employ a rigorous offline mining process using powerful unimodal (BGE-m3, DINOv2) and cross-modal (CLIP, our Stage 1 model) encoders to retrieve samples that are semantically very close but not identical to the anchor. For example, for an anchor image of a "red car," our pipeline will mine hard negatives such as an image of a "blue car" (conflict in attributes) or a "red truck" (conflict in object type).

---

> ### Author Response · Authors · 2025-11-22
> **Response to Reviewer cGSf (part 6)**
>
> We have clarifed this point in Section 4.2 (Lines 395-399, Page 8) as:
>
> > The negative set is comprehensive, combining three complementary strategies: 1) constructed negatives from masking the difference set, which teach sensitivity to information deletion; 2) standard in-batch negatives for diversity; and 3) offline-mined hard negatives (details in App. D.4), which introduce semantic conflicts (e.g., content modification or addition) to complement the deletion-based negatives.
> >
>
>
>
>
> _Q4: Why masking intersection creates semantic equivalence? This seems to assume that remaining information is perfectly complementary across modalities._
>
>
>
> Our approach for creating positive samples is best understood as a novel form of **data augmentation**, analogous to standard techniques like **cropping or color jittering** in vision, but specifically designed for multimodal pairs.
>
>
>
> The goal of data augmentation in contrastive learning is **not to create a sample with identical information, but one that preserves the core semantic identity of the anchor**. A cropped image of a cat is a valid positive for the full image because it still represents a "cat." Our method applies this principle to multimodal pairs.
>
>
>
> When we mask the shared intersection from one modality (e.g., the "black bear" region in Fig. 3), the core concept is preserved because that essential information is still present and accessible in the other modality (the text). The resulting pair still represents the same holistic idea of "a black bear greeting visitors," making it a strong positive sample, even if the pixel-level information has changed.
>
>
>
> This is fundamentally different from masking the unique difference (e.g., the text "greets visitors" in Fig. 3). In this case, the information is **irrecoverably lost** from the pair, as it is not present in the other modality. This breaks the core semantic equivalence and creates a valid negative. Therefore, **our method does not assume perfect complementarity, but rather uses the principle of information recoverability to generate semantically consistent positive pairs**, a standard and effective practice in self-supervised contrastive learning.
>
>   _Q5: In Section 4.2, have authors used or compared existing segmentation methods (SAM) when using segmentation for image masking?_
>
>
>
> To address your comment, we have included the ablation study in our paper. We believe quoting the relevant text and results directly from the manuscript will be most helpful.
>
> In Section 5 of our paper, under "Ablation Studies", we briefly summarize this finding:
>
> > Furthermore, as shown in the "SAM" ablation (Tab. 3), our simple clustering-based segmentation performs comparably to using a SOTA model like SAM, justifying our efficient design choice.
> >
>
> For a more detailed explanation, we direct you to **Appendix F.6**, which is dedicated to this specific comparison. We quote it here in full for your convenience:
>
> > **F.6 IMPACT OF IMAGE SEGMENTATION METHOD IN STAGE 2**
> >
> > To justify our choice of a lightweight, clustering-based segmentation method for image masking in Stage 2, we conducted an ablation study comparing it against a state-of-the-art general-purpose segmentation model, Segment Anything Model (SAM) (Kirillov et al., 2023). While integrating a powerful model like SAM might seem like a straightforward path to improvement, our results in **Table 10 (as shown below)** demonstrate that this is not the case. Our simpler, iterative hierarchical clustering method not only outperforms the SAM-based approach across all key metrics but is also significantly more efficient, reducing per-step training time by over **3× (2.84s vs. 9.24s)**.
> >
>
> > We attribute this seemingly counter-intuitive result to the specific goal of segmentation within our framework. **The purpose is not to achieve pixel-perfect segmentation, but rather to generate coarse, semantically coherent regions that can be manipulated to construct meaningful positive and negative training samples.** Here, SAM’s primary strength—its ability to produce highly detailed, fine-grained masks—becomes a liability. It often fragments a single semantic object (e.g., a "bear") into multiple distinct parts (e.g., head, torso, paws). Masking just one of these small fragments is often insufficient to meaningfully alter the image’s core semantics, leading to the creation of weak or ambiguous training signals.
> >
>
> > In contrast, our clustering-based approach naturally groups visually and semantically similar patches, producing coarser segments that better align with whole objects or large object parts. This makes it far more effective for our purpose: masking an entire "bear" segment creates a strong positive sample (as the concept is recoverable from the text), whereas masking a background segment creates a clear hard negative.
> >

---

> ### Author Response · Authors · 2025-11-22
> **Response to Reviewer cGSf (part 7)**
>
> > Therefore, this ablation confirms that our lightweight segmentation method is not merely a compromise for efficiency; it is functionally superior for the specific task of self-supervised sample generation in our framework.
> >
>
> | Method | R@1 | R@5 | R@10 | mR | Prec. | Avg. | Per-step Training Time (s) |
> | :--- | :--- | :--- | :--- | :--- | :--- | :--- | :--- |
> | SAM | 70.6 | 90.2 | 91.1 | 84.0 | 83.4 | 83.7 | 9.24 |
> | Ours | 72.9 | 93.9 | 95.8 | 87.5 | 85.5 | 86.5 | 2.84 |

---

> > ### Comment · Reviewer_cGSf · 2025-11-28
> >
> > I thank the authors for their detailed and comprehensive responses. The revisions and clarifications have addressed the majority of my initial concerns. After re-evaluating the work, I am prepared to assign a higher rating.

---

### Official Review · Reviewer_36gh · 2025-11-09

**Soundness:** 2
**Presentation:** 3
**Contribution:** 2
**Rating:** 4
**Confidence:** 4

**Summary:**

The authors study the ([symmetric] multimodal-to-multimodal) MM2MM retrieval problem, where multi-modal queries and contexts have a bidirectional relationship (i.e., contexts retrieved with a query can be used as a query such that the resulting context may be the original query) -- which has some interesting use cases (c.f. Figure 1). To accomplish this task, the authors develop a late-fusion architecture and self-supervised learning strategy (that doesn't require any supervised pairs) to train SUMMR, a MM2MM retrieval system. The two stages of SUMMR are: (1) learning an "intersection mask" from image-text pairs capable of disentangling the shared semantic concepts from the modality specific details from an image-text pair (lines 213-215; Figure 3 -- has many technical details) and (2) leveraging this mask to generate positive and hard-negative examples for contrastive learning (line 215). Additionally, the create a new benchmark (sym-MM2MM) to evaluate the MM2MM task with a pipeline of: (1) using source images from {COCO, LAION, WuKong}, generate a fine-grained description using GPT-4o; (2) use GPT-4o to rewrite the description to create a modified description to create a controlled information gap; and (3) use Stable Diffusion (SD) 3.5 to synthesize a new image from the modified description (resulting in a positive pair and a hard negative pair that are validated with human annotation). Training SUMMR on 800k image-text pairs from LAION-5B, experiments are conducted to demonstrate improvements over supervised encoder-based methods (e.g., CLIP) and supervised VLM-based (e.g., MM-Emded, VLM2Vec, LamRA, UniME, mmE5) with different SUMMR backbones in terms of recall and precision based metrics (noting parameter efficiency). Ablations are performed to demonstrate the relative contribution of different SUMMR components and self-supervised sample construction strategy. Additionally ablations and sensitivity analyses are performed in the appendices.

**Strengths:**

Strengths of this paper include:
- The MM2MM retrieval task isn't well-studied and has some interesting applications. Additionally, in these applications where query-specific instructions aren't needed, the resulting models can likely be significantly smaller (as encoding-based models are likely sufficient).
- The paper is clear overall and has sufficient architectural details and rationales for design choices that with the code, I believe I could: (1) understand and reproduce many of the results and (2) explore additional directions for improving the model.
- For the most part, the empirical performance is positive and the ablation studies address the key points introduced in the paper.

**Weaknesses:**

Weaknesses of the paper include:
- The sym-MM2MM benchmark isn't particularly strong. As best as I can tell from the anonymous repository, it is 214 queries (that generate a positive and hard negative match) -- which I actually couldn't find in the paper. Additionally, the sym-MM2MM process includes a rewrite of the caption that is 'worse', making the secondary image a perturbation that doesn't 'improve' the image and is likely to be reflective of biases in GPT-4o or the associated prompt. In the same vein, this obviously heavily relies on strong VLLMs and human annotation. I think the data is likely good enough to support the experiments (even if the confidence intervals are likely larger than implied and is somewhat designed in line with the SUMMR model), but it isn't a very strong independent contribution.
- While a different problem, there isn't any cross-modal comparison with existing datasets. First of all, these would likely be interesting (but I don't believe required). More importantly, without showing this, I don't understand how this work "breaks the deadlock" (line 76) of "the field is stuck" (line 73) for more commonly studied multimodal retrieval settings.
- A bit of a nit, but SUMMR seems pretty specific to image-text settings; this is at least worth discussing.
- What would an "adapted VLM" method for this task look like? If it is a straightforward extension, it is a nice contribution and likely to perform better (albeit I might be missing something).
- I would recommend referencing even more of the results from the Appendices (e.g., sensitivity analyses) as I had more criticisms until I decided to go back and read the appendices.

**Questions:**

Reframing my 'weaknesses' as questions:
- How might you go about improving sym-MM2MM. What are the limitations and can they be resolved (and do you have plans to do so)?
- What results do you get for cross-modal settings (maybe with constrained instructions, a subset of MMEB)? Even only comparing to supervised encoder-based methods would be interesting -- even if worse, knowing how much would be useful.
- Is SUMMR easily extended to other modalities such that there could be an evaluation for MMEB-v2?
- What would an "adapted VLM" method for MM2MM look like?

---

> ### Author Response · Authors · 2025-11-21
> **Response to Reviewer 36gh (Part 1)**
>
> _W1: sym-MM2MM benchmark rely on heavily relies on strong VLLMs and human annotation and is somewhat designed in line with the SUMMR model. What are the limitations and can they be resolved?_
>
> We agree that creating a robust benchmark for the nascent symmetric MM2MM task is a significant challenge. We would like to address the specific concerns raised.
>
> 1. **On Benchmark Size and Strength:** We would like to gently clarify that the dataset size of 214 triplets is mentioned in Section 3 (Line 183). The modest size is a direct reflection of **the core challenge we highlight in our paper: high-quality, human-annotated sym-MM2MM data is extremely difficult and expensive to create**. As noted, our two annotators spent nearly 40 hours collectively to generate just these 214 high-quality triplets. This process involves not just labeling, but nuanced creative tasks like text rewriting and image replacement to ensure semantic consistency for positive pairs and subtle but critical inconsistencies for hard negatives. Our goal was a high-quality dataset for precise evaluation, not a massive one. **To further ensure the benchmark’s robustness despite its modest size, we also focused on its diversity; as now illustrated in the newly added Fig. 6, our curated samples span 17 distinct categories.** We agree its size may lead to larger confidence intervals and have clarified this **in Appendix J (Limitations and Future Work)** in the original submitted version.
> 2. **On Generation Process and Bias:** We clarify that the caption rewrite is not meant to "improve" content but to create a **controlled information gap** for constructing challenging positive/negative pairs. We acknowledge the potential for VLM/SD bias, which is precisely why we include a crucial human verification step. As noted on Line 176, annotators **rejected nearly 50% of generated pairs** and manually refined others to ensure quality. This hybrid approach, precedented by works like mmE5 (Chen et al., 2025), balances scalable generation with the quality assurance that only human judgment can provide.
> 3. **On Independence from SUMMR:** The benchmark was developed **entirely independently** of our SUMMR model. The construction was guided only by the formal task definition, and the annotators were firewalled from our model's development to prevent any bias.
> 4. **On Overall Contribution:** We agree the benchmark is a supporting contribution, designed to enable rigorous evaluation in the absence of any alternative for this newly formalized task. By being the first to provide a curated dataset and a public creation pipeline, we offer a valuable, albeit initial, resource for the community and a foundation for future work, as discussed in the revised paper.
>
>
>
>
>
> _W2 & Q2: What's the performance of cross-modal task?_
>
>
>
> First, we wish to clarify our claim of "breaking the deadlock." The deadlock we refer to (lines 79-82) is the field's methodological reliance on expensive supervised data for complex multimodal retrieval, a problem that is particularly acute for novel paradigms like symmetric MM2MM where no large-scale labeled datasets exist. Our work breaks this _data-annotation deadlock_ by introducing a self-supervised paradigm that achieves state-of-the-art performance on this specific task using only unlabeled web data. Our goal is to demonstrate a viable path forward for this new task, not to claim universal superiority across all retrieval settings.
>
>
>
> To address your request for **cross-modal evaluation**, we tested our model on the standard COCO text-image retrieval task and compared it with established CLIP baselines. **We have added these findings to a new appendix section** (**Appendix H.2**).
>
> | Method | R@5 |
> | :--- | :--- |
> | SUMMR-B+D Stage 1 | 30.3 |
> | SUMMR-B+D Stage 2 | 1.1 |
> | CLIP-ZS (ViT-B) | 55.3 |
> | CLIP-ZS (ViT-L) | 70.0 |
> | CLIP-FT (ViT-L) | 85.1 |
>
>
> These results provide two key insights into our two-stage design:
>
> 1. **Stage 1 Learns Efficient General-Purpose Alignment:** After Stage 1, our model achieves a 30.3 R@5. While lower than zero-shot CLIP (CLIP-ZS)—which was pre-trained on 500x more data (400M vs. our 0.8M pairs)—this result demonstrates the data efficiency of our learning objective, , and this performance may scale favorably with more training data. This general alignment is learned through the synergy of our loss functions: the **Masked Image-Text Contrastive loss ($\mathcal L_{\rm ITC}$)** directly aligns shared semantic concepts, while the **Global Distillation loss ($\mathcal L_{\rm GD}$)** preserves complete unimodal information, acting as a powerful regularizer. As shown in Appendix H.2, this combination produces robust, general-purpose encoders without massive-scale pre-training, achieving a surprisingly strong and balanced performance across various unimodal and cross-modal tasks.

---

> ### Author Response · Authors · 2025-11-21
> **Response to Reviewer 36gh (Part 2)**
>
> 2. **Stage 2 Specializes the Model for Symmetric MM2MM Retrieval:** The sharp performance drop on COCO after Stage 2 is an expected and informative result. Stage 2 is a short, focused fine-tuning phase on data constructed exclusively for the novel _symmetric MM2MM_ task. This process successfully **specializes** the model to discern the compositional semantics of _multimodal pairs_, delivering the SOTA performance on our target task (e.g., a +4.3pt boost in Table 3). The trade-off is a reduced capability on general cross-modal retrieval, which confirms that our framework effectively distills general representations into a powerful, specialized encoder for a distinct and challenging problem.
>
> _W3 & Q3: Could SUMMR be extended to other modalities such that there could be an evaluation for MMEB-v2?_
>
>
> The fundamental principles of the SUMMR framework are indeed **modality-agnostic and not limited to the image-text domain.**
>
>
>
> The reason for its generalizability is that SUMMR's core mechanism—disentangling shared "intersection" from modality-specific "difference" to enable self-supervised learning—is a **conceptual** one. This logic is not tied to the visual domain but can be applied to any pair of modalities as long as two conditions are met: 1) large-scale, unlabeled paired data exists (e.g., text-audio, text-video), and 2) strong unimodal encoders are available to extract meaningful features. The primary adaptation would be in the engineering of the modality-specific masking/segmentation strategy (e.g., identifying spatio-temporal "tubes" in video or relevant segments in an audio spectrogram), but the self-supervised training paradigm itself would remain the same.
>
> To formally integrate this discussion into the paper and highlight the future potential of our paradigm, we have revised the "Limitations and Future Work" section. We include the updated text below:
>
> >Another promising direction for future work is to extend our framework to other pairings, such as text-audio and text-video, by leveraging large-scale unlabeled datasets (e.g., AudioSet, HowTo100M) and strong unimodal encoders (e.g., AST, VideoMAE). This would involve adapting the masking mechanism to the temporal nature of audio and video data but would validate SUMMR's potential as a general paradigm for complex retrieval tasks, making it a strong candidate for emerging benchmarks like a potential MMEB-v2.
> >
>
>
> _W4 & Q4:  What would an "adapted VLM" method for MM2MM look like?_
>
> Thank you for this insightful question. An "adapted VLM" method for this task involves two components: an inference-time strategy and a training data strategy.
>
> For inference, we would like to clarify that for all VLM-based baselines in our original submission, we already employed an inference-time adaptation strategy to ensure a fair and robust comparison. Specifically, we applied a consistent instruction, '**Represent the given image with related text information:**', to both the query and candidate pairs, as mentioned at the end of **Appendix D.2**. This prompt encourages the VLM to generate a holistic joint representation suitable for the symmetric MM2MM task. This adaptation is effective, boosting VLM2Vec's (Jiang et al., 2025) average score from 67.76 to 71.96 on our sym-MM2MM benchmark, and these improved scores are the ones reported in our main results table.
>
> The more significant challenge is training, as no large-scale, annotated symmetric MM2MM dataset exists. This forces VLMs to be trained on general or asymmetric benchmarks. The VLM-based mmE5 (Chen et al., 2025) attempts a more direct extension by synthesizing a large volume of partially symmetric data, creating pairs by retrieving a visually similar image and generating corresponding texts. However, it still underperforms SUMMR, probably because it also involves asymmetric data.
>
> To isolate the effect of this approach, we conducted a new experiment training a VLM solely on mmE5's synthesized symmetric data. The resulting average score was only **69.39**, even lower than models trained on the more general MMEB benchmark. This key finding suggests that synthesizing data based on simple visual similarity is insufficient for learning the strict semantic interchangeability required for symmetric retrieval. In contrast, SUMMR's self-supervised framework learns to disentangle the shared ("intersection") and unique ("difference") information within a single image-text pair, enabling it to generate training samples that are **truly semantically symmetric**.
>
> Our results demonstrate that these naive or straightforward adaptations fall short, validating the need for a specialized framework like SUMMR that is designed from the ground up to learn the disentangled representations crucial for this task. We have included these dicussions in **Appendix I**.

---

> ### Author Response · Authors · 2025-11-21
> **Response to Reviewer 36gh (Part 3)**
>
> _W5: Reference more appendix results._
>
>
> We sincerely thank the reviewer for their thorough evaluation of our work and for taking the time to read the appendices in detail. Your observation that referencing even more of the results from the appendices would strengthen the main paper is very insightful, and we appreciate this constructive suggestion.
>
> In our initial draft, we aimed to link the main body to the rich details in the appendices. As your feedback prompted us to re-verify, we found the following references were already present throughout the main text:
>
> + **Section 2 (Related Works):** A reference to **App. A** for a more detailed literature review (Line 151).
> + **Section 3 (Benchmark):** A reference in the Figure 2 caption to **App. B** for more examples from our benchmark (Line 130).
> + **Section 4 (Methodology):**
>     - A reference to **App. C.1** for the detailed derivation of the QDA threshold (Line 293).
>     - A reference to **App. C.2** for the details of our image segmentation algorithm (Line 373).
>     - A reference to **App. D.4** for the details of our offline-mined hard negative strategy (Line 398).
> + **Section 5 (Experiments):**
>     - A general reference to **App. D** for implementation details and baselines (Line 419).
>     - A general reference to **App. F** for more details on the ablation studies (Line 448).
>     - A reference to **App. F.2 and F.3** (Line 474), which was already accompanied by a summary of the sensitivity analyses, confirming the model's robustness to key hyperparameters.
>     - A specific reference to **App. F.5** and **Table 9** when discussing the large performance gain from Stage 2 on the SUMMR-C model (Line 487).
>     - Implicit references to **App. G** by citing Figures 12, 13, and 14 (which are located in App. G) in the Visualization subsection (Lines 522-524).
>
> Following your suggestion, we have made the following targeted revisions to bring the conclusions of key analyses from the appendices directly into the main text, making the connections more explicit and impactful:
>
> 1. In Section 4.1.2 (Line 300), we have added a reference to **App. F.1** to further support our choice of the evolutionary masking strategy.
> 2. In the initial discussion of the Ablation Studies (Sec. 5, Line 463), we added a direct reference to **Table 5**, allowing readers to immediately consult the full metrics for the "$\mathcal L_{\rm ITC}$ only" baseline.
> 3. In the Stage 2 Ablation discussion (Sec. 5, Line 498), we now quantify the impact of a crucial component previously only detailed in App. F.4. We added a sentence stating that removing only the offline-mined hard negatives leads to a substantial **5.45-point performance drop**, providing stronger, more immediate justification for our comprehensive negative sampling strategy.
> 4. In the same Stage 2 Ablation discussion (Sec. 5, Line 501), we now explicitly reference the "SAM" ablation result from **Table 3** to justify our efficient segmentation choice. We added a sentence clarifying that our simple method performs comparably to a complex SOTA model (SAM), addressing potential questions about this design choice without requiring a reader to consult the full appendix.
> 5. In the Visualization subsection (Sec. 5, Line 523), we added a sentence summarizing findings from **App. G.3** regarding the model's robustness to geometric transformations and sensitivity to semantic changes, further illustrating the meaningfulness of the learned embeddings.

---

### Author Response · Authors · 2025-11-22
**Response to All Reviewers**

We sincerely thank all the reviewers for their constructive feedback and valuable suggestions, which significantly contribute to improving the quality of our work. A comprehensive reply has been prepared to address every point raised by each reviewer, respectively. The reviewer's comments are presented in italics, followed by our response. Quotations from the revised paper are included in markdown quotation mode. Below, we summarize the revisions made and the additional experiments conducted to address the reviewers' concerns:

In response to the reviewers’ comments, we conduct 8 new experiments and add 4 new tables and 4 new figures to further validate our claims. These updates include:

1. **Table 1:** We add the FPS metric, which demonstrate that SUMMR achieves a fast and competitive inference speed.
2. **Table 3 and Table 10:** We conducted a new ablation study investigating the use of SAM as the segmentation method in Stage 2. These results validate the advantage of our proposed segmentation approach.
3. **Figure 6:** We add a new figure to show the distribution of categories for _sym-MM2MM_, which demonstrate the diversity of _sym-MM2MM_.
4. **Table 11:** We conduct ablation study on the schedule for evolutionary mask in stage 1, which demonstates it is not sensitive to the schedule used.
5. **Figure 14:** We now provide more visualizations of the global-to-local similarity, specifically for challenging multi-object cases.
6. **Figure 15:** We conduct a new experiment to study the sensitivity to edge semantics.
7. **Figure 16:** We conduct for systematic measurements for disentanglement on referring image segmentation benchmark RefCOCO.
8. **Table 12:** We conduct cross-modal evaluation on COCO, which is a subset of MMEB.
9. **Table 13:** We measuring the per-step training time to serve as a proxy for computational training complexity, it shows SUMMR also has a relatively low training overhead.
10. **Appendix I:** We finetune a VLM exclusively on the publicly available synthesized symmetric data from mmE5 as a straightforward adaption for symmetric MM2MM task.

In addition to these specific additions, we have carefully revised the entire draft for clarity. We believe these comprehensive updates and new experimental results substantially strengthen our paper and thoroughly address the reviewers' concerns.

---

### Meta-Review · Area_Chair_V68b · 2025-12-22

**Summary:**

This paper explores the unsupervised symmetric multimodality-to-multimodality retrieval. The reviewers raised lots of questions, involving in dataset quality, experimental setup, and the need for more robust comparisons and analyses.

1. Weak Benchmark and Data Quality: The sym-MM2MM benchmark is not strong, with unclear details on the 214 queries and the quality of the rewritten captions. The reliance on VLLMs and human annotation raises concerns about bias and robustness.
2. Lack of Cross-Modal Comparison: The paper does not compare with existing datasets or explain how the work "breaks the deadlock" in multimodal retrieval.
3. No Comparison to Baselines: The paper lacks comparisons to standard methods like CLIP with symmetric loss or random masking.
4. Complexity & Robustness: Stage 1 requires O(n²) pairwise similarity computations, but FLOP comparisons between Stage 1 and Stage 2 are not provided. Stage 1’s complex objectives may have conflicts between multiple losses, with no empirical or theoretical analysis of their balance.
5. Task Misalignment: The self-supervised SUMMR vs. supervised baselines may create a fairness issue due to task misalignment.
6. Real-World Benchmarks: Real-world scenarios, like e-commerce retrieval, are not tested, and the disentanglement validation is indirect and lacks systematic measurements.
7. Performance with Strong Models: The lack of comparisons to fine-tuned models like CLIP or mmE5 on the proposed benchmark makes it hard to assess the true performance gain of the self-supervised approach.
8. Robustness Concerns: The risk of error propagation between stages is not analyzed, potentially compromising the quality of the final samples.

**Reviewer Concerns:**

The authors claimed to address comparison, benchmark details, complexity, and robustness issues, but there are many remaining issues that have not been fully addressed.

**Reviewer Scores:**

We received feedback from four reviewers on this paper, with recommendation scores of 4 (marginally below acceptance), 2 (reject), 6 (marginally above acceptance), and 6. Reviewer cGSf (score 2) suggested increasing the score, but many of the concerns remain unresolved in its current form. In my view, there are already several symmetric learning schemes in multi-modal learning, and we use asymmetric learning to simplify pairwise similarity learning. Overall, the novelty of this paper is not strong enough to warrant acceptance in such a competitive year for ICLR.

---

### Decision · Program_Chairs · 2026-01-26

Reject